# Learnable Kernel Density Estimation for Graphs

## Abstract

This work proposes a framework LGKDE that learns kernel density estimation for graphs. The key challenge in graph density estimation lies in effectively capturing both structural patterns and semantic variations while maintaining theoretical guarantees. Combining graph kernels and kernel density estimation (KDE) is a standard approach to graph density estimation, but has unsatisfactory performance due to the handcrafted and fixed features of kernels. Our method LGKDE leverages graph neural networks to represent each graph as a discrete distribution and utilizes maximum mean discrepancy to learn the graph metric for multi-scale KDE, where all parameters are learned by maximizing the density of graphs relative to the density of their well-designed perturbed counterparts. The perturbations are conducted on both node features and graph spectra, which helps better characterize the boundary of normal density regions. Theoretically, we establish consistency and convergence guarantees for LGKDE, including bounds on the mean integrated squared error, robustness, and generalization. We validate LGKDE by demonstrating its effectiveness in recovering the underlying density of synthetic graph distributions and applying it to graph anomaly detection across diverse benchmark datasets. Extensive empirical evaluation shows that LGKDE demonstrates superior performance compared to state-of-the-art baselines on most benchmark datasets.

## 1 Introduction

Graphs serve as powerful representations for modeling complex relationships and interactions in numerous domains (Wu et al., 2020; Kipf & Welling, 2017; Hamilton et al., 2017; Errica et al., 2020; Nachman & Shih, 2020; Muzio et al., 2020; Rong et al., 2020; Jin et al., 2021b). The prevalence of graph-structured data has led to significant advances in graph learning, particularly in tasks such as node classification, link prediction, and graph classification (Liu et al., 2023a; Wu et al., 2020).

In this paper, we tackle the fundamental challenge of **modeling the probability density function of graph-structured data**, which serves as a cornerstone for identifying anomalous patterns in graph collections. The significance of this problem spans across numerous real-world applications: from detecting fraudulent communities in social networks (Akoglu et al., 2015; Ding et al., 2019) to identifying rare molecular structures in drug discovery pipelines (Ma et al., 2021; Shen et al., 2024). Beyond these domains, accurate density estimation of graphs has proven invaluable in biological research, particularly in uncovering novel protein structures that could reveal critical biological mechanisms (Lanciano et al., 2020). The ubiquity and complexity of these applications underscore the pressing need for effective graph density estimation methods that can capture both structural and semantic patterns while maintaining computational efficiency.

Traditional approaches to graph density estimation primarily rely on graph kernels combined with kernel density estimation (KDE) (Vishwanathan et al., 2010). These methods define similarity measures between graphs using various kernels, such as shortest-path kernels (Borgwardt & Kriegel, 2005), Weisfeiler-Lehman subtree (WL) kernels (Shervashidze et al., 2011), and propagation kernel (PK) (Neumann et al., 2016). However, these approaches face several limitations: i) handcrafted graph kernels may fail to capture complex structural patterns; ii) the computational complexity of kernel computation often scales poorly with graph size; and iii) the fixed bandwidth in traditional KDE may not adapt well to the varying scale of graph patterns.

Figure 1 shows t-SNE (Van der Maaten & Hinton, 2008) visualizations of kernel matrices computed using different methods on the MUTAG dataset. Traditional graph kernels like WL and PK struggle

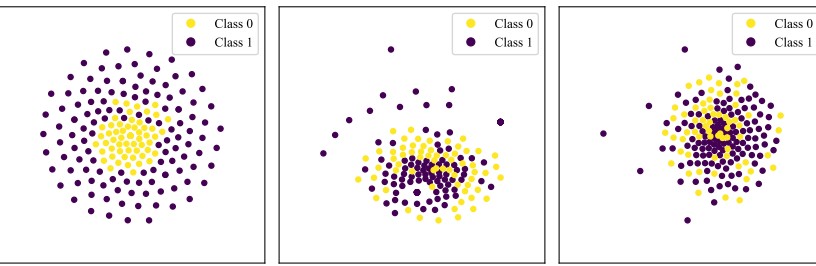

Figure 1: t-SNE visualization of learned kernel matrix on MUTAG dataset. Left: LGKDE learned, Middle: WL Kernel, Right: Propagation Kernel

to effectively separate graphs from different classes while our LGKDE learned kernel achieves clear separation between classes while maintaining smooth transitions in the metric space. This comparison underscores the importance of learning adaptive kernel functions that can go beyond purely structural similarities to capture semantic patterns that distinguish functionally different graphs.

More recent methods leverage unsupervised representation learning followed by density estimation. Graph neural networks (GNNs) (Kipf & Welling, 2017; Xu et al., 2019) have been employed to learn graph embeddings, which are then used for density estimation or anomaly detection. Several end-to-end approaches have been proposed, such as graph variational autoencoders (Kipf & Welling, 2016), one-class graph neural networks (Zhao & Akoglu, 2021), and contrastive learning-based methods (Qiu et al., 2022; Ma et al., 2022).

However, these methods also have limitations:
i) they often make strong assumptions about the shape of the normal data distribution (e.g., hypersphere); ii) the learned representations may not preserve important structural information; and iii) the lack of theoretical guarantees makes it difficult to understand their behaviors.

To address the challenges, we propose Learnable Kernel Density Estimation for Graphs (LGKDE), a principled framework that bridges deep graph representation learning with adaptive kernel density estimation. LGKDE learns a multi-scale density estimator in a deep Maximum Mean Discrepancy (Gretton et al., 2012) (MMD) based metric space while simultaneously refining graph embeddings through density contrasting. Our main contributions are as follows:

- We propose a novel deep learning method to estimate the densities of graphs. The key technical contribution includes a novel graph density formulation, a novel density contrasting objective, and a structure-aware graph perturbation strategy. Note that the study on density estimation for graphs is rare in the literature.
- We establish comprehensive theoretical guarantees for our density estimator, including consistency, convergence, robustness, and generalization, which are supported by extensive empirical validation on synthetic graph distribution recovery and superior performance in graph anomaly detection.

## 2 RELATED WORK

This section provides a thorough literature review on density estimation for graphs and graph anomaly detection. Due to space limits, the extended related literature review on graph representation learning and additional discussion are presented in Appendix C

### 2.1 DENSITY ESTIMATION ON GRAPHS

Graph density estimation presents unique challenges due to graph structures' discrete and combinatorial nature. Traditional approaches primarily rely on graph kernels (Vishwanathan et al., 2010), which define similarity measures between graphs through various structural features. Representative examples include the random walk kernel (Kashima et al., 2003), shortest-path kernel (Borgwardt & Kriegel, 2005), and Weisfeiler-Lehman subtree kernel (Shervashidze et al., 2011). These kernels or the embeddings computed from the kernel matrices, combined with kernel density estimation (KDE), provide a principled way to model graph distributions. However, they face several limitations: i) the fixed kernel design may not capture complex structural patterns; ii) the computational complexity typically scales poorly with graph size; and iii) the bandwidth selection in KDE remains challenging for graph-structured data.

Recent advances in deep learning have inspired new approaches to graph density estimation. Graph variational autoencoders (Kipf & Welling, 2016) attempt to learn a continuous latent space where density estimation becomes more tractable. Flow-based models (Liu et al., 2019) and energy-based models (Liu et al., 2020) offer alternative frameworks for modeling graph distributions. However, these methods often make strong assumptions about the underlying distribution or struggle with the discrete nature of graphs. Moreover, the lack of theoretical guarantees makes it difficult to understand their behavior and limitations (detailed in Appendix C.2), particularly in the context of anomaly or outlier detection (Pang et al., 2021; Jin et al., 2021a; Liu et al., 2023b; Cai et al., 2024).

## 2.2 Graph Anomaly Detection

Graph anomaly detection has attracted significant attention due to its broad applications in network security, fraud detection, and molecular property prediction (Akoglu et al., 2015; Ma et al., 2021). Early approaches primarily focus on node or edge-level anomalies within a single graph (Ding et al., 2019), while graph-level anomaly detection presents distinct challenges in characterizing the normality of entire graph structures (Zhao & Akoglu, 2021). Recent studies have further expanded this field into graph-level out-of-distribution (OOD) detection, which aims to identify whether a test graph comes from a different distribution than the training data (Liu et al., 2023a; Kim et al., 2024).

Recent developments in deep learning have led to several innovative approaches. One prominent direction adapts deep one-class classification frameworks to graphs. For instance, OCGIN (Zhao & Akoglu, 2021) combines Graph Isomorphism Networks with deep SVDD to learn a hyperspherical decision boundary in the embedding space. OCGTL (Qiu et al., 2022) further enhances this approach through neural transformation learning to address the performance flip issue. Another line of research leverages reconstruction-based methods, where graph variational autoencoders (Kipf & Welling, 2016) or adversarial architectures are employed to learn normal graph patterns.

Knowledge distillation and contrastive learning have emerged as powerful tools for graph anomaly detection. GLocalKD (Ma et al., 2022) distills knowledge from both global and local perspectives to capture comprehensive normal patterns. Recent works like iGAD (Zhang et al., 2022) propose dual-discriminative approaches combining attribute and structural information, while CVTGAD (Li et al., 2023) employs cross-view training for more robust detection. SIGNET (Liu et al., 2023b) proposes a self-interpretable approach by introducing a multi-view subgraph information bottleneck for detecting anomalies. In the realm of graph OOD detection, methods such as GOOD-D (Liu et al., 2023a) and GraphDE (Li et al., 2022) have been developed to handle distribution shifts in graph data, demonstrating the close connection between anomaly detection and OOD detection tasks. Despite the advances, existing methods face limitations: i) strong assumptions about the distribution of normal graphs (e.g., hyperspherical or Gaussian); ii) limited theoretical understanding of their behavior; iii) challenges in handling the heterogeneity and non-IID nature of graph data distributions (Kairouz et al., 2021; Xie et al., 2021; Cai et al., 2024; Song et al., 2025).

**Bridging Density Estimation and Anomaly Detection.** Anomaly detection serves as a principal evaluation and application approach in the density estimation literature (Parzen, 1962; Beckman & Cook, 1983; Barnett et al., 1994). The intrinsic connection between anomaly detection and density estimation lies in the observation that normal patterns typically concentrate in high-density regions of the data distribution. This well-established principle (Breunig et al., 2000; Schölkopf et al., 2001; Kim & Scott, 2012) suggests that effective density estimation naturally enables anomaly detection. However, a critical gap persists at the intersection of density estimation and the learning on graph data: traditional graph density methods (e.g., graph kernels) lack the flexibility to model complex distributions, while modern deep GAD methods often bypass explicit density modeling, sacrificing theoretical guarantees for empirical performance. This highlights **a compelling need for a framework that unifies the expressive power of deep learning with the principled foundation of density estimation for graphs.**

## 3 Methodology

### 3.1 Problem Definition

Consider a collection of normal graphs $\mathcal{G} = \{G_1, ..., G_N\}$, where each graph $G_i = (V_i, E_i, \mathbf{X}_i)$ consists of a node set $V_i$, an edge set $E_i$, and node features $\mathbf{X}_i \in \mathbb{R}^{|V_i| \times d}$. We denote the adjacency matrix of $G_i$ as $\mathbf{A}_i$. Our goal is to learn a density estimator $f : \mathbb{G} \to \mathbb{R}_+$ that maps graphs to non-negative density values, capturing the underlying distribution of $\mathcal{G}$. Note that $\mathbb{G}$ denotes the set of all graphs in the form of $G = (V, E, \mathbf{X})$, where the $\mathbf{X}$ are drawn from some continuous distribution.

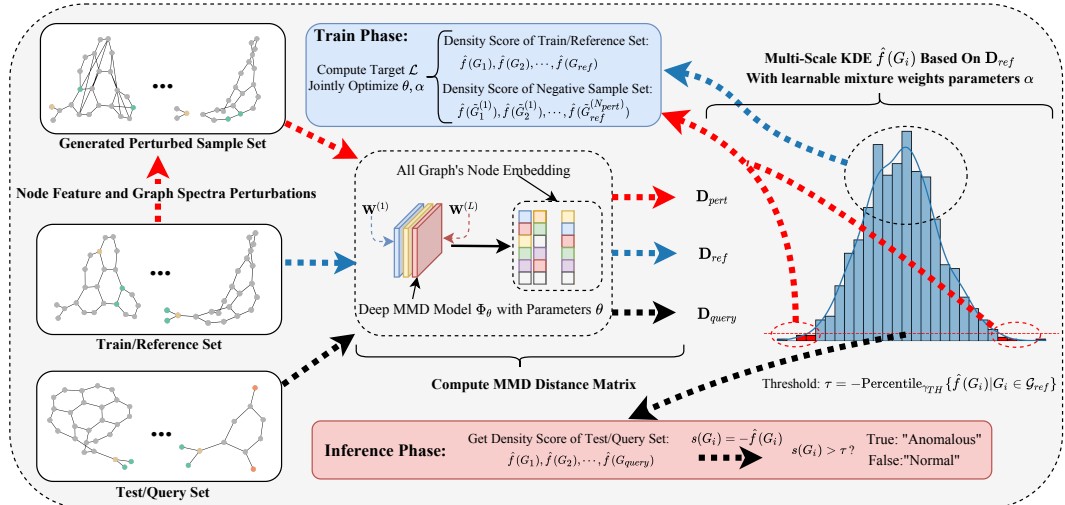

Figure 2: Framework of our proposed LGKDE

The key technical challenges stem from several unique characteristics of graph data:

- **Distribution Complexity**: Graph-structured data exhibits complex patterns at multiple scales and can undergo various types of distributional shifts. These include structural variations (from local substructures to global topological properties), size shifts (varying numbers of nodes and edges), and feature shifts (changes in node attributes), requiring robust and adaptive modeling approaches.
- **Isomorphism Invariance**: The density estimator must be invariant to node permutations while remaining sensitive to structural differences that indicate anomalies.
- **Limited Supervision**: The unsupervised nature of the problem requires learning meaningful representations and density estimates without access to labeled anomalies.

These challenges motivate our development of LGKDE, **a scalable kernel density estimation framework that effectively models the distributions of graphs through deep learning.**

### 3.2 OVERVIEW OF LGKDE FRAMEWORK

Since graphs are non-Euclidean data and often very complex, it is non-trivial to train a deep learning model to estimate the density of graphs when there is no available supervision information. If we directly maximize the density of all graphs, the model will collapse and all graphs will have the same embeddings in the latent space, leading to an identical density for all graphs. To address the challenges, we propose to perturb the training graphs and train the model under the principle that the density of a perturbed graph is often lower than that of the original graph.

Formally, our framework consists of three main components (Figure 2): (1) a deep graph MMD model that learns meaningful distances between graphs; (2) a multi-scale kernel density estimator with learnable weights that adaptively captures density patterns at different scales; and (3) a structure-aware sample generation mechanism that enables contrastive learning to refine the learned representations.

The whole method is formulated as

$$\max_{\boldsymbol{\theta}} \sum_{i=1}^{N} \sum_{j=1}^{N_{\text{pert}}} \frac{p_{\boldsymbol{\theta}}(G_i) - p_{\boldsymbol{\theta}}(\tilde{G}_i^{(j)})}{p_{\boldsymbol{\theta}}(G_i)} \tag{1}$$

where $p_{\boldsymbol{\theta}}(G)$ denotes the probability density of a graph $G$, $\boldsymbol{\theta}$ denotes the parameters to learn in our LGKDE model, and $\tilde{G}_i^{(j)}$ denotes the $j$th perturbed counterpart of $G_i$. More details about the motivation and rationale of (1) will be provided in the following sections.

### 3.3 LEARNING GRAPH DENSITY ESTIMATION

Based on the learned MMD metric space, we propose a learnable density estimation framework that combines structured graph perturbations with adaptive kernel density estimation. Our framework learns all parameters by maximizing the density of normal graphs relative to their perturbed counterparts, enabling effective capture of both structural and semantic patterns.

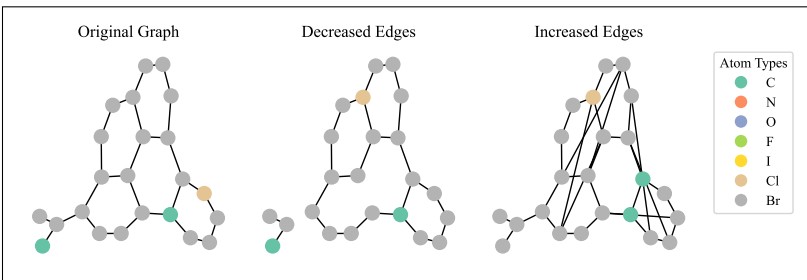

Figure 3: Demonstration of the node feature and energy-based spectral perturbation on a MUTAG molecule. Left: original graph; Middle: edge removal through high-energy group division; Right: edge addition through low-energy group multiplication. See more case studies and validation in Appendix D and B.4.

### 3.3.1 Structure-aware Sample Generation

According to (1), we need to devise a reliable strategy to perturb training graphs. Here, we introduce a structure-aware sample generation mechanism. The key idea is to create perturbed versions of normal graphs that preserve essential structural properties while introducing controlled variations. For each normal graph $G$, we generate perturbed samples through two types of perturbations:

**1) Node Feature Perturbation**: We randomly modify the features of a subset of nodes while preserving the graph structure:

$$\mathbf{X}'_v = \begin{cases} \mathbf{X}_{perm(v)} & \text{if } v \in \mathcal{V}_{swap} \\ \mathbf{X}_v & \text{otherwise} \end{cases} \tag{2}$$

where $perm(v)$ is a random permutation and $\mathcal{V}_{swap}$ contains $r_{swap}|V|$ randomly selected nodes.

**2) Energy-based Spectral Perturbation**: Given an adjacency matrix $\mathbf{A}$, we perform SVD decomposition $\mathbf{A} = \mathbf{U}\mathbf{\Sigma}\mathbf{V}^T$. Let $\{\sigma_i\}_{i=1}^n$ be the singular values in descending order. We define the cumulative energy ratio up to index $k$ as:

$$E(k) = \frac{\sum_{i=1}^k \sigma_i^2}{\sum_{i=1}^n \sigma_i^2} \tag{3}$$

Using energy thresholds $\tau_1 = 0.5$ and $\tau_2 = 0.75$, we partition the singular values into three groups:

$$\mathcal{S}_h = \{\sigma_i : E(i) \le \tau_1\} \quad \mathcal{S}_m = \{\sigma_i : \tau_1 < E(i) \le \tau_2\} \quad \mathcal{S}_l = \{\sigma_i : E(i) > \tau_2\} \tag{4}$$

So we can decide the fraction $p_{pert}$ of total singular values to be modified. We compute an adaptive ratio $r = \min(\mu_h/\mu_l, r_{\max})$, where $\mu_h$ and $\mu_l$ are the means of $\mathcal{S}_h$ and $\mathcal{S}_l$ respectively, and $r_{\max} = 10$. For edge addition/removal (operation $flag \in \{1, 0\}$), we modify the singular values as:

$$\tilde{\sigma}_i = \begin{cases} \sigma_i/r & \text{if } \sigma_i \in \mathcal{S}_h \text{ and } flag = 0 \quad \text{(edges removal)} \\ r\sigma_i & \text{if } \sigma_i \in \mathcal{S}_l \text{ and } flag = 1 \quad \text{(edges addition)} \\ \sigma_i & \text{otherwise} \end{cases} \tag{5}$$

The perturbed adjacency matrix is reconstructed as $\tilde{\mathbf{A}} = \mathbf{U}\tilde{\mathbf{\Sigma}}\mathbf{V}^T$ (with the fraction $p_{pert}$ of total singular values is modified). As shown in Figure 3, this generates structurally meaningful variations while preserving the graph's core topology. Algorithm 1 shows the pseudocode. Detailed analysis and study on the energy-based spectral perturbation are provided in the Appendix, including the case studies (Appendix D), detailed sensitivity ablation on $\tau_1, \tau_2$ (Appendix B.4), further validating its advantage in generating higher-quality contrastive counterparts compared with plain random structure permutation.

### 3.3.2 Parameterized Distance between Graphs

Inspired by (Sun & Fan, 2024), we use a deep graph MMD model to compute meaningful distances between graphs. The key idea is to represent each graph as a distribution over its node embeddings and measure graph similarity through MMD (Gretton et al., 2012).

Specifically, given a graph $G_i = (V_i, E_i, \mathbf{X}_i)$ with adjacency matrix $\mathbf{A}_i$ and node feature matrix $\mathbf{X}_i$, we learn node embeddings via a GNN with parameters $\boldsymbol{\theta}$:

$$\mathbf{Z}_i = \text{GNN}_{\boldsymbol{\theta}}(\mathbf{A}_i, \mathbf{X}_i) \tag{6}$$

where $\mathbf{Z}_i \in \mathbb{R}^{n_i \times d_{\text{out}}}$, $n_i = |V_i|$, and each row of $\mathbf{Z}_i$ is the $d_{\text{out}}$-dimension embedding of a node on $G_i$. Let $\mathbf{z}_p^{(i)}$ is the $p$-th row of $\mathbf{Z}_i$, the deep MMD distance between graphs (Sun & Fan, 2024) is

$$d_{MMD}(G_i, G_j) = \sup_{k_\gamma^{\text{emb}} \in \mathcal{K}_{\text{emb}}} \Big( \frac{1}{n_i^2} \sum_{p,q=1}^{n_i, n_i} k_\gamma^{\text{emb}}(\mathbf{z}_p^{(i)}, \mathbf{z}_q^{(i)}) + \frac{1}{n_j^2} \sum_{p,q=1}^{n_j, n_j} k_\gamma^{\text{emb}}(\mathbf{z}_p^{(j)}, \mathbf{z}_q^{(j)})$$
$$- \frac{2}{n_i n_j} \sum_{p=1}^{n_i} \sum_{q=1}^{n_j} k_\gamma^{\text{emb}}(\mathbf{z}_p^{(i)}, \mathbf{z}_q^{(j)}) \Big)^{1/2} \tag{7}$$

where $\mathcal{K}_{\text{emb}}$ is a family of characteristic kernels (e.g., Gaussian function) and each instance $k_\gamma^{\text{emb}}(\cdot, \cdot)$ uses a hyperparameter $\gamma \in \Gamma_{\text{emb}} := \{\gamma_1, \ldots, \gamma_S\}$. For a Gaussian family, $k_\gamma^{\text{emb}}(\mathbf{u}, \mathbf{v}) = \exp\left(-\gamma_s \|\mathbf{u} - \mathbf{v}\|^2\right)$. Intuitively, for each $\gamma$, we compare the distributions of node embeddings in $G_i$ and $G_j$. And the MMD distance takes the supremum over all kernels in $\mathcal{K}_{\text{emb}}$ to capture multi-scale structure.

This deep graph MMD model serves three crucial purposes: (1) learning structure-aware graph representations; (2) providing a theoretically grounded metric space for density estimation; and (3) enabling end-to-end learning through its fully differentiable architecture.

### 3.3.3 DENSITY LEARNING FRAMEWORK

We design an adaptive density estimation framework that jointly learns kernel parameters and density estimates through contrastive optimization. For a graph $G$ in our reference set $\mathcal{G} = \{G_1, \ldots, G_N\}$, we define its density using a multi-scale kernel density estimator:

$$\hat{f}(G) = \sum_{k=1}^{M} \pi_k(\boldsymbol{\alpha}) \phi_k(G), \quad \pi_k(\boldsymbol{\alpha}) = \frac{\exp(\alpha_k)}{\sum_{l=1}^{M} \exp(\alpha_l)} \tag{8}$$

where each component density $\phi_k(G)$ represents a kernel density estimate at bandwidth $h_k \in H_{KDE}$, and $\pi_k(\boldsymbol{\alpha})$ are learnable mixture weights parameterized by $\boldsymbol{\alpha}$ through a softmax function. Each component density is computed as:

$$\phi_k(G) = \frac{1}{N} \sum_{i=1}^{N} K_{KDE}(d_{MMD}(G, G_i), h_k), \quad h_k \in H_{KDE} \tag{9}$$

where $K_{KDE}(d, h) = \frac{1}{C_{d_{\text{int}}} h^{d_{\text{int}}}} K_0\left(\frac{d^2}{h^2}\right)$ is a kernel function with bandwidth $h$ and kernel profile $K_0$ and $d_{\text{int}}$ is the intrinsic dimension of the space induced by the input distance metric $d$. For a Gaussian kernel profile, $K_0(t) = e^{-t/2}$ and the normalization constant $C_{d_{\text{int}}} = (2\pi)^{d_{\text{int}}/2}$. Under the learned pairwise MMD distances between graphs, $d_{\text{int}} = 1$, making $C_{d_{\text{int}}} = \sqrt{2\pi}$.

The multi-scale design with bandwidths $H_{KDE} = \{h_k\}_{k=1}^{M}$ enables our model to capture patterns at different granularities, while the learnable weights automatically determine each scale's importance for the dataset. Building on our structure-aware sample generation mechanism, we train our model by maximizing the density ratio between normal graphs and their perturbed counterparts:

$$\min_{\boldsymbol{\theta}, \boldsymbol{\alpha}} \mathcal{L} := -\sum_{i=1}^{N} \sum_{j=1}^{N_{\text{pert}}} \frac{\hat{f}(G_i) - \hat{f}(\tilde{G}_i^{(j)})}{\hat{f}(G_i)} \tag{10}$$

where the objective encourages higher density values for normal graphs compared to their perturbed versions. The parameters $\boldsymbol{\theta}$ control the MMD metric learning while $\boldsymbol{\alpha}$ determines the kernel mixing weights. Note that $\hat{f}$ is a realization of $p_{\boldsymbol{\theta}}$ in (1). The rationale of (10) is that the density of a perturbed graph is often lower than that of the original graph and the model should be able to recognize this nature. Importantly, $\tilde{G}_i^{(j)}$ may not be anomalous and hence cannot be regarded as a negative sample and used in the manner of contrastive learning (You et al., 2020; Xu et al., 2021). Our objective naturally quantifies the perturbation effect via density instead of assigning the hard anomaly label.

During inference, we compute anomaly scores as $s(G) = -\hat{f}(G)$, with higher scores indicating higher likelihood of being anomalous. The detection threshold $\tau$ is estimated as the negative $\gamma_{TH}$-th percentile of reference set densities:

$$\tau = -\text{Percentile}_{\gamma_{TH}} \{\hat{f}(G_i) | G_i \in \mathcal{G}_{ref}\} \tag{11}$$

where $\gamma_{TH}$ controls the expected anomaly rate (empirically, we use $\gamma_{TH} = 0.1$). The complete algorithm is provided in Algorithm 3 with implementation details in Appendix E.

## 3.4 COMPUTATIONAL COMPLEXITY AND LARGE-SCALE EXTENSION

The time complexity of our LGKDE is detailed by Proposition E.1 in the Appendix E.4 and compared with the baselines. LGKDE process per graph's density estimation with $O(L(md + nd^2) + NSn^2d)$. This complexity is further reduced to $O(L(md + nd^2) + QSn^2d)$ by the technique introduced in Appendix E.4.3, where $Q \ll N$. As a result, LGKDE is scalable to large graph datasets.

## 4 THEORETICAL ANALYSIS

This section establishes comprehensive theoretical foundations for LGKDE, demonstrating its statistical consistency, convergence properties, robustness guarantees, and generalization. These theoretical results not only validate our algorithmic design choices but also provide insights into why the density-based loss with energy-based perturbations leads to stable and effective learning. Due to space constraints, we present the main results and key insights here, with all proofs, auxiliary lemmas, and extended discussions like generalization bound (Theorem 4.6) provided in the Appendices F.

**Theorem 4.1** (Consistency of LGKDE). *If the KDE bandwidths satisfy $h_k \to 0$ and $Nh_k^{d_{\text{int}}} \to \infty$ as $N \to \infty$ for all $k = 1, \ldots, M$, then $\hat{f} \xrightarrow{p} f^*$ in $L_1$ norm.*

This result (Proof in Appendix F.2) establishes that LGKDE converges to the true underlying graph density function $f^*$ as the number of samples increases. We further quantify the rate of convergence:

**Theorem 4.2** (Convergence Rate). *Under the conditions of Theorem 4.1, the Mean Integrated Squared Error (MISE) of LGKDE with optimal bandwidths $h^* \sim N^{-1/(4+d_{\text{int}})}$ achieves:*

$$\mathbb{E} \int_{\mathbb{G}} (\hat{f}^*(G) - f^*(G))^2 d\mathbb{P}^*(G) = O(N^{-\frac{4}{4+d_{\text{int}}}}) \tag{12}$$

This rate matches the minimax optimal rate for nonparametric density estimation in $d_{\text{int}}$ dimensions, when utilizing the pairwise MMD distance for KDE, *MISE* bounded to $O(N^{-0.8})$ for $d_{\text{int}} = 1$, demonstrating the statistical efficiency. Detailed proof and discussion in Appendix F.3.

Robustness ensures that LGKDE's density estimate $\hat{f}$ is stable against small changes in input graphs. Furthermore, our proposed novel energy-based perturbation strategy and the density learning target are theoretically justified by the following robustness analysis. Detailed in Appendix F.4, we introduce our three main results as follows.

**Theorem 4.3** (Robustness of KDE to Metric Perturbations). *Suppose $K_0$ is the Gaussian kernel and let $d_{12}(G_1, G_2) = d_{MMD}|_{\boldsymbol{\theta}}(G_1, G_2)$ and $c_h = \min_k h_k$. For any two graphs $G_1, G_2 \in \mathbb{G}$, it holds that*

$$|\hat{f}(G_1) - \hat{f}(G_2)| \leq \frac{1}{\sqrt{2e\pi}c_h^2} d_{12}(G_1, G_2) \tag{13}$$

Theorem 4.3 shows that when two graphs are close in terms of the learned MMD metric, their densities are similar, provided that the least bandwidth of the Gaussian kernel is not too small. Therefore, in the experiments, we don't use a very small $h_k$ and take a set of five logarithmically spaced bandwidths $H_{KDE} = \{h_k\}_{k=1}^M = \{10^{-2}, 10^{-1}, 10^0, 10^1, 10^2\}$.

**Theorem 4.4** (Robustness of MMD to Graph Perturbations). *Suppose $G = (\mathbf{A}, \mathbf{X})$ is perturbed to $\tilde{G} = (\tilde{\mathbf{A}}, \tilde{\mathbf{X}})$, where $\tilde{\mathbf{X}} - \mathbf{X} = \Delta_{\mathbf{X}}$ and $\tilde{\mathbf{A}} - \mathbf{A} = \Delta_{\mathbf{A}}$. Let $\kappa = \min(1^\top \Delta_{\mathbf{A}})$, $\alpha$ be the maximum node degree, and $n$ be the number of nodes. Suppose the GNN has $L$ layers, the weight matrix of each layer is $W_l$, $l \in [L]$, and all activation functions are 1-Lipschitz continuous. Then*

$$d_{MMD}|_{\boldsymbol{\theta}}(G, \tilde{G}) \leq \left(4\bar{\gamma}^2 + 2\epsilon^4\bar{\gamma}^4\right)\left(2\Delta_G^4 + n\Delta_G^2 + \frac{\epsilon}{\sqrt{n}}\Delta_G\right) \tag{14}$$

*where $\epsilon = 2(1 + \alpha)^{-L}(1 + \|\mathbf{A}\|_2)^L\|\mathbf{X}\|_2$, $\bar{\gamma} = \max_{\gamma \in \Gamma_{emb}} \gamma$, and*

$$\Delta_G = \left(\frac{1}{1+\alpha+\kappa}\right)^L \prod_{l=1}^L \|\mathbf{W}_l\|_2 \left((\|\mathbf{A}\|_2 + \|\Delta_{\mathbf{A}}\|_2)^L\|\Delta_{\mathbf{X}}\|_F + \|\mathbf{X}\|_F \sum_{l=0}^{L-1} \|\mathbf{A}\|_2^l \left(\sqrt{(\|\mathbf{A}\|_2 + \|\Delta_{\mathbf{A}}\|_2)^2 - \|\mathbf{A}\|_2^2}\right)^{L-l}\right).$$

Theorem 4.4 quantifies the stability of the learned metric $d_{MMD}$ itself under graph perturbations. It is more robust if $\|\mathbf{W}_l\|_2$, $\|\mathbf{A}_l\|_2$, and $L$ are smaller. Combining Theorem 4.4 and Theorem 4.3, we have the following corollary.

**Corollary 4.5.** *With the same conditions in Theorem 4.3 and Theorem 4.4, it holds that*

$$|\hat{f}(G) - \hat{f}(\tilde{G})| \leq \frac{1}{\sqrt{2e\pi}c_h^2} \left(4\bar{\gamma}^2 + 2\epsilon^4\bar{\gamma}^4\right)\left(2\Delta_G^4 + n\Delta_G^2 + \frac{\epsilon}{\sqrt{n}}\Delta_G\right) \tag{15}$$

According to the corollary, suppose both $G$ and $\tilde{G}$ are normal graphs; the difference between their densities will be smaller. This guarantees a low false positive rate in anomalous graph detection. It can also be used to understand the role of the proposed structure-aware sample generation strategy. According to Eq (5), the graph $\tilde{G}$ given by our strategy satisfies $\|\Delta_{\mathbf{A}}\|_2 = \max_i \max(|\sigma_i/r - \sigma_i|, |\sigma_i - r\sigma_i|)$. Therefore, by controlling $r$, the generated graph $\tilde{G}$ will have a similar density as the original graph $G$. It means that solving (10) can ensure a strong discrimination ability for our model, which is crucial for anomaly detection.

**Theorem 4.6** (Generalization Bound). *Let $\alpha = \sqrt{\sum_{i=1}^N \|\mathbf{X}_i\|_F^2}\|\mathbf{W}_1\|_{2,1}\prod_{l=2}^L \|\mathbf{W}_l\|_2$, and $\hat{\Delta}_{\mathcal{G}} = \frac{1}{N}\sum_{i=1}^N |\hat{f}(G_i) - f^*(G_i)|$. Suppose all activation functions are 1-Lipschitz continuous. Over the randomness of the sample $\mathcal{G}$, the following inequality holds with probability at least $1 - \delta$:*

$$\mathbb{E}[|\hat{f}(G) - f^*(G)|] \leq \hat{\Delta}_{\mathcal{G}} + \frac{8\sqrt{en\pi}c_h^2 + 24\bar{\gamma}\alpha\sqrt{\ln(2d^2)}}{N\sqrt{en\pi}c_h^2}\ln(N) + \sqrt{\frac{\log(1/\delta)}{2N}} \tag{16}$$

Theorem 4.6 (Proof in Appendix F.5) reveals the impact of network architecture, weight matrices, and data size and complexity on the generalization ability of our model. It implies that our density estimator can generalize well to unseen graphs when $N$ is large and the norms of the weight matrices are small. In addition, the graph node number $n$ has little impact on the bound since $\alpha$ is linear with $\sqrt{n}$. This means the ability of our method is not influenced by $n$. Indeed, in the experiments, our method outperformed the baselines on graphs of disparate sizes (e.g, PROTEINS, REDDIT-B).

## 5 EXPERIMENTS

We conduct comprehensive experiments to rigorously evaluate the proposed LGKDE framework. We begin by directly assessing LGKDE's capability in recovering underlying graph distributions on synthetic data where the ground truth is known. Subsequently, acknowledging the intricate nature of real-world graph distributions (e.g., in molecular structures and social networks), which poses significant challenges for direct density modeling, we leverage graph anomaly detection as a primary application to validate LGKDE's effectiveness against state-of-the-art methods.

### 5.1 VALIDATING DENSITY ESTIMATION CAPABILITIES ON SYNTHETIC DATA

To establish LGKDE's fundamental ability to perform density estimation, we first conducted targeted experiments on synthetic Erdős–Rényi (ER) graphs, then extended our evaluation to diverse graph types like Barabási-Albert and Watts-Strogatz (See Appendix B.5.2). In these controlled settings, the true generative parameters of the graphs were known.

We generated ER graphs where node counts $n$ were sampled uniformly from $[20, 50]$ and edge probabilities $p$ were drawn from a Beta(2,2) distribution, which is unimodal and peaked at $p = 0.5$. LGKDE was tasked with estimating the density of these graphs.

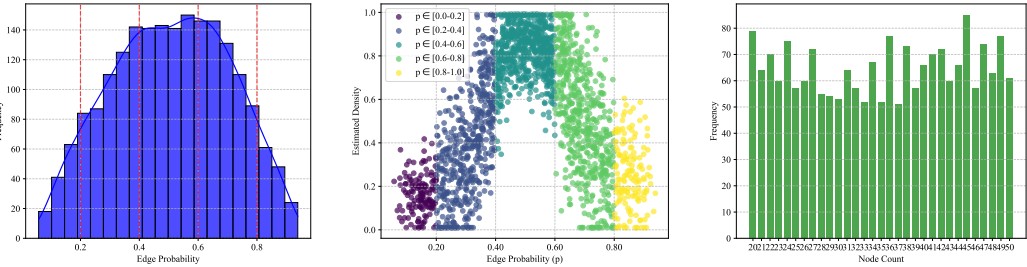

Figure 4: Density estimation results on synthetic Erdős–Rényi graphs. Left: Ground truth Beta(2, 2) distribution for edge probability $p$. Middle: Learned density estimate from LGKDE versus edge probability $p$, showing the expected peak around $p = 0.5$. Right: Distribution of node counts (uniform as generated).

As Fig. 4 shows, LGKDE successfully recovered the underlying distribution of edge probabilities, assigning significantly higher densities to graphs with $p$ around 0.5 (with Avg. $p \in [0.4 - 0.6] = 0.82$, details in Table 10 of Appendix B.5.1), irrespective of variations in node count.

Due to space limitations, additional experiments on Barabási-Albert, Watts-Strogatz, and Stochastic Block Models graphs are provided in Appendix B.5.2. These foundational validations provide strong evidence that LGKDE can capture the underlying density on synthetic graph distributions and support its application to the following more complex, real-world graph data anomaly detection tasks.

## 5.2 Graph Anomaly Detection on Real-World Benchmarks

### 5.2.1 Experimental Settings

**Datasets:** We use twelve widely adopted graph benchmark datasets from the TU Database (Morris et al., 2020). Following the class-based anomaly detection setup from established practices like (Liu et al., 2023a; Qiao et al., 2024; Wang et al., 2024), minority classes are treated as anomalous, and the majority as normal. Detailed statistics are in Appendix 3.

**Baselines:** LGKDE is compared against **traditional methods** (**Graph Kernel + Detector**, e.g., WL (Shervashidze et al., 2011) with iForest (Liu et al., 2008)) and various **GNN-based deep learning approaches**: OCGIN (Zhao & Akoglu, 2021), GLocalKD (Ma et al., 2022), OCGTL (Qiu et al., 2022), SIGNET (Liu et al., 2023b), GLADC (Luo et al., 2022), CVTGAD (Li et al., 2023), MUSE (Kim et al., 2024) and UniFORM (Song et al., 2025). Details of baselines are in Appendix G.4.

**Implementation Details:** LGKDE is implemented in PyTorch, using a GCN backbone and the Adam optimizer with a learning rate of 0.001. For multi-scale KDE, we use $M = 5$ bandwidths. For fair comparison, all GNN-based baselines use a similar GNN architecture where applicable, adhering to settings from unified benchmarks (Wang et al., 2024). Further details on LGKDE's specific hyperparameter settings and those for baselines are in Appendix G.3 and Appendix G.4. All results are averaged over five trials.

**Evaluation Metrics:** Performance is assessed using Area Under the ROC Curve (AUROC), Area Under the Precision-Recall Curve (AUPRC), and False Positive Rate at 95% True Positive Rate (FPR95). Higher(↑) AUROC/AUPRC and lower(↓) FPR95 indicate better performance. Metric definitions are in Appendix G.2.

### 5.2.2 Main Results and Analysis

Table 1 presents the AUROC. AUPRC and FPR95 results are in Appendix B.3 (Tables 4 and 5).

**Overall Performance.** LGKDE consistently demonstrates superior performance, achieving the highest average AUROC (79.37%) and AUPRC (84.01%) across 12 datasets. These results outperform strong baselines like MUSE and SIGNET, particularly on datasets such as DHFR (AUROC: 82.58% vs. SIGNET's 72.87% and MUSE's 71.45%). LGKDE also obtains the best average FPR95 (59.92%, Table 5). This underscores LGKDE's efficacy in modeling complex normal graph distributions to detect anomalies, naturally aligning our theoretical analysis on consistency (Theorem 4.1) and convergence (Theorem 4.2) in Section 4.

**Method-specific Analysis.** Our extensive results reveal that traditional graph kernel methods generally underperform, reflecting the limitations of fixed, handcrafted kernels. Deep learning methods show improvements, though approaches like OCGIN and CVTGAD are brittle under long-tailed priors and weak cluster separability in complex benchmarks.

LGKDE's end-to-end learning of both the metric space and density estimator serves as a principled view to solve GAD tasks, which also offers a distinct advantage over two-stage methods (with extra comparison on InfoGraph (Sun et al., 2019a) and GAE (Kipf & Welling, 2016) followed by a separate KDE), as evidenced by its significantly better performance on MUTAG (e.g., 91.63% AUROC vs. 81.94% for GAE+KDE; details in Table 12). This efficacy stems from our principled design, where the joint optimization target (Eq. (10)) pairs novel energy-based perturbations with a density-based loss to learn a task-attuned metric space. LGKDE's stability is theoretically grounded by our robustness analysis (Corollary 4.5), which underpins the consistent high performance across diverse datasets.

Table 1: Comparison in terms of AUROC(%↑). "Avg. AUROC" and "Avg. Rank" are computed across all datasets. Top-3 performance are indicated by light gold / light silver / light bronze cell shading with superscripts ①/②/③.

| Method | MUTAG | PROTEINS | DD | ENZYMES | DHFR | BZR | COX2 | AIDS | IMDB-B | NCI1 | COLLAB | REDDIT-B | Avg. AUROC | Avg. Rank |
|---|---|---|---|---|---|---|---|---|---|---|---|---|---|---|
| *Graph Kernel + Detector* | | | | | | | | | | | | | | |
| PK-SVM | $46.06_{\pm0.47}$ | $49.43_{\pm0.69}$ | $47.69_{\pm0.24}$ | $52.45_{\pm0.29}$ | $48.31_{\pm0.47}$ | $46.67_{\pm0.52}$ | $52.15_{\pm0.16}$ | $50.93_{\pm0.19}$ | $51.75_{\pm0.30}$ | $51.39_{\pm0.19}$ | $49.72_{\pm0.60}$ | $48.36_{\pm0.67}$ | 49.58 | 12.42 |
| PK-iF | $47.98_{\pm0.41}$ | $61.24_{\pm0.34}$ | $75.29_{\pm0.46}$ | $49.82_{\pm0.67}$ | $52.79_{\pm0.35}$ | $59.08_{\pm0.29}$ | $52.48_{\pm0.38}$ | $52.01_{\pm0.53}$ | $52.83_{\pm0.51}$ | $50.22_{\pm0.12}$ | $51.38_{\pm0.20}$ | $46.19_{\pm0.21}$ | 54.28 | 11.25 |
| WL-SVM | $62.18_{\pm0.29}$ | $53.85_{\pm0.26}$ | $47.98_{\pm0.32}$ | $53.75_{\pm0.34}$ | $50.30_{\pm0.31}$ | $51.16_{\pm0.36}$ | $53.34_{\pm0.27}$ | $52.56_{\pm0.41}$ | $52.98_{\pm0.69}$ | $54.18_{\pm0.67}$ | $54.62_{\pm1.28}$ | $49.50_{\pm0.54}$ | 53.03 | 10.67 |
| WL-iF | $65.71_{\pm0.38}$ | $65.75_{\pm0.35}$ | $70.49_{\pm0.28}$ | $51.03_{\pm0.42}$ | $51.64_{\pm0.22}$ | $51.71_{\pm0.45}$ | $49.56_{\pm0.11}$ | $61.42_{\pm0.50}$ | $51.79_{\pm0.32}$ | $50.41_{\pm0.31}$ | $51.41_{\pm0.39}$ | $49.84_{\pm0.11}$ | 55.90 | 11.00 |
| *GNN-based Deep Learning Methods* | | | | | | | | | | | | | | |
| OCGIN | $79.55_{\pm0.22}$ | $76.46_{\pm0.13}$ | $79.08_{\pm0.19}$ | $62.44_{\pm0.38}$ | $61.09_{\pm0.27}$ | $69.13_{\pm0.13}$ | $57.81_{\pm0.50}$ | $96.89_{\pm0.20}$ | $61.47_{\pm0.18}$ | $69.46_{\pm0.36}$ | $60.58_{\pm0.27}$ | $82.10_{\pm0.37}$ | 71.34 | 7.25 |
| GLocalKD | $86.25_{\pm0.57}$ | $77.29_{\pm0.41}$② | $80.76_{\pm0.50}$① | $61.75_{\pm0.10}$ | $61.79_{\pm0.54}$ | $68.55_{\pm0.15}$ | $58.93_{\pm0.47}$ | $96.93_{\pm0.34}$ | $53.31_{\pm0.53}$ | $65.29_{\pm0.21}$ | $51.85_{\pm0.18}$ | $80.32_{\pm0.10}$ | 70.25 | 6.92 |
| OCGTL | $88.02_{\pm0.43}$ | $72.89_{\pm0.57}$ | $77.76_{\pm0.48}$ | $63.59_{\pm0.11}$ | $59.82_{\pm0.44}$ | $51.89_{\pm0.46}$ | $59.81_{\pm0.30}$ | $99.36_{\pm0.67}$① | $65.27_{\pm0.24}$ | $75.75_{\pm0.47}$② | $48.13_{\pm0.41}$ | $88.03_{\pm0.22}$② | 70.86 | 6.33 |
| SIGNET | $88.84_{\pm0.15}$② | $75.86_{\pm0.30}$ | $74.53_{\pm0.11}$ | $63.12_{\pm0.52}$ | $72.87_{\pm0.28}$② | $80.79_{\pm0.38}$② | $72.35_{\pm0.58}$① | $97.60_{\pm0.28}$ | $70.12_{\pm0.61}$① | $74.32_{\pm0.34}$ | $72.45_{\pm0.11}$① | $85.24_{\pm0.45}$ | 77.34② | 4.17① |
| GLADC | $83.07_{\pm0.29}$ | $77.43_{\pm0.19}$② | $76.54_{\pm0.25}$ | $63.44_{\pm0.30}$ | $61.25_{\pm0.19}$ | $68.23_{\pm0.31}$ | $64.13_{\pm0.23}$ | $98.02_{\pm0.23}$ | $65.94_{\pm0.26}$ | $68.32_{\pm0.22}$ | $54.32_{\pm0.37}$ | $78.87_{\pm0.56}$ | 71.63 | 6.83 |
| CVTGAD | $86.64_{\pm0.32}$ | $76.49_{\pm0.29}$ | $78.84_{\pm0.40}$ | $68.56_{\pm0.43}$③ | $63.23_{\pm0.38}$ | $77.69_{\pm0.28}$ | $64.36_{\pm0.16}$ | $99.21_{\pm0.27}$② | $69.82_{\pm0.13}$② | $69.13_{\pm0.26}$ | $71.01_{\pm0.58}$③ | $87.43_{\pm0.60}$③ | 76.03 | 4.17③ |
| MUSE | $85.92_{\pm0.28}$ | $76.87_{\pm0.32}$ | $79.23_{\pm0.35}$③ | $67.82_{\pm0.38}$ | $71.45_{\pm0.33}$③ | $78.34_{\pm0.30}$ | $65.87_{\pm0.29}$③ | $98.95_{\pm0.31}$ | $67.84_{\pm0.27}$ | $74.45_{\pm0.26}$③ | $67.48_{\pm0.36}$ | $85.32_{\pm0.33}$ | 76.63② | 4.08② |
| UniFORM | $88.45_{\pm0.24}$③ | $77.15_{\pm0.27}$ | $78.56_{\pm0.31}$ | $69.34_{\pm0.41}$② | $69.78_{\pm0.36}$ | $79.85_{\pm0.32}$③ | $65.12_{\pm0.31}$ | $98.52_{\pm0.22}$ | $68.42_{\pm0.29}$ | $72.93_{\pm0.31}$ | $66.23_{\pm0.38}$ | $84.67_{\pm0.35}$ | 76.58 | 4.25 |
| **LGKDE** | $91.63_{\pm0.31}$① | $78.97_{\pm0.26}$① | $79.84_{\pm0.41}$② | $71.04_{\pm0.45}$① | $82.58_{\pm0.33}$① | $81.11_{\pm0.32}$① | $66.69_{\pm0.39}$② | $99.06_{\pm0.25}$③ | $68.77_{\pm0.29}$③ | $76.67_{\pm0.30}$① | $67.94_{\pm0.41}$③ | $88.12_{\pm0.39}$① | 79.37① | 1.67① |

**Dataset-specific Insights and Learned Representations.** LGKDE excels on molecular datasets (e.g., MUTAG, PROTEINS), indicating its proficiency in capturing intricate chemical structures. This is supported by t-SNE visualizations (Figure 5) on MUTAG, which show LGKDE learning a more discriminative embedding space with clearer class separation compared to other methods. Qualitative analysis of graphs from MUTAG corresponding to "average", highest, and lowest learned densities (Appendix B.5.5, Figure 9) further reveals that LGKDE captures structurally meaningful patterns: high-density graphs often exhibit complex ring structures typical of the normal class, while low-density graphs display simpler or atypical formations. On social network datasets (e.g., IMDB-B and COLLAB), LGKDE remains competitive and performs best on REDDIT-B despite its substantial disparity in graph size $n$ (average $429.63$ nodes with standard deviation $554.06$), demonstrating its adaptability and aligning with our analysis on generalization bound (Theorem 4.6).

### 5.3 ABLATION STUDIES

Comprehensive ablation studies (detailed in Appendix B.4 due to space limit) validate the key components and design choices of LGKDE. Our key results in Table 6 demonstrate that: (1) The multi-scale KDE design is crucial, outperforming single-bandwidth variants. (2) The learned MMD distance is superior to simpler graph readout functions for capturing relevant graph similarities. (3) Learnable bandwidth mixture weights adaptively improve performance over fixed weights. (4) Figure 6 (upper) shows that a moderate GNN depth (2-3 layers) is optimal, avoiding over-smoothing.

We further systematically compare and analyze our core structure-aware sample generation strategy in Appendix B.4, Table 7 reveals that our proposed energy-based spectral perturbations contribute more significantly than common random perturbations by providing well-designed structure perturbed counterparts for density-based contrastive learning, and their combination achieves synergistic gains. Sensitivity analysis results on the energy-based thresholds $(\tau_1, \tau_2)$ are given in Table 8, confirming our design choice and aligning with our analysis in Theorem 4.4. Extra parameter-controlled experiments (Table 9) demonstrate that LGKDE with a moderate GNN capacity significantly outperforms unlearnable or single-scale KDE with massive GNN backbone parameters, validating that the gains stem from adaptive density estimation. These findings confirm the efficacy of our design choices.

The robustness of LGKDE, as demonstrated by its resilience to training data contamination in Figure 6 (bottom) and the extra stable density gap analysis in Appendix B.5.4. We also conducted controlled distribution shift experiments in Appendix B.6, showing LGKDE maintains superior performance when test distributions deviate from training. Specifically, these results align with Corollary 4.5 and Theorem 4.6 for robustness on unseen graphs, providing strong empirical validation for our theoretical framework.

## 6 CONCLUSION

We proposed LGKDE, a novel framework for learnable kernel density estimation on graphs, which jointly optimizes graph representations and multi-scale kernel parameters via maximizing the density of normal graphs relative to their well-designed perturbed counterparts. It also naturally provides a principled density estimation perspective for graph-level anomaly detection. Theoretical analysis and extensive experiments validate LGKDE's effectiveness and superior performance in graph density estimation and anomaly detection.

## ETHICS STATEMENT

This work presents LGKDE, a technical contribution to graph kernel density estimation. Our research is purely methodological and does not involve human subjects, sensitive personal data, or any ethically concerning applications. The proposed method focuses on improving the theoretical foundations and practical performance of graph density estimation and graph-level anomaly detection algorithms. We have carefully considered the potential applications of our work and found no foreseeable negative societal impacts. The datasets used in our experiments are publicly available benchmark datasets commonly used in the graph learning community, containing no personal or sensitive information. Our work does not touch upon political, religious, racial, or cultural topics, nor does it introduce any discriminatory biases. The research adheres to standard academic integrity practices and does not raise any ethical concerns related to the ICLR Code of Ethics.

## REPRODUCIBILITY STATEMENT

To ensure the reproducibility of our work, we have made significant efforts to provide comprehensive implementation details and will release all necessary resources after acceptance. All datasets used in our experiments are publicly available benchmark datasets from established graph learning repositories. Our implementation is based on widely-used open-source frameworks, including PyTorch Geometric and DGL. The complete source code, including the main LGKDE implementation, data preprocessing scripts, and evaluation protocols, will be made publicly available on GitHub upon paper acceptance. The anonymous supplementary materials already contain the core implementation for review purposes.

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

# A  SYMBOL AND NOTATION

This appendix provides a comprehensive reference (Table 2) for the principal notation used throughout the paper. While we strive to maintain consistent notation, specialized symbols that appear exclusively in proofs or specific derivations are defined in their respective contexts. We adhere to the following conventions:

- bold uppercase letters (e.g., $\mathbf{A}$, $\mathbf{X}$) denote matrices; bold lowercase letters (e.g., $\mathbf{z}$) represent vectors;
- calligraphic letters (e.g., $\mathcal{G}$, $\mathcal{H}$) indicate sets or spaces; uppercase letters (e.g., $G$, $N$) typically represent graphs, counts, or constants; lowercase letters (e.g., $h$, $d$) typically represent scalar values or functions; and Greek letters (e.g., $\gamma$, $\pi$) denote parameters or functional elements.
- Superscripts and subscripts are used to index specific instances (e.g., $G_i$ for the $i$-th graph) or to indicate special properties or variants of a symbol.

Table 2: Summary of Notation and Symbols

| Symbol(s) | Description |
|---|---|
| $\mathbb{G}$ | Space of attributed graphs. |
| $G_i = (V_i, E_i, \mathbf{X}_i)$ | Graph $i$ with nodes $V_i$, edges $E_i$, and node features $\mathbf{X}_i$. |
| $\mathbf{A}_i, \mathbf{X}_i, n_i$ | Adjacency matrix, feature matrix ($d_{in}$-dim), and node count for $G_i$. |
| $\tilde{G}$ | Perturbed version of a graph $G$. |
| $\hat{\mathbf{A}}$ | Normalized adjacency matrix: $\mathbf{D}^{-1/2}(\mathbf{A} + \mathbf{I})\mathbf{D}^{-1/2}$. |
| $\mathcal{S}_h, \mathcal{S}_m, \mathcal{S}_l; E(k); \tau_1, \tau_2$ | Singular value sets, cumulative energy, and thresholds for spectral perturbation. |
| $r_{swap}, p_{pert}, \textit{flag}$ | Ratio of nodes for feature swapping; the fraction of singular values be modified; edge operation $\textit{flag} \in \{0, 1\}$ (0: removal, 1: addition) |
| $\mathbb{P}^*, f^*$ | True underlying probability measure and density function of graphs. |
| $\hat{f}$ | Estimated KDE density function for graphs. |
| $\mu, P_{\mathbf{Z}}$ | Base measure on $\mathbb{G}$; Empirical distribution of node embeddings. |
| $\boldsymbol{\theta}, \Phi_{\boldsymbol{\theta}}$ | GNN parameters ($\{\mathbf{W}^{(l)}\}_{l=1}^L$) and the GNN encoder. |
| $\mathbf{Z}_i, \mathbf{z}_p^{(i)}$ | Node embedding matrix for $G_i$ ($\in \mathbb{R}^{n_i \times d_{out}}$), and embedding of $p$-th node in $G_i$. |
| $d_{hid}$ and $d_{out}, \mathbf{W}^{(l)}, L$ | GNN hidden and output dimension, $l$-th layer weights, number of GNN layers. |
| $d_{\text{MMD}}$ | MMD distance between graphs. |
| $\mathcal{K}_{\text{emb}}, k_{\text{emb}}, \gamma, \Gamma_{\text{emb}}$ | Family of MMD kernels ($k_{\text{emb}}$, typically Gaussian) acting on node embeddings, its bandwidth $\gamma$, and set of $S$ hyperparameters $\Gamma_{\text{emb}} = \{\gamma_s\}_{s=1}^S$. |
| $\mathcal{H}, \mu_{\delta_{\mathbf{z}}}, \mu_{P_{\mathbf{Z}}}, \mathcal{M}$ | RKHS for $k_{\text{emb}}$; Kernel mean embedding of a Dirac delta and of distribution $P_{\mathbf{Z}}$; Riemannian manifold induced by the MMD metric. |
| $K_{KDE}, K_0, h, H_{KDE}$ | KDE kernel function (profile $K_0$, typically Gaussian) operating on $d_{\text{MMD}}$, its bandwidth $h$, and set of $M$ bandwidths $H_{KDE} = \{h_j\}_{j=1}^M$. |
| $\boldsymbol{\alpha}, \mathcal{A}, \pi_j, \phi_j$ | KDE mixture weight parameters ($\{\alpha_j\}$ in space $\mathcal{A}$), $j$-th mixture weight $\pi_j(\boldsymbol{\alpha})$, and $j$-th KDE component density $\phi_j$. |
| $d_{int}, C_{d_{int}}$ | Intrinsic dimension (set to 1 for KDE on pairwise distances), KDE normalization constant. |
| $\mathcal{L}, \gamma_{TH}$ | Training loss function; Anomaly rate threshold parameter. |
| $L_\mu(k), L_\mu^*$ | Lipschitz constant of kernel mean map for $k \in \mathcal{K}_{\text{emb}}$, and its supremum. |
| $L_K(h), L_{KDE}, C_{KDE}$ | Lipschitz constant of $K_{KDE}(d, h)$ w.r.t $d$; Overall Lipschitz and 2nd-order coefficient for $\hat{f}_{\text{KDE}}$. |
| $\beta, \epsilon, \Delta_{perturb}$ | Smoothness of $f^*$; Accuracy in sample complexity; MMD bound from perturbation. |
| $B_{\Delta X}, B_{\Delta A}, B_{\Delta \hat{A}}, B_W, B_X$ | Norm bounds for perturbations and GNN/feature matrices. |
| $\|\cdot\|_F$ | Frobenius norm for matrices, defined as $\|\mathbf{A}\|_F = \sqrt{\sum_{i,j} |a_{ij}|^2}$. |
| $\|\cdot\|_2$ | Euclidean norm for vectors; for matrices, denotes the spectral norm (largest singular value). |
| $\|\cdot\|_\infty$ | $L_\infty$ norm: for functions, $\|f\|_\infty = \sup_x |f(x)|$; for vectors, $\|\mathbf{x}\|_\infty = \max_i |x_i|$. |
| $\|\cdot\|_{\mathcal{H}}$ | RKHS norm in the Hilbert space $\mathcal{H}$ induced by kernel $k$. |

# B  MORE EXPERIMENT RESULTS

## B.1  STATISTICS OF DATASETS

See Table 3.

Table 3: Statistics of the benchmark datasets.

| Dataset | #Graphs | Avg. Nodes | Avg. Edges | #Classes | Anomaly Ratio |
|---|---|---|---|---|---|
| MUTAG | 188 | 17.93 | 19.79 | 2 | 33.5% |
| PROTEINS | 1113 | 39.06 | 72.82 | 2 | 40.5% |
| DD | 1178 | 284.32 | 715.66 | 2 | 41.7% |
| ENZYMES | 600 | 32.63 | 62.14 | 6 | 16.7% |
| DHFR | 756 | 42.43 | 44.54 | 2 | 39.0% |
| BZR | 405 | 35.75 | 38.36 | 2 | 35.3% |
| COX2 | 467 | 41.22 | 43.45 | 2 | 37.8% |
| AIDS | 2000 | 15.69 | 16.20 | 2 | 31.2% |
| IMDB-BINARY | 1000 | 19.77 | 96.53 | 2 | 50.0% |
| NCI1 | 4110 | 29.87 | 32.30 | 2 | 35.1% |
| COLLAB | 5000 | 74.49 | 2457.78 | 3 | 21.6% |
| REDDIT-BINARY | 2000 | 429.63 | 497.75 | 2 | 50.0% |

*IMDB-BINARY and REDDIT-BINARY are henceforth abbreviated as IMDB-B and REDDIT-B, respectively.*

## B.2  VISUALIZATION AND COMPARISON ON MUTAG DATASETS

See Figure 5 for t-SNE visualizations comparing different representative methods of our benchmarks.
1) **Separation Quality**: LGKDE achieves the clearest separation between normal (Class 0) and anomalous (Class 1) graphs, displaying a concentric circular structure with normal samples forming a well-defined core surrounded by anomalous samples. This distinctive pattern aligns with its superior performance (AUROC: 91.63%, AUPRC: 96.75%).
2) **Deep Learning Methods**: - SIGNET shows partial separation but with more scattered distribution (AUROC: 88.84%, second best). - OCGTL and UniFORM exhibit some clustering but with significant overlap between classes (AUROC: 88.02% and 88.45%). - OCGIN shows the least structured distribution among deep methods (AUROC: 79.55%).
3) **Traditional Kernels**: - WL and PK kernels both show substantial overlap between classes, explaining their relatively poor performance (WL-SVM AUROC: 62.18%, PK-SVM AUROC: 46.06%). - Their t-SNE visualizations lack clear structural patterns, suggesting limited ability to capture meaningful graph similarities.
4) **Distribution Structure**: LGKDE's concentric arrangement suggests it successfully learns a meaningful metric space where normal graphs are tightly clustered while anomalous graphs are naturally pushed toward the periphery. This structure reflects the principle of density estimation where normal patterns should be concentrated in high-density regions. The visualization results strongly correlate with quantitative metrics across all methods, with clearer visual separation corresponding to better performance in AUROC, AUPRC, and FPR95. This demonstrates the effectiveness of our learned kernel approach.

## B.3  COMPARISON IN TERMS OF AUPRC AND FPR95 RESULTS

See Table 4 and Table 5.

Our proposed LGKDE demonstrates compelling performance advantages in both precision-recall capabilities and false alarm control. For AUPRC, LGKDE achieves the highest average score of 84.01%, substantially outperforming the second-best method CVTGAD (73.77%). The performance improvement is particularly pronounced on datasets like ENZYMES (52.79% vs. next best 43.65%), MUTAG (96.75% vs. 82.67%), and NCI1 (92.22% vs. 67.45%). On datasets where LGKDE is not the top performer, it still maintains competitive performance with minimal gaps (e.g., BZR: 88.79% vs. best 91.98%; COLLAB: 68.51% vs. best 70.59%). In terms of controlling false positives at high recall rates, LGKDE achieves the lowest average FPR95 of 59.92%, outperforming runner-up approaches CVTGAD (65.11%) and MUSE (66.90%). This represents a significant reduction in false alarms while maintaining high detection rates. LGKDE secures the best FPR95 with notable improvements on DHFR (68.95% vs. next best 73.38%).

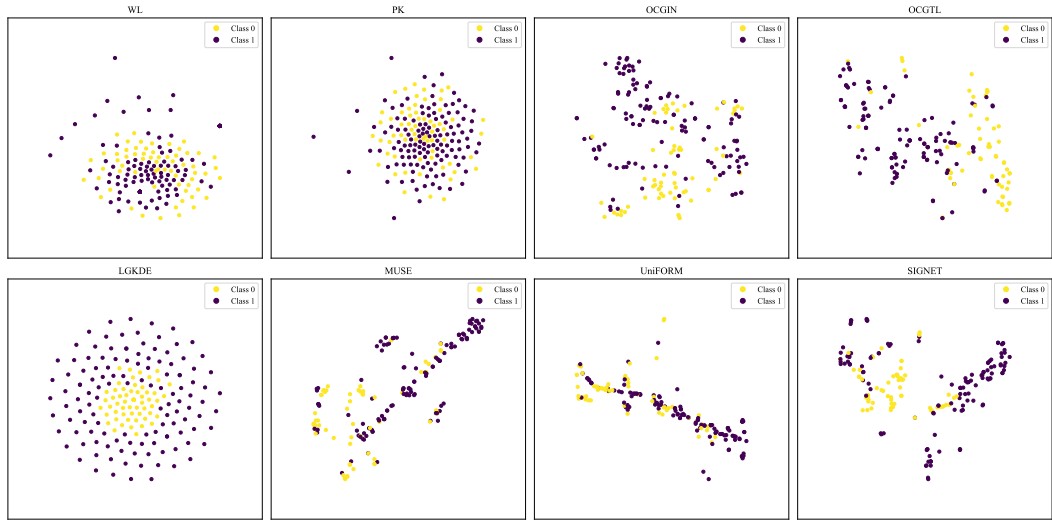

Figure 5: t-SNE visualization (perplexity=30) of learned kernel matrix on the MUTAG dataset. For LGKDE, WL, and PK kernels, we use the learned kernel matrix to perform SVD to get $\mathbf{U\Sigma V^T}$ and use the $\mathbf{\Sigma V^T}$ as the t-SNE inputs. For MUSE, UniFORM, SIGNET, OCGTL and OCGIN, we use their learned graph-level readout embeddings as the t-SNE inputs.

The consistent superior performance across both metrics is further validated by LGKDE achieving the best average ranks of 1.75 for both AUPRC and FPR95, demonstrating its robust detection capabilities across diverse graph structures and domains.

Table 4: Comparison in terms of AUPRC(%↑). "Avg. AUPRC" and "Avg. Rank" are computed across all datasets. Top-3 performance are indicated by light gold / light silver / light bronze cell shading with superscripts ①/②/③.

| Method | MUTAG | PROTEINS | DD | ENZYMES | DHFR | BZR | COX2 | AIDS | IMDB-B | NCI1 | COLLAB | REDDIT-B | Avg. AUPRC | Avg. Rank |
|---|---|---|---|---|---|---|---|---|---|---|---|---|---|---|
| *Graph Kernel + Detector* | | | | | | | | | | | | | | |
| PK-SVM | 47.80±0.25 | 44.07±0.20 | 45.94±0.25 | 40.83±0.22 | 45.28±0.20 | 47.04±0.20 | 41.15±0.18 | 42.46±0.23 | 41.58±0.23 | 41.97±0.22 | 43.76±0.25 | 45.22±0.21 | 43.93 | 10.58 |
| PK-iF | 45.89±0.21 | 31.38±0.19 | 16.29±0.11 | 43.65±0.19② | 40.46±0.18 | 33.70±0.15 | 40.80±0.19 | 41.30±0.22 | 40.42±0.21 | 43.22±0.19 | 41.98±0.20 | 47.55±0.24 | 38.89 | 11.42 |
| WL-SVM | 50.94±0.24 | 43.39±0.23 | 41.89±0.19 | 21.56±0.18 | 30.43±0.15 | 74.22±0.32 | 75.41±0.30 | 10.97±0.11 | 49.04±0.19 | 63.42±0.28 | 43.12±0.19 | 61.20±0.30 | 47.13 | 11.25 |
| WL-iF | 55.15±0.22 | 48.02±0.27 | 46.61±0.18 | 24.14±0.21 | 34.42±0.19 | 76.27±0.28 | 81.08±0.32 | 29.69±0.15 | 59.29±0.32 | 51.60±0.23 | 52.15±0.35 | 74.60±0.34 | 52.75 | 9.42 |
| *GNN-based Deep Learning Methods* | | | | | | | | | | | | | | |
| OCGIN | 74.07±0.24 | 79.78±0.39 | 84.16±0.33 | 32.67±0.30 | 50.84±0.29 | 88.32±0.33 | 82.95±0.35 | 93.42±0.31 | 60.34±0.18 | 54.19±0.22 | 56.97±0.32 | 85.63±0.36 | 70.28 | 5.83 |
| GLocalKD | 75.45±0.24 | 76.98±0.30 | 87.45±0.39② | 29.32±0.24 | 47.32±0.25 | 84.17±0.28 | 76.55±0.29 | 95.28±0.35① | 54.49±0.20 | 40.15±0.16 | 47.89±0.25 | 81.47±0.28 | 66.38 | 7.42 |
| OCGTL | 79.70±0.33 | 70.85±0.28 | 73.50±0.21 | 24.09±0.18 | 38.92±0.21 | 79.82±0.25 | 82.45±0.32 | 97.89±0.40① | 58.45±0.21 | 67.45±0.32② | 46.11±0.18 | 88.37±0.33② | 67.30 | 6.75 |
| SIGNET | 77.41±0.25 | 75.21±0.20 | 75.43±0.25 | 22.17±0.12 | 58.64±0.35 | 91.98±0.40① | 86.35±0.39① | 63.58±0.20 | 67.12±0.38③ | 66.45±0.31③ | 69.24±0.40② | 85.46±0.31 | 69.92 | 5.25 |
| GLADC | 72.12±0.28 | 81.93±0.40 | 74.14±0.28 | 23.88±0.21 | 44.65±0.23 | 83.02±0.27 | 82.47±0.30 | 89.43±0.31 | 61.72±0.26 | 48.83±0.25 | 58.47±0.22 | 79.81±0.23 | 66.71 | 7.50 |
| CVTGAD | 81.29±0.30③ | 83.92±0.35② | 72.87±0.30 | 30.18±0.20 | 52.43±0.30 | 90.12±0.35② | 85.23±0.37② | 96.35±0.38② | 68.89±0.35② | 65.12±0.25 | 70.59±0.35③ | 88.23±0.27③ | 73.77② | 3.58② |
| MUSE | 82.67±0.27② | 82.34±0.31③ | 85.23±0.34② | 31.89±0.24 | 61.34±0.28② | 87.23±0.29 | 65.87±0.29 | 94.23±0.18 | 58.67±0.28 | 65.45±0.32 | 63.89±0.34 | 87.89±0.29 | 71.93② | 4.50③ |
| UniFORM | 80.45±0.29 | 81.56±0.33 | 83.78±0.32 | 34.56±0.32③ | 58.92±0.31③ | 87.65±0.33 | 61.45±0.29 | 94.12±0.20 | 56.34±0.31 | 64.78±0.35 | 62.34±0.37 | 86.45±0.32 | 71.03 | 5.67 |
| **LGKDE** | 96.75±0.35① | 85.08±0.22① | 89.31±0.29① | 52.79±0.33① | 78.64±0.27① | 88.79±0.35① | 84.98±0.28③ | 95.27±0.13 | 87.17±0.34① | 92.22±0.58① | 68.51±0.23① | 88.59±0.25① | 84.01① | 1.75① |

Table 5: Comparison in terms of FPR95(%↓). "Avg. FPR95" and "Avg. Rank" are computed across all datasets. Top-3 performance are indicated by light gold / light silver / light bronze cell shading with superscripts ①/②/③.

| Method | MUTAG | PROTEINS | DD | ENZYMES | DHFR | BZR | COX2 | AIDS | IMDB-B | NCI1 | COLLAB | REDDIT-B | Avg. FPR95 | Avg. Rank |
|---|---|---|---|---|---|---|---|---|---|---|---|---|---|---|
| *Graph Kernel + Detector* | | | | | | | | | | | | | | |
| PK-SVM | 88.40±0.15 | 99.16±0.20 | 97.58±0.20 | 75.08±0.19 | 97.14±0.19 | 92.18±0.20 | 97.83±0.19 | 98.40±0.20 | 98.61±0.19 | 78.07±0.25 | 99.07±0.20 | 99.36±0.23 | 93.41 | 10.75 |
| PK-iF | 86.00±0.24 | 96.28±0.21 | 96.92±0.19 | 87.57±0.20 | 93.30±0.18 | 94.10±0.19 | 88.61±0.16 | 58.22±0.12 | 88.32±0.16 | 96.41±0.21 | 95.82±0.18 | 81.97±0.16 | 88.63 | 10.00 |
| WL-SVM | 78.80±0.29 | 99.78±0.25 | 98.98±0.22 | 76.40±0.18 | 98.26±0.21 | 92.94±0.21 | 97.05±0.18 | 99.78±0.22 | 99.60±0.22 | 78.51±0.22 | 99.79±0.23 | 98.00±0.20 | 93.16 | 11.17 |
| WL-iF | 86.00±0.16 | 96.89±0.22 | 97.74±0.21 | 86.80±0.21 | 93.70±0.19 | 93.01±0.20 | 89.29±0.17 | 58.75±0.14 | 89.80±0.17 | 96.69±0.23 | 96.50±0.20 | 82.90±0.17 | 89.01 | 10.67 |
| *GNN-based Deep Learning Methods* | | | | | | | | | | | | | | |
| OCGIN | 45.60±0.21 | 67.82±0.32① | 63.19±0.32③ | 78.93±0.19 | 85.21±0.20 | 72.89±0.19 | 90.07±0.19 | 13.57±0.08 | 87.89±0.20 | 98.63±0.23 | 89.64±0.19 | 53.55±0.35 | 70.58 | 6.92 |
| GLocalKD | 48.00±0.20 | 76.20±0.22 | 69.52±0.19 | 77.42±0.17 | 78.84±0.18 | 73.92±0.20 | 91.04±0.20 | 14.30±0.09 | 97.83±0.23 | 98.09±0.22 | 91.56±0.20 | 46.59±0.30③ | 71.94 | 7.67 |
| OCGTL | 39.20±0.39 | 94.88±0.24 | 92.22±0.21 | 84.97±0.21 | 94.03±0.21 | 96.65±0.23 | 85.12±0.26 | 1.40±0.01① | 83.21±0.18 | 61.25±0.30③ | 87.22±0.21 | 50.38±0.19 | 72.54 | 6.83 |
| SIGNET | 32.00±0.24② | 84.51±0.20 | 75.36±0.18 | 88.10±0.23 | 73.38±0.19③ | 71.54±0.29 | 79.14±0.30② | 24.98±0.10 | 75.50±0.25② | 80.98±0.19 | 81.56±0.26③ | 52.11±0.21 | 68.26 | 5.33 |
| GLADC | 39.20±0.03 | 71.07±0.18② | 68.20±0.17② | 69.35±0.32 | 82.76±0.20 | 76.61±0.21 | 85.67±0.23 | 6.79±0.05 | 77.10±0.24③ | 96.81±0.21 | 81.07±0.21② | 50.92±0.18 | 67.13 | 5.25 |
| CVTGAD | 38.80±0.35 | 71.16±0.30 | 72.43±0.20 | 65.37±0.18② | 81.73±0.20 | 73.31±0.20 | 85.62±0.25 | 4.22±0.03② | 75.34±0.27① | 85.12±0.20 | 82.35±0.25 | 45.87±0.17② | 65.11② | 4.42② |
| MUSE | 34.40±0.28③ | 72.45±0.29 | 70.78±0.28 | 68.23±0.32 | 75.67±0.30③ | 69.78±0.28② | 84.23±0.25 | 27.34±0.16 | 78.50±0.23 | 77.23±0.24③ | 85.89±0.30 | 58.34±0.28 | 66.90③ | 4.75③ |
| UniFORM | 35.20±0.31 | 73.12±0.31 | 69.95±0.30 | 63.45±0.34② | 78.34±0.24 | 68.23±0.29② | 85.89±0.24 | 28.45±0.18 | 79.23±0.25 | 80.67±0.28 | 82.45±0.33 | 62.78±0.31 | 67.31 | 5.41 |
| **LGKDE** | 30.80±0.27① | 68.35±0.30② | 67.26±0.25① | 60.00±0.24① | 68.95±0.23① | 66.25±0.35① | 78.08±0.41① | 6.19±0.08③ | 78.05±0.26 | 71.84±0.33② | 80.15±0.28① | 43.12±0.23① | 59.92① | 1.75① |

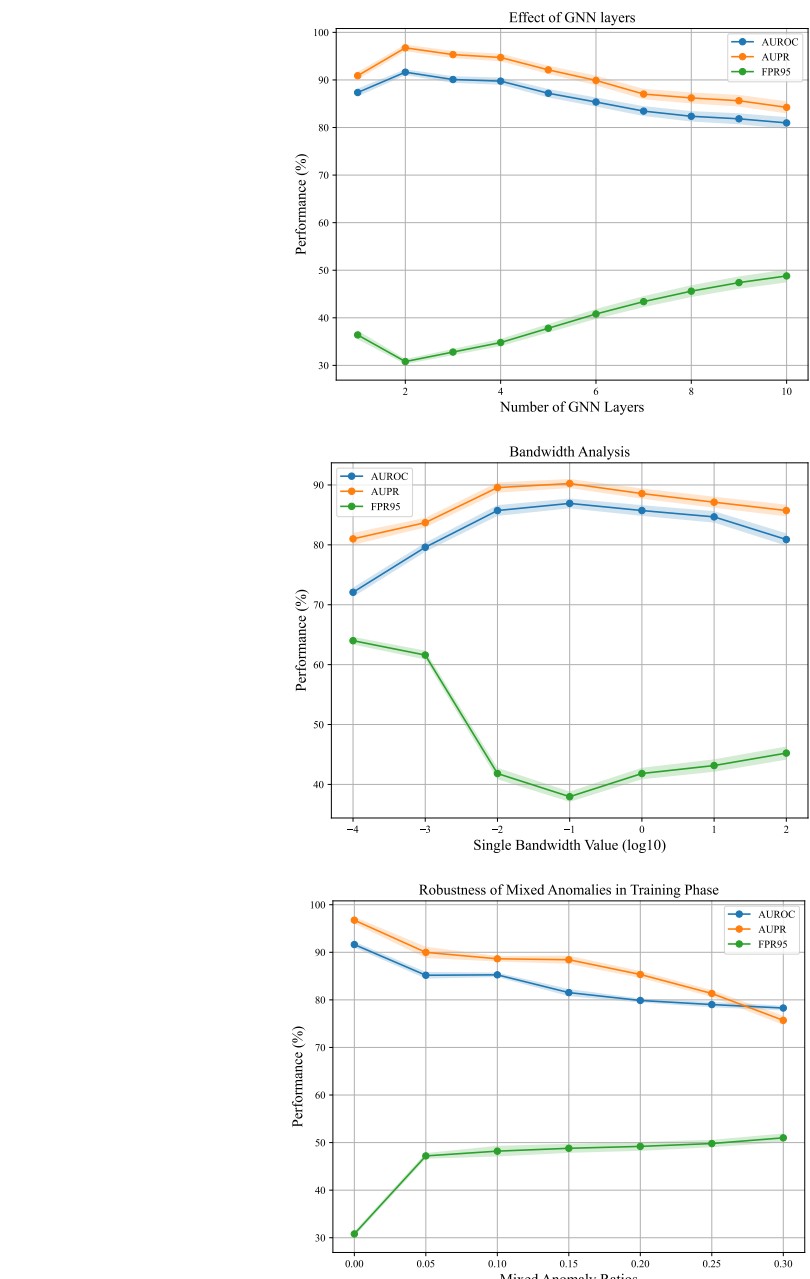

Figure 6: Ablation Studies on MUTAG Dataset

## B.4    ABLATION STUDIES

We conduct comprehensive ablation studies to analyze the impact of various components and hyper-parameters in LGKDE on the MUTAG Dataset. Table 6 and Figure 6 present the results on several key aspects:

**Multi-scale KDE Impact**    As shown in Table 6, removing the multi-scale KDE component leads to significant performance degradation. Single bandwidth variants exhibit varying performance levels, with optimal performance at moderate bandwidth values (h = 0.1) achieving 86.92% AUROC, still notably below the full model's 91.63%. This validates the effectiveness of our multi-scale approach in capturing graph patterns at different granularities.

**GNN Layer Analysis**    Figure 6(upper) shows the impact of GNN layer depth on model performance. Performance peaks at 2 layers (91.63% AUROC, 96.75% AUPRC) before gradually declining, indicating the onset of over-smoothing at deeper architectures. This suggests that moderate-depth GNNs are sufficient for capturing relevant structural information while avoiding over-smoothing effects.

**Bandwidth Sensitivity**    Figure 6 (middle) illustrates how single bandwidth values affect performance. The model shows optimal performance in the range of 0.01-1.0, with significant degradation at extreme values. This demonstrates the importance of appropriate bandwidth selection and justifies our multi-scale approach with learnable weights.

**Robustness to Training Contamination and Density Separation**    Figure 6 (bottom) examines model robustness when training data is contaminated with anomalous samples. Performance remains relatively stable up to 10% contamination (85.26% AUROC) before showing noticeable degradation at higher ratios. Complementary experiments analyzing the density gap between known normal and anomalous samples under varying contamination levels further confirm the model's ability to learn a meaningful density separation, even with polluted training data (see Appendix B.5.4 and Figure 8). This demonstrates the model's resilience to moderate levels of training data pollution, although maintaining clean training data is preferable for optimal performance.

**MMD Distance and Learnable Weights**    Replacing the MMD distance with simple graph readouts results in approximately 7% AUROC degradation. Similarly, removing learnable bandwidth weights leads to a 2.7% AUROC decrease, highlighting the importance of adaptive multi-scale modeling. These results validate the key design choices in our framework.

Table 6: Ablation study on key components of LGKDE. Results are reported as mean(%) $\pm$ std(%) over five runs.

| Variant | | AUROC | AUPRC | FPR95 |
|---|---|---|---|---|
| w/o Multi-scale KDE | $h = 10^{-2}$ | $85.73_{\pm 0.45}$ | $89.56_{\pm 0.43}$ | $41.82_{\pm 0.48}$ |
| | $h = 10^{-1}$ | $86.92_{\pm 0.41}$ | $90.24_{\pm 0.38}$ | $37.95_{\pm 0.43}$ |
| | $h = 10^{0}$ | $85.73_{\pm 0.45}$ | $88.56_{\pm 0.43}$ | $41.82_{\pm 0.48}$ |
| | $h = 10^{1}$ | $84.67_{\pm 0.48}$ | $87.12_{\pm 0.46}$ | $43.15_{\pm 0.52}$ |
| | $h = 10^{2}$ | $80.89_{\pm 0.52}$ | $85.73_{\pm 0.49}$ | $45.23_{\pm 0.55}$ |
| w/o MMD Distance | Graph Readout (avg) | $83.89_{\pm 0.49}$ | $86.92_{\pm 0.47}$ | $44.73_{\pm 0.53}$ |
| | Graph Readout (sum) | $84.73_{\pm 0.48}$ | $87.56_{\pm 0.46}$ | $43.82_{\pm 0.52}$ |
| w/o Learnable Weights | | $88.92_{\pm 0.43}$ | $89.64_{\pm 0.41}$ | $36.97_{\pm 0.45}$ |
| Full model of LGKDE | | $\mathbf{91.63_{\pm 0.31}}$ | $\mathbf{96.75_{\pm 0.35}}$ | $\mathbf{30.80_{\pm 0.27}}$ |

**Ablation on Perturbation Strategies**    To systematically evaluate the contribution of each perturbation component, we conduct controlled experiments comparing different perturbation variants on MUTAG and PROTEINS datasets. Table 7 presents the results:

Table 7: Ablation study on perturbation strategies. Results demonstrate that structure-aware spectral perturbations contribute significantly more than feature-only or random perturbations, with the full combination achieving synergistic performance gains.

| Perturbation Variant | MUTAG AUROC (%) | MUTAG AUPRC (%) | PROTEINS AUROC (%) | PROTEINS AUPRC (%) |
|---|---|---|---|---|
| Feature-only | $84.23_{\pm 0.51}$ | $89.67_{\pm 0.48}$ | $72.34_{\pm 0.34}$ | $78.56_{\pm 0.38}$ |
| Spectral-only | $87.89_{\pm 0.46}$ | $92.45_{\pm 0.42}$ | $75.67_{\pm 0.31}$ | $81.23_{\pm 0.35}$ |
| Random edges (20%) | $81.45_{\pm 0.58}$ | $86.78_{\pm 0.55}$ | $69.12_{\pm 0.41}$ | $75.89_{\pm 0.44}$ |
| **Full (feat + spectral)** | $\mathbf{91.63_{\pm 0.31}}$ | $\mathbf{96.75_{\pm 0.35}}$ | $\mathbf{78.97_{\pm 0.26}}$ | $\mathbf{85.08_{\pm 0.22}}$ |

The results reveal several key insights: (1) **Spectral perturbations are more effective than feature perturbations** (87.89% vs. 84.23% on MUTAG), demonstrating that structure-aware modifications capture more critical anomaly patterns than attribute-level noise. (2) **Random edge perturbations underperform significantly** (81.45%), validating that our energy-based spectral perturbation strategy generates higher-quality contrastive samples by targeting spectrally meaningful components rather than arbitrary edges. (3) **Combined perturbations achieve synergistic gains** (+3.74% over spectral-only), indicating that feature-level and structure-level perturbations capture complementary aspects of graph normality. These empirical findings align with our theoretical robustness analysis (Theorem 4.4,

Corollary 4.5), which formally characterizes how controlled perturbations affect the MMD distance and enable robust density learning.

**Sensitivity Analysis of Energy-Based Permutation Thresholds**  To validate the robustness of our energy-based spectral perturbation strategy, we conduct comprehensive grid-search experiments on the threshold parameters $(\tau_1, \tau_2)$. Table 8 reports AUROC performance across different threshold combinations on MUTAG, while Table 15 quantifies the corresponding average edge modification ratios.

Table 8: AUROC (%) performance for different $(\tau_1, \tau_2)$ combinations on MUTAG. The default setting $(\tau_1, \tau_2) = (0.5, 0.75)$ achieves optimal performance while maintaining robustness to modest variations.

| $\tau_1$ \ $\tau_2$ | 0.70 | 0.75 | 0.80 | 0.85 |
|---|---|---|---|---|
| 0.45 | $87.23_{\pm 0.52}$ | $89.14_{\pm 0.48}$ | $85.67_{\pm 0.54}$ | $77.89_{\pm 0.61}$ |
| 0.50 | $88.91_{\pm 0.46}$ | $\mathbf{91.63_{\pm 0.31}}$ | $89.45_{\pm 0.44}$ | $82.34_{\pm 0.56}$ |
| 0.55 | $87.56_{\pm 0.49}$ | $88.72_{\pm 0.47}$ | $85.23_{\pm 0.53}$ | $78.91_{\pm 0.58}$ |
| 0.60 | $83.45_{\pm 0.55}$ | $84.67_{\pm 0.53}$ | $80.12_{\pm 0.59}$ | $74.38_{\pm 0.64}$ |

The sensitivity results reveals that **our default setting** $(\tau_1, \tau_2) = (0.5, 0.75)$ **achieves optimal performance** (91.63% AUROC) while maintaining **relative robustness to modest variations** (88-90% AUROC within $\pm 0.05$ range for both parameters). When perturbations become too weak (small $\tau_2 - \tau_1$ gap), the density contrast diminishes and performance saturates; conversely, overly strong perturbations (large $\tau_2$ or small $\tau_1$) heavily distort graph structures, causing sharp performance degradation (e.g., 74.38% at $(\tau_1, \tau_2) = (0.60, 0.85)$).

Critically, the default setting modifies approximately 15-25% of edges (Table 15), which aligns well with the moderate perturbation strength commonly adopted in graph contrastive learning (You et al., 2020; Xu et al., 2021). However, unlike random edge perturbations that yield only 81.45% AUROC (Table 7), our energy-based strategy targets spectrally meaningful components, generating higher-quality contrastive counterparts that better characterize the boundary of normal density regions. This design choice is further validated by our robustness theorems (Theorem 4.4, Corollary 4.5), which quantify how controlled spectral perturbations affect the learned density estimates.

**Parameter-Controlled Experiments: KDE vs. GNN Capacity**  To isolate the contribution of the learnable multi-scale KDE component from the effect of additional model parameters, we conduct controlled experiments varying: (1) GNN hidden dimensions $\{32, 64, 128, 256\}$, (2) GNN layers $\{1, 2, 3, 4, 5\}$, and (3) KDE configurations: Multi-scale Learnable (ours), Multi-scale Fixed (equal weights), and Single Best bandwidth ($h = 0.1$). Table 9 presents AUROC (%) results on MUTAG across all configurations.

Table 9: AUROC (%) performance on MUTAG across different GNN architectures and KDE configurations. The learnable multi-scale KDE (top block) achieves optimal performance with moderate GNN capacity, while fixed weights (middle block) and single bandwidth (bottom block) underperform even with massively increased parameters, demonstrating that gains stem from adaptive density estimation rather than model capacity.

| KDE Configuration | Hidden Dim | Number of GNN Layers | | | | |
|---|---|---|---|---|---|---|
| | | $L = 1$ | $L = 2$ | $L = 3$ | $L = 4$ | $L = 5$ |
| **Multi-scale Learnable** *(Full LGKDE)* | 32 | $85.12_{\pm 0.48}$ | $88.94_{\pm 0.39}$ | $87.53_{\pm 0.42}$ | $87.08_{\pm 0.44}$ | $84.96_{\pm 0.49}$ |
| | 64 | $86.21_{\pm 0.45}$ | $90.28_{\pm 0.36}$ | $88.98_{\pm 0.39}$ | $88.52_{\pm 0.41}$ | $86.08_{\pm 0.47}$ |
| | 128 | $87.36_{\pm 0.42}$ | $\mathbf{91.63_{\pm 0.31}}$ | $90.09_{\pm 0.35}$ | $\mathbf{89.75_{\pm 0.38}}$ | $\mathbf{87.20_{\pm 0.45}}$ |
| | 256 | $\mathbf{87.69_{\pm 0.43}}$ | $91.48_{\pm 0.33}$ | $\mathbf{90.21_{\pm 0.37}}$ | $89.53_{\pm 0.40}$ | $87.03_{\pm 0.48}$ |
| **Multi-scale Fixed** *(Equal weights)* | 32 | $82.85_{\pm 0.51}$ | $86.34_{\pm 0.47}$ | $85.12_{\pm 0.49}$ | $84.67_{\pm 0.51}$ | $82.73_{\pm 0.54}$ |
| | 64 | $83.92_{\pm 0.49}$ | $87.65_{\pm 0.45}$ | $86.45_{\pm 0.47}$ | $86.01_{\pm 0.49}$ | $83.81_{\pm 0.52}$ |
| | 128 | $\mathbf{85.08_{\pm 0.46}}$ | $88.92_{\pm 0.43}$ | $\mathbf{87.78_{\pm 0.45}}$ | $\mathbf{87.32_{\pm 0.47}}$ | $\mathbf{84.95_{\pm 0.49}}$ |
| | 256 | $85.03_{\pm 0.47}$ | $\mathbf{89.08_{\pm 0.44}}$ | $87.51_{\pm 0.51}$ | $87.15_{\pm 0.48}$ | $84.82_{\pm 0.51}$ |
| **Single Bandwidth** *($h = 0.1$)* | 32 | $81.43_{\pm 0.53}$ | $84.12_{\pm 0.50}$ | $83.24_{\pm 0.52}$ | $82.81_{\pm 0.54}$ | $81.01_{\pm 0.56}$ |
| | 64 | $82.67_{\pm 0.51}$ | $85.58_{\pm 0.48}$ | $84.72_{\pm 0.50}$ | $84.29_{\pm 0.52}$ | $82.45_{\pm 0.54}$ |
| | 128 | $84.78_{\pm 0.48}$ | $\mathbf{86.92_{\pm 0.41}}$ | $\mathbf{86.12_{\pm 0.44}}$ | $\mathbf{85.63_{\pm 0.46}}$ | $\mathbf{83.87_{\pm 0.49}}$ |
| | 256 | $\mathbf{84.91_{\pm 0.49}}$ | $86.85_{\pm 0.43}$ | $86.03_{\pm 0.45}$ | $85.48_{\pm 0.47}$ | $83.72_{\pm 0.51}$ |

As shown in Table 9, the learnable multi-scale KDE component, not additional GNN parameters, drives performance gains:

- Learnable KDE adds only neglectable parameters (mixture weights $\alpha_k$) yet provides 4-6% AUROC gain over single bandwidth (91.63% vs. 86.92% at optimal 128-dim/2-layer) and 2-3% over fixed weights (91.63% vs. 88.92%).
- Increasing GNN capacity cannot compensate for lacking learnable KDE: Even with 74% more parameters (256-dim vs. 128-dim), single-bandwidth KDE achieves only 86.85%, remaining 4.78% below the full LGKDE with 128-dim, decisively refuting the hypothesis that gains stem from parameter inflation.
- Moderate GNN architecture suffices with adaptive KDE: Performance saturates beyond 128 hidden dimensions and 2-3 layers, indicating that the multi-scale KDE effectively captures necessary density patterns without requiring excessive representational capacity.

These results, combined with our theoretical analysis (Theorem 4.1-4.2) establishing minimax-optimal convergence rate $O(N^{-0.8})$ for the learnable multi-scale KDE, conclusively demonstrate that observed gains stem from principled density estimation rather than mere parameter inflation.

### B.5    ADDITIONAL VALIDATION

This section presents additional experiments to further validate specific aspects of LGKDE, regarding genuine density estimation capabilities and comparisons to two-stage approaches of KDE.

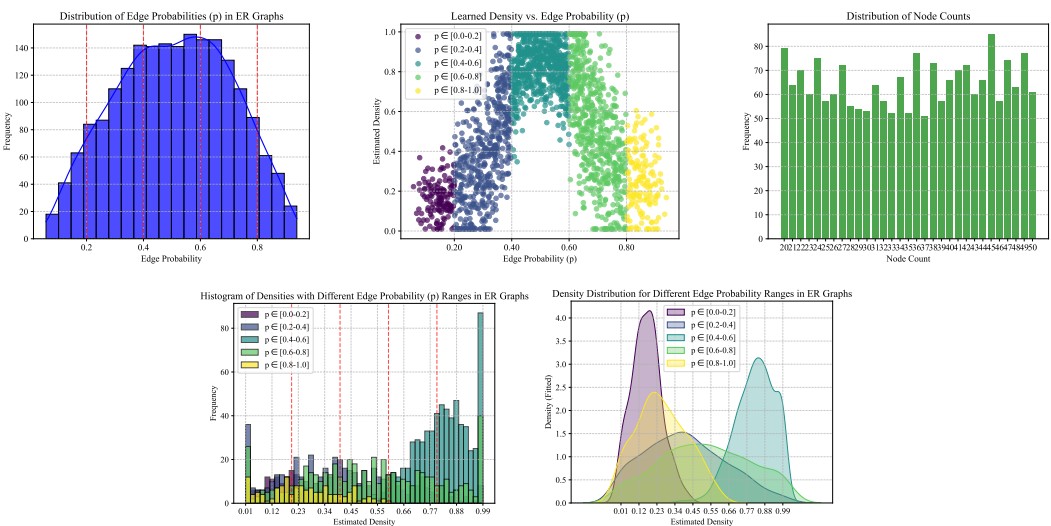

Figure 7: Density estimation results on synthetic Erdős–Rényi graphs. **Row 1 Left:** Ground truth Beta(2, 2) distribution for edge probability $p$; **Row 1 Middle:** Learned density estimate from LGKDE versus edge probability $p$, showing the expected peak around $p = 0.5$; **Row 1 Right:** Distribution of node counts (uniform as generated); **Row 2:** Histogram and Density distributions grouped by edge probability ranges.

### B.5.1    VALIDATION ON SYNTHETIC ERDŐS–RÉNYI GRAPHS

To directly assess whether LGKDE learns the underlying probability density function (PDF) of graphs, we performed experiments on synthetically generated Erdős–Rényi (ER) graphs with known generative parameters.

**Erdős–Rényi Graph Model.**    The Erdős–Rényi model $Gem_{ER}(n, p)$ generates random graphs where each possible edge between $n$ vertices occurs independently with probability $p \in [0, 1]$. Formally, for a graph $G = (V, E, \mathbf{X}) \in \mathbb{G}$ with adjacency matrix $\mathbf{A}$, each entry $\mathbf{A}_{ij}$ for $i < j$ is an

independent Bernoulli random variable with parameter $p$:

$$\mathbf{A}_{ij} = \begin{cases} 1 & \text{with probability } p \\ 0 & \text{with probability } 1 - p \end{cases} \quad \forall i, j \in \{1, 2, \ldots, n\}, i < j \tag{17}$$

The resulting graph is undirected with $\mathbf{A}_{ji} = \mathbf{A}_{ij}$ and has no self-loops ($\mathbf{A}_{ii} = 0$). For node features $\mathbf{X}$, we use the normalized degree of node $i$.

**Experimental Setup.** We generated 2000 ER graphs where the number of nodes $n$ was sampled uniformly from $[20, 50]$ and the edge probability $p$ was sampled from a Beta(2, 2) distribution, which peaks at $p = 0.5$. This specific Beta distribution yields a unimodal density that places most probability mass near $p = 0.5$, with decreasing likelihood toward the extremes of $p = 0$ and $p = 1$. LGKDE was trained on this collection of random graphs to test whether it could recover the underlying Beta(2, 2) distribution. If LGKDE accurately estimates the density, it should assign higher density values to graphs whose edge probability $p$ is closer to the mode of the Beta(2, 2) distribution (i.e., $p \approx 0.5$), irrespective of the node count $n$. Figure 7 and Table 10 present the results.

Table 10: Average estimated density by LGKDE for synthetic ER graphs, grouped by edge probability range.

| Edge Probability Range ($p$) | Average Estimated Density $\pm$ Std Dev |
|---|---|
| 0.0 – 0.2 | 0.1654 $\pm$0.0888 |
| 0.2 – 0.4 | 0.3998 $\pm$0.2428 |
| **0.4 – 0.6** | **0.8222 $\pm$0.1197** |
| 0.6 – 0.8 | 0.5222 $\pm$0.2707 |
| 0.8 – 1.0 | 0.2579 $\pm$0.1522 |

The results clearly show that LGKDE assigns the highest density to graphs with edge probabilities near 0.5, successfully recovering the underlying generative distribution parameter. This provides strong evidence that LGKDE performs genuine density estimation, not just anomaly boundary learning.

### B.5.2 ADDITIONAL SYNTHETIC GRAPH EXPERIMENTS

To comprehensively validate LGKDE's density estimation capabilities, we extend our evaluation to diverse synthetic graph types with distinct structural properties, based on the widely used generators implemented in NetworkX (Hagberg et al., 2008).

- **Barabási-Albert (Scale-free) Graphs**, which model the growth of real-world networks via preferential attachment (Barabási & Albert, 1999).
    - **Generation:** Using `barabasi_albert_graph(n, m)` to produce a power-law degree distribution $P(k) \sim k^{-3}$.
    - **Anomalies:** (i) Weakened preferential attachment ($m = 1$) or (ii) random rewiring of 30% of edges, both of which distort the power-law tail.
    - **Target density:** $f_{BA}(G) = (1 + \mathrm{KL}(P_{\mathrm{emp}} \| k^{-3}))^{-1}$, where graphs whose empirical degree distribution ($P_{\mathrm{emp}}$) better matches the theoretical power law receive higher density scores.
- **Watts-Strogatz (Small-world) Graphs**, which generate graphs with high local clustering and short average path lengths, a common feature in social networks (Watts & Strogatz, 1998).
    - **Generation:** Using `watts_strogatz_graph(n, k=4, p=0.20)`.
    - **Anomalies:** Over-regular networks (i.e., lattices, $p \leq 0.05$) or near-random networks ($p \geq 0.80$).
    - **Target density:** $f_{WS}(G) = \frac{1}{2}[s_C(C, C^*) + s_L(L, L^*)]$, where $s_C = (1 + 10(C - C^*)^2)^{-1}$ and $s_L = (1 + 0.1(L - L^*)^2)^{-1}$ reward closeness to ideal clustering $C^* = 0.5$ and path length $L^* = \log n / \log 4$.
- **Stochastic Block Model (SBM) Graphs**, a generative model for graphs with explicit community structures (Holland et al., 1983).
    - **Generation:** Three equal-sized communities with high intra-community probability ($p_{\mathrm{in}} \in [0.4, 0.8]$) and low inter-community probability ($p_{\mathrm{out}} \in [0.01, 0.10]$).
    - **Anomalies:** A flattened structure where communities dissolve ($p_{\mathrm{in}} \approx p_{\mathrm{out}}$) or an inverted (disassortative) structure ($p_{\mathrm{in}} \ll p_{\mathrm{out}}$).
    - **Target density:** $f_{SBM}(G) = \max(0, \text{modularity}) \times [1 + 0.3|c - 3|]^{-1}$, which rewards both high modularity and the correct identification of three communities ($c = 3$).

Table 11: Density estimation results on diverse synthetic graph types

| Graph Type | Normal Density | Anomaly Density | Density Gap | Detection AUC |
|---|---|---|---|---|
| Erdős-Rényi | $0.185 \pm 0.018$ | $0.062 \pm 0.015$ | 0.123 | 0.895 |
| Barabási-Albert | $0.174 \pm 0.021$ | $0.071 \pm 0.018$ | 0.103 | 0.878 |
| Watts-Strogatz | $0.168 \pm 0.019$ | $0.079 \pm 0.016$ | 0.089 | 0.856 |
| Stochastic Block | $0.201 \pm 0.022$ | $0.088 \pm 0.019$ | 0.113 | 0.923 |

These results demonstrate LGKDE's consistent ability to distinguish normal from anomalous patterns across fundamentally different graph generation models, validating its general applicability beyond specific graph types.

### B.5.3 COMPARISON WITH TWO-STAGE APPROACHES

To evaluate the benefit of LGKDE's end-to-end learning of the metric and density estimator, we compared it against two-stage approaches on the MUTAG dataset. These approaches first learn graph embeddings using standard unsupervised graph representation learning methods (InfoGraph (Sun et al., 2019a) and Graph Autoencoder - GAE (Kipf & Welling, 2016)) and then apply a standard multi-bandwidth KDE (using the same bandwidths as LGKDE but with uniform weights) on the learned embeddings.

Table 12: Comparison of LGKDE against two-stage methods (Traditional Methods and Representation Learning + KDE) and traditional baselines on MUTAG.

| Method | AUROC (%) ↑ | AUPRC (%) ↑ | FPR95 (%) ↓ |
|---|---|---|---|
| PK-SVM | $46.06 \pm 0.47$ | $47.80 \pm 0.25$ | $88.40 \pm 0.15$ |
| PK-iF | $47.98 \pm 0.41$ | $45.89 \pm 0.21$ | $86.00 \pm 0.24$ |
| WL-SVM | $62.18 \pm 0.29$ | $50.94 \pm 0.25$ | $78.80 \pm 0.29$ |
| WL-iF | $65.71 \pm 0.38$ | $55.15 \pm 0.22$ | $86.00 \pm 0.16$ |
| InfoGraph+KDE | $79.77 \pm 3.05$ | $88.36 \pm 1.53$ | $44.00 \pm 5.66$ |
| GAE+KDE | $81.94 \pm 2.21$ | $91.00 \pm 1.28$ | $49.60 \pm 5.43$ |
| LGKDE (Ours) | $\mathbf{91.63 \pm 0.31}$ | $\mathbf{96.75 \pm 0.35}$ | $\mathbf{30.80 \pm 0.27}$ |

Table 12 shows that LGKDE significantly outperforms both two-stage methods of traditional graph kernels + detector and graph representation learning + KDE. While InfoGraph and GAE learn general-purpose embeddings, these are suboptimal for the specific task of density estimation compared to the metric learned jointly within the LGKDE framework. This highlights the advantage of our integrated approach in learning task-relevant representations and distances for accurate density estimation.

### B.5.4 DENSITY GAP ANALYSIS UNDER TRAINING CONTAMINATION

To quantitatively assess density separation and robustness to label noise, we performed an experiment on MUTAG where we trained LGKDE on the normal class mixed with a varying number of known anomalous samples (treated as normal during training). We then measured the average density assigned by the trained model to the true normal graphs versus the true anomalous graphs (held-out).

Table 13: Density gap results on MUTAG under training contamination.

| # Anomalies in Training | Avg. Normal Density | Avg. Anomaly Density | Density Gap |
|---|---|---|---|
| 50 | 0.151 | 0.090 | 0.061 |
| 40 | 0.158 | 0.083 | 0.075 |
| 30 | 0.168 | 0.077 | 0.091 |
| 20 | 0.180 | 0.066 | 0.115 |
| 10 | 0.192 | 0.058 | 0.134 |
| 5 | 0.202 | 0.053 | 0.149 |
| 0 | 0.215 | 0.048 | 0.167 |

Figure 8 and Table 13 show that even with significant contamination (50 anomalous samples included), LGKDE maintains a positive density gap. As the contamination level decreases, the model learns a better representation of the true normal distribution, resulting in higher density estimates for normal graphs, lower estimates for anomalous graphs, and a monotonically increasing density gap. This demonstrates both the model's robustness and its ability to learn a meaningful density separation.

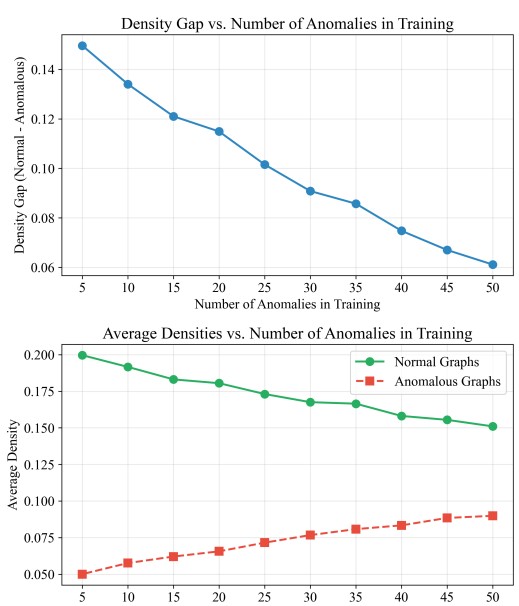

Figure 8: Density gap analysis on MUTAG. Shows the average density assigned to true normal vs. true anomalous graphs when the model is trained with varying numbers of anomalous samples included in the 'normal' training set. The gap increases as contamination decreases.

### B.5.5 QUALITATIVE ANALYSIS OF LEARNED DENSITIES

To provide qualitative insight into what LGKDE learns, we visualized graphs from the MUTAG dataset corresponding to different density levels estimated by the trained LGKDE model. We identified the "average" graph (minimum mean MMD distance to all others) and samples with the highest and lowest estimated densities.

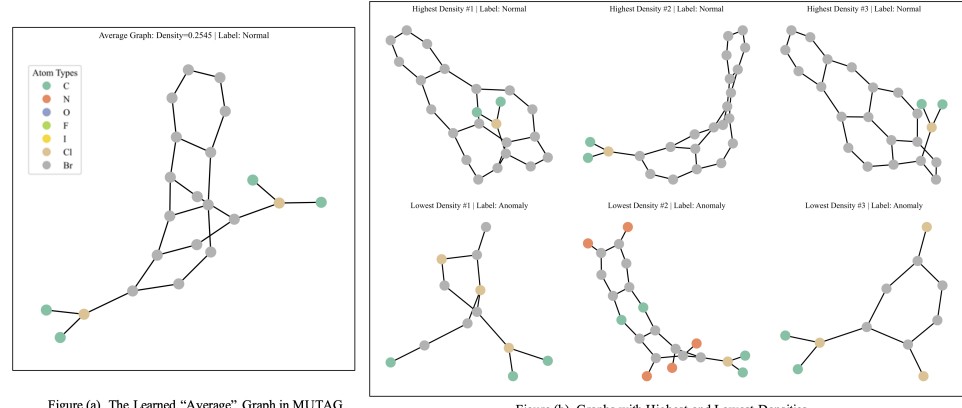

Figure (a). The Learned "Average" Graph in MUTAG.

Figure (b). Graphs with Highest and Lowest Densities.

Figure 9: Visualization of MUTAG graphs. (a) The graph with minimum average MMD distance to all other graphs ("average" graph). (b) Top row: examples of graphs assigned the highest density by LGKDE. Bottom row: examples of graphs assigned the lowest density.

Figure 9 reveals structural differences captured by the learned density. High-density (normal) graphs often exhibit more complex ring structures characteristic of the majority class in MUTAG, while low-density (potentially anomalous) graphs tend to have simpler or atypical structures. This visualization supports the idea that LGKDE learns structurally relevant patterns for density estimation.

## B.6 ROBUSTNESS UNDER CONTROLLED DISTRIBUTION SHIFTS

To further quantitatively validate our motivation that discriminative graph anomaly detection methods are fragile to distribution shifts, we conduct systematic controlled experiments on synthetic Erdős–Rényi (ER) graphs.

**Training Phase (Identical for All Methods).** To ensure fair comparison, all methods (LGKDE, OCGIN, SIGNET) are trained on the same "normal" distribution:

- **Sample size**: 2000 ER graphs
- **Edge probability**: $p \sim \mathcal{N}(0.5, 0.1)$, truncated to $[0.2, 0.8]$ to ensure valid graph structures
- **Node count**: $n \sim \text{Uniform}[20, 50]$
- **Training objective**: Each method learns to model this "normal" distribution using its respective loss function (LGKDE uses density contrast; OCGIN/SIGNET use discriminative objectives)

**Test Phase (Revealing Robustness Differences).** After training, we evaluate all methods on three progressively challenging test scenarios with controlled distribution shifts. Each test scenario contains 500 normal samples and 500 anomalous samples:

1. **In-Distribution (ID)**: Training and test distributions match (baseline performance)
   - Normal samples: $p \sim \mathcal{N}(0.5, 0.1)$ (same as training)
   - Anomalous samples: $p \sim \text{Uniform}[0.0, 0.2] \cup [0.8, 1.0]$ (extreme edge densities)
2. **Distribution Shift 1 (Sparse)**: Test normal distribution shifts toward sparse graphs
   - Normal samples: $p \sim \mathcal{N}(0.3, 0.1)$ (shifted $-0.2$ from training mean)
   - Anomalous samples: $p \sim \text{Uniform}[0.0, 0.15]$ or heavily rewired graphs (50% edge swaps)
3. **Distribution Shift 2 (Dense)**: Test normal distribution shifts toward dense graphs
   - Normal samples: $p \sim \mathcal{N}(0.7, 0.1)$ (shifted $+0.2$ from training mean)
   - Anomalous samples: $p \sim \text{Uniform}[0.85, 1.0]$ or heavily rewired graphs (50% edge swaps)

We report AUROC (%) for anomaly detection performance, with degradation measured as:

$$\text{Avg. Degradation} = \frac{1}{2}\left(\text{AUROC}_{\text{ID}} - \text{AUROC}_{\text{Shift1}}\right) + \frac{1}{2}\left(\text{AUROC}_{\text{ID}} - \text{AUROC}_{\text{Shift2}}\right) \quad (18)$$

Table 14: AUROC (%) performance under controlled distribution shifts on synthetic ER graphs. LGKDE maintains significantly better robustness (only 8.41% average degradation) compared to discriminative baselines OCGIN and SIGNET (16-17% degradation), validating our motivation that density-based methods are more resilient to distribution shifts.

| Method | In-Distribution | Shift 1 (Sparse) | Shift 2 (Dense) | Avg. Degradation |
|--------|-----------------|------------------|-----------------|------------------|
| OCGIN | $77.38_{\pm 0.64}$ | $61.24_{\pm 1.27}$ | $62.15_{\pm 1.46}$ | $-15.69\%$ |
| SIGNET | $85.12_{\pm 0.80}$ | $68.73_{\pm 1.51}$ | $69.89_{\pm 1.63}$ | $-15.81\%$ |
| **LGKDE** | $\mathbf{89.45_{\pm 0.53}}$ | $\mathbf{82.67_{\pm 0.90}}$ | $\mathbf{83.91_{\pm 0.87}}$ | $\mathbf{-6.16\%}$ |

Table 14 empirically validates the resilience of density-based methods against distribution shifts. While discriminative baselines (OCGIN, SIGNET) suffer severe performance collapse ($\sim 16\%$ drop) due to rigid decision boundaries becoming misaligned with shifted data, LGKDE maintains remarkable stability with only a 6.16% average decline. This robustness stems from our approach of modeling the intrinsic density landscape rather than fixed separation hyperplanes, allowing LGKDE to preserve accurate ranking orders across both sparse and dense shifts. Notably, LGKDE's performance under shift ($> 82\%$) still surpasses the *in-distribution* results of the strongest baseline ($77.38\%$), highlighting its superior generalization capability.

## C RELATED WORK

Due to space limits, the related literature review on graph representation learning is presented in this Appendix C.1. And the extra discussion and comparison with deep density related methods is provided in the following Appendix C.2.

## C.1 Graph Representation Learning

Graph representation learning has emerged as a fundamental paradigm for analyzing graph-structured data (Hamilton, 2020). Early approaches focused on matrix factorization and random walk-based methods (Perozzi et al., 2014; Grover & Leskovec, 2016), which learn node embeddings by preserving local neighborhood structures. The advent of Graph Neural Networks (GNNs) has revolutionized this field by enabling end-to-end learning of graph representations through message passing (Gilmer et al., 2017). Modern GNN architectures, such as Graph Convolutional Networks (GCN) (Kipf & Welling, 2017) and Graph Isomorphism Networks (GIN) (Xu et al., 2019), have demonstrated remarkable success in various supervised learning tasks. These rich representation capability makes graphs an essential tool for understanding and analyzing real-world systems ranging from social networks to molecular structures (Wang & Li, 2020; Sun et al., 2019b; Wang et al., 2022).

However, the challenge of learning effective graph representations without supervision remains significant. Recent unsupervised approaches have explored various strategies, including graph autoencoders (Kipf & Welling, 2016), contrastive learning (Velickovic et al., 2019), and mutual information maximization (Sun et al., 2019a), graph entropy maximization (Sun et al., 2024). While these methods have shown promise, they often struggle to capture fine-grained structural patterns and maintain theoretical guarantees, particularly when applied to graph-level representations. Moreover, the lack of supervision makes it especially challenging to learn representations that can effectively distinguish normal patterns from anomalous ones (Ma et al., 2021).

## C.2 Comparison with Deep Density Related Methods

LGKDE differs fundamentally from typical deep density estimation approaches (normalizing flows, VAEs, EBMs) in both methodology and applicability to graphs:

**Normalizing Flows for Graphs.** Flow-based models (Liu et al., 2019) parameterize densities via invertible neural networks that transform simple base distributions. While theoretically elegant, these methods face critical challenges for graphs: (1) Invertibility constraints impose architectural restrictions that struggle with variable graph sizes and topology changes; (2) Fixed-size assumptions typically require padding or graph coarsening, introducing artifacts; (3) Lack of theoretical guarantees for consistency and convergence in discrete graph spaces. In contrast, LGKDE naturally handles variable-size graphs via GNN+MMD embeddings without invertibility constraints and provides explicit consistency guarantees (Theorem 4.1) and optimal convergence rates (Theorem 4.2).

**Energy-Based Models (EBMs) for Graphs.** Graph EBMs (Liu et al., 2020; 2021) define energy functions over graphs and rely on expensive MCMC sampling (Langevin dynamics) for training and inference. While flexible, EBMs suffer from: (1) Computational instability due to high-variance gradient estimation; (2) Mode coverage issues when sampling complex multi-modal graph distributions; (3) Lack of explicit density parameterization, making anomaly scoring indirect. LGKDE performs direct density estimation via KDE with closed-form evaluation, avoiding sampling altogether while maintaining computational efficiency (Appendix E.4.1).

**Graph Variational Autoencoders (VAEs).** VAE-based methods (Kipf & Welling, 2016) optimize an evidence lower bound (ELBO) but face: (1) Loose ELBO leading to suboptimal density estimates; (2) Posterior collapse where the decoder ignores latent codes; (3) No explicit density reconstruction error serves as a heuristic proxy. LGKDE explicitly models density via KDE with proven consistency, avoiding variational approximation gaps.

**Where LGKDE's Density Lives.** Unlike flows/VAEs/EBMs that define densities in learned latent spaces, LGKDE maintains density directly in a nonparametric form over the graph space, represented as KDE in a learned MMD metric space. The GNN+MMD module learns only the metric; the density itself is from the multi-scale KDE with learnable mixture weights, jointly optimized end-to-end. This design combines the flexibility of nonparametric methods with the representational power of deep learning, yielding a principled density estimator with complete theoretical guarantees (Theorems 4.1-4.4, Corollary 4.5, Theorem 4.6).

Sun & Fan (2024) introduces a deep MMD graph kernel that learns a metric for tasks such as clustering and classification. Our deep MMD module is inspired by this work, so we also construct a learnable MMD-based metric over graphs. However, LGKDE builds on top of this backbone in several directions:

- We utilize the learned metric to a multi-scale KDE with learnable mixture weights, which produces an explicit nonparametric density over graphs rather than only a similarity or kernel value, aim to address the challenge and the gap of density estimation over graphs.
- We couple this KDE with a novel structure-aware perturbation mechanism and a density-contrast objective designed for unsupervised graph anomaly detection, which can effectively capture the underlying graph distribution.
- We established a theoretical framework for consistency, convergence, robustness, and generalization, and provided a detailed analysis of complexity and scalability for the resulting density estimator.

Specifically, when fixing the KDE bandwidths and mixture weights, LGKDE effectively reduces to learn a deep MMD graph kernel, then running a basic KDE on the resulting distances. This can be viewed as a Sun & Fan (2024) + KDE style baseline within our own framework. To avoid an uninformative comparison between LGKDE and such a degenerate special case of itself, we instead evaluate more general two-stage baselines in the Appendix B.5.3.

## D  MORE PERTURBED SAMPLE GENERATION DETAILS

### D.1  PSEUDOCODE OF STRUCTURE-AWARE SAMPLE GENERATION

See Algorithm 1.

### D.2  CASE STUDY ON ENERGY-BASED SPECTRAL PERTURBATION

Figure 10 and 11 show the visualization of the different combinations of hyperparameters $\tau_1$ and $\tau_2$ in our energy-based spectral perturbation strategy on the first graph from MUTAG datasets. We can find that when $\tau_1$ around $0.5$ with $\tau_2$ around $0.75$, our method generates structurally meaningful variations (add/remove a suitable number of links in the ring structure) while preserving the graph's core topology.

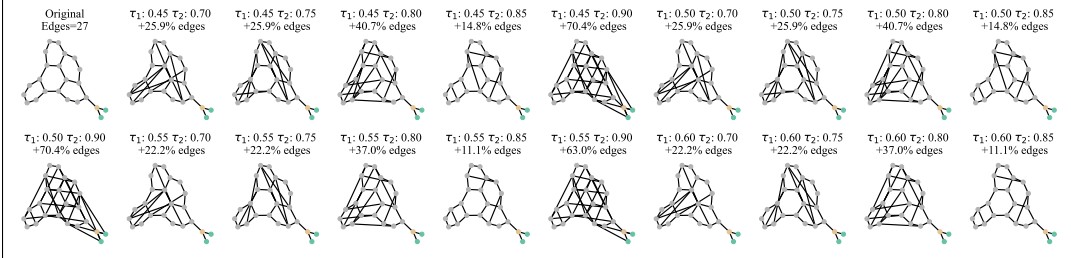

Figure 10: Spectral Perturbation: Add Edges Mode via Amplify $\mathcal{S}_l$ on the 1st MUTAG sample

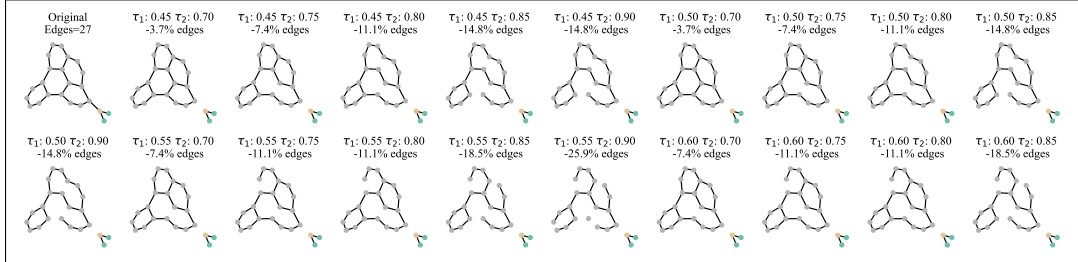

Figure 11: Spectral Perturbation: Remove Edges Mode via Shrink $\mathcal{S}_h$ on the 1st MUTAG sample

Figure 12. shows that the histogram of edge change ratio of spectral perturbation on the whole MUTAG datasets with $\tau_1 = 0.5$ and $\tau_2 = 0.75$, we can find that the majority of graphs have the

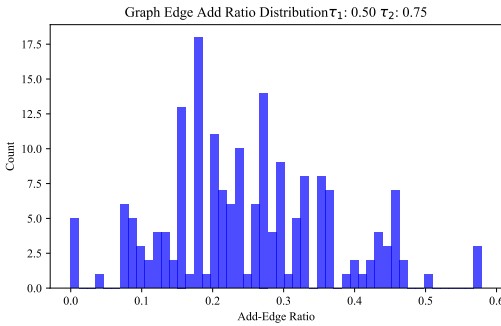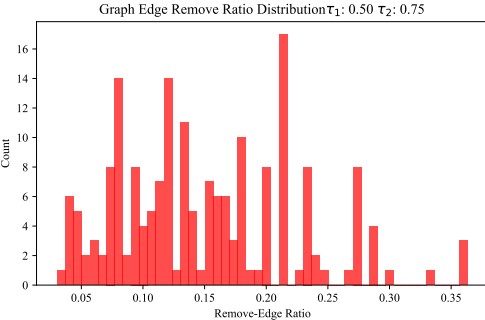

Figure 12: Edge Change Ratio of Spectral Perturbation on the whole MUTAG datasets with $\tau_1 = 0.5$ and $\tau_2 = 0.75$

permutation around add 20% edges or reduce 15% edges. What's more, these permutation is taken on the graph key structure instead of random edge permutations, which tends to generate well-designed perturbed counterparts and help better characterize the boundary of normal density regions during the unsupervised contrastive learning. Table 15 further gives detailed Avg. edge change ratios (%) $\pm$ standard deviation across MUTAG graphs for different $(\tau_1, \tau_2)$ combinations.

Table 15: Average edge change ratios (%) $\pm$ standard deviation across MUTAG graphs for different $(\tau_1, \tau_2)$ combinations.

| $(\tau_1, \tau_2)$ | Add Edges (%) | Remove Edges (%) | $(\tau_1, \tau_2)$ | Add Edges (%) | Remove Edges (%) |
|---|---|---|---|---|---|
| $(0.45, 0.70)$ | $26.42_{\pm13.11}$ | $10.90_{\pm6.45}$ | $(0.55, 0.70)$ | $19.39_{\pm11.02}$ | $14.76_{\pm8.08}$ |
| $(0.45, 0.75)$ | $28.67_{\pm12.95}$ | $12.88_{\pm7.26}$ | $(0.55, 0.75)$ | $21.69_{\pm11.14}$ | $17.64_{\pm8.23}$ |
| $(0.45, 0.80)$ | $25.25_{\pm11.88}$ | $14.68_{\pm7.35}$ | $(0.55, 0.80)$ | $20.13_{\pm10.15}$ | $22.09_{\pm10.11}$ |
| $(0.45, 0.85)$ | $21.20_{\pm10.90}$ | $17.43_{\pm7.34}$ | $(0.55, 0.85)$ | $16.85_{\pm10.63}$ | $27.29_{\pm10.87}$ |
| $(0.50, 0.70)$ | $22.64_{\pm11.87}$ | $13.13_{\pm6.98}$ | $(0.60, 0.70)$ | $16.65_{\pm11.05}$ | $19.77_{\pm11.59}$ |
| $(0.50, 0.75)$ | $25.06_{\pm11.89}$ | $15.14_{\pm7.26}$ | $(0.60, 0.75)$ | $18.60_{\pm10.88}$ | $23.70_{\pm12.13}$ |
| $(0.50, 0.80)$ | $23.20_{\pm10.65}$ | $17.88_{\pm8.41}$ | $(0.60, 0.80)$ | $17.40_{\pm9.97}$ | $28.80_{\pm12.66}$ |
| $(0.50, 0.85)$ | $19.27_{\pm10.90}$ | $21.41_{\pm8.90}$ | $(0.60, 0.85)$ | $13.66_{\pm8.91}$ | $37.50_{\pm15.32}$ |

# E    LGKDE FRAMEWORK DETAILS

## E.1    MULTI-SCALE KERNEL DESIGN

The hyperparameter set $\Gamma_{\text{emb}} = \{\gamma_1, \ldots, \gamma_S\}$ is chosen to cover multiple scales of the MMD calculation. In practice, we use $S = 5$ and $1/\gamma$ as bandwidths in gaussian kernel $k_\gamma^{\text{emb}}$, which are logarithmically spaced between $\gamma_{min}$ and $\gamma_{max}$. The bandwidth set $H_{KDE} = \{h_1, \ldots, h_M\}$ is used for the multi-scale kernel density estimator. Since they are also Gaussian kernels in MMD calculation and density estimation, and the learnable weights design gives the flexibility to automatically determine the relative kernel importance, we set $M = 5$ and use the same bandwidth in multi-scale KDE $H_{KDE}$ as that in deep MMD calculation. All these designs ensure the kernels can capture both fine-grained local structures and global patterns.

The mixture weights $\{\pi_k\}$ are initialized uniformly and updated through gradient descent on $\boldsymbol{\alpha}$. The softmax parameterization ensures $\sum_k \pi_k = 1$ while allowing unconstrained optimization of $\boldsymbol{\alpha}$.

## E.2    OPTIMIZATION STRATEGY

The complete training and inference procedure of LGKDE is detailed in Algorithm 3. Here we elaborate on key implementation aspects of our optimization approach. The contrastive objective in Eq. (10) can be interpreted as maximizing the difference of densities between normal graphs and their perturbed versions:

$$\min_{\boldsymbol{\theta}, \boldsymbol{\alpha}} \mathcal{L} := -\sum_{i=1}^{N} \sum_{j=1}^{N_{\text{pert}}} \frac{\hat{f}(G_i) - \hat{f}(\tilde{G}_i^{(j)})}{\hat{f}(G_i) + \epsilon} \tag{19}$$

---

**Algorithm 1 Structure-aware Sample Generation**

---

**Require:** Graph $G = (\mathbf{A}, \mathbf{X})$, swap ratio $r_{\text{swap}}$, energy thresholds $\tau_1, \tau_2$, edge operation flag $flag \in \{0, 1\}$ (0: removal, 1: addition)
**Ensure:** Perturbed graph $\tilde{G}$ with altered structure and features
  1: **function** GENERATESAMPLE($G, r_{\text{swap}}, \tau_1, \tau_2, flag$)
  2:     $\tilde{\mathbf{X}} \leftarrow$ NODEFEATUREPERTURB$\big(\mathbf{X}, r_{\text{swap}}\big)$          ▷ Swap a fraction of node features
  3:     $\tilde{\mathbf{A}} \leftarrow$ SPECTRALPERTURB$\big(\mathbf{A}, \tau_1, \tau_2, flag\big)$      ▷ Energy-based spectral modification
  4:     $\tilde{G} \leftarrow (\tilde{\mathbf{A}}, \tilde{\mathbf{X}})$
  5:     **return** $\tilde{G}$
  6: **end function**

  7: **function** NODEFEATUREPERTURB($\mathbf{X}, r_{\text{swap}}$)
  8:     $\tilde{\mathbf{X}} \leftarrow \mathbf{X}$                                    ▷ Initialize copy
  9:     Let $\mathcal{V}_{\text{swap}}$ be a random subset of node indices of size $\lfloor r_{\text{swap}} \cdot |V| \rfloor$
10:     $\tilde{\mathbf{X}}_v \leftarrow \mathbf{X}_{perm(v)}$ for each $v \in \mathcal{V}_{\text{swap}}$      ▷ Shuffle selected nodes' features
11:     **return** $\tilde{\mathbf{X}}$
12: **end function**

13: **function** SPECTRALPERTURB($\mathbf{A}, \tau_1, \tau_2, flag$)
14:     $\mathbf{U}, \boldsymbol{\Sigma}, \mathbf{V}^\top \leftarrow$ SVD($\mathbf{A}$)                  ▷ SVD decomposition
15:     Compute cumulative energy ratio $E(k)$ via Eq. (3)
16:     Partition singular values into $\mathcal{S}_h, \mathcal{S}_m, \mathcal{S}_l$ using thresholds $\tau_1, \tau_2$
17:     Compute adaptive ratio $r = \min(\mu_h/\mu_l, r_{\max})$, where $\mu_h$ and $\mu_l$ are means of $\mathcal{S}_h$ and $\mathcal{S}_l$
18:     Modify singular values based on $flag$ via Eq. (5):
19:       If $flag = 0$ (removal): $\tilde{\sigma}_i = \sigma_i/r$ for $\sigma_i \in \mathcal{S}_h$
20:       If $flag = 1$ (addition): $\tilde{\sigma}_i = r\sigma_i$ for $\sigma_i \in \mathcal{S}_l$
21:     $\tilde{\boldsymbol{\Sigma}} \leftarrow$ updated diagonal matrix with modified singular values
22:     $\tilde{\mathbf{A}} \leftarrow \mathbf{U} \, \tilde{\boldsymbol{\Sigma}} \, \mathbf{V}^\top$
23:     **return** $\tilde{\mathbf{A}}$
24: **end function**

---

where $\tilde{G}_i^{(j)}$ denotes the $j$th perturbed counterpart of $G_i$ and we add $\epsilon$ ($\epsilon = 1.0\text{e-}6$ in our experiment) for numerical stability. The objective effectively pushes normal graphs towards high-density regions while moving perturbed versions towards lower-density regions, creating a clear separation in the density landscape.

The parameters $\boldsymbol{\theta}$ and $\boldsymbol{\alpha}$ can be optimized jointly using Adam with learning rates $\eta_{\boldsymbol{\theta}}$ and $\eta_{\boldsymbol{\alpha}}$ respectively. We employ gradient clipping and early stopping based on validation set performance to prevent overfitting. The learning rates are scheduled with a warm-up period followed by cosine decay.

### E.3   INFERENCE PROTOCOL

The percentile-based threshold estimation is robust to outliers and automatically adapts to the scale of density values. In practice, we set $\gamma_{TH} = 10$ as a reasonable default, though this can be adjusted based on domain knowledge about expected anomaly rates.

For computational efficiency during inference, we maintain a reference set of pre-computed embeddings for normal graphs. This allows fast computation of MMD distances without recomputing embeddings for the reference graphs. The memory requirement scales linearly with the size of the reference set.

### E.4   COMPUTATIONAL COMPLEXITY AND SCALABILITY

Let $N$ be the number of graphs in a batch, $n$ and $m$ be the average number of nodes and edges in these $N$ graphs, respectively. Let $d_{in}, d_{hid}$, and $d_{out}$ be the input, hidden, and output dimensions of

---

**Algorithm 2** Deep Graph MMD Model for Graph Distance Computation

---

**Require:**
1: Two sets of graphs $\mathcal{G}_1 = \{G_1^1, ..., G_1^{N_1}\}$, $\mathcal{G}_2 = \{G_2^1, ..., G_2^{N_2}\}$ (if $\mathcal{G}_2 =$ None, set $\mathcal{G}_2 \leftarrow \mathcal{G}_1$)
2: GNN parameters $\boldsymbol{\theta} = \{\mathbf{W}^{(l)} \in \mathbb{R}^{d_{l-1} \times d_l}\}_{l=1}^L$ for model $\Phi_{\boldsymbol{\theta}}$
3: MMD kernel family $\mathcal{K}_{\text{emb}}$ with bandwidths $\Gamma_{\text{emb}} = \{\gamma_1, \ldots, \gamma_S\}$
4: Activation function (ReLU): $\sigma$
**Ensure:** Distance matrix $\mathbf{D} \in \mathbb{R}^{N_1 \times N_2}$ containing pairwise MMD distances
5: **function** COMPUTEMMDDISTANCES($\mathcal{G}_1, \mathcal{G}_2, \boldsymbol{\theta}, \mathcal{K}_{\text{emb}}$)
6:     // Generate node embeddings via GNN encoder $\Phi_{\boldsymbol{\theta}}$
7:     **for** $s \in \{1, 2\}$ **do**
8:         **for** $i = 1$ to $N_s$ **do**
9:             // Compute $\mathbf{Z}_s^i = \Phi_{\boldsymbol{\theta}}(G_s^i)$ with parameters $\boldsymbol{\theta}$
10:            Let $\mathbf{A}_s^i$ be the adjacency matrix of $G_s^i$
11:            Let $\mathbf{D}_s^i = \text{diag}(\sum_j \mathbf{A}_s^i[j, :] + 1)$ be the degree matrix with self-loops
12:            $\mathbf{Z}_s^{i,(0)} \leftarrow \mathbf{X}_s^i$            ▷ Initial node features
13:            **for** $l = 1$ to $L$ **do**
14:                $\hat{\mathbf{A}}_s^i \leftarrow (\mathbf{D}_s^i)^{-1/2}(\mathbf{A}_s^i + \mathbf{I})(\mathbf{D}_s^i)^{-1/2}$    ▷ Normalized adjacency
15:                $\mathbf{Z}_s^{i,(l)} \leftarrow \sigma(\hat{\mathbf{A}}_s^i \mathbf{Z}_s^{i,(l-1)} \mathbf{W}^{(l)})$     ▷ GNN layer update
16:            **end for**
17:            $\mathbf{Z}_s^i \leftarrow \mathbf{Z}_s^{i,(L)}$            ▷ Final node embeddings
18:         **end for**
19:     **end for**
20:     // Compute pairwise MMD distances using node embeddings
21:     **for** $i = 1$ to $N_1$ **do**
22:         **for** $j = 1$ to $N_2$ **do**
23:             Compute $\mathbf{D}_{ij} = d_{MMD}(G_1^i, G_2^j)$ using Eq. (7)
24:              where each kernel evaluation uses Eq. (21) with embeddings $\mathbf{Z}_1^i$ and $\mathbf{Z}_2^j$
25:         **end for**
26:     **end for**
27:     **return D**            ▷ Matrix of pairwise MMD distances
28: **end function**

---

the GNN, and $L$ be the number of GNN layers. $M$ is the number of KDE bandwidths, and $S$ is the number of MMD kernel bandwidths.

**Proposition E.1** (Computational Complexity). *The computational complexity for embedding a graph is dominated by the GNN ($O(L(md_{hid} + nd_{hid}^2))$), and computing pairwise MMD using the quadratic estimator takes $O(Sn^2d_{\text{out}})$. Inference involves $N$ MMD computations, which require $O(NSn^2d_{\text{out}})$. Combined, we can get the LGKDE process per graph's density estimation with $O(L(md_{hid} + nd_{hid}^2) + NSn^2d_{\text{out}})$.*

*Proof.* We first analyze the complexity component-wise.

- **GNN Embedding:** For a single graph, a standard GNN forward pass takes $O(L(md_{hid} + nd_{hid}^2))$.
- **Pairwise MMD Computation:** The deep MMD distance between two graphs with $n_i$ and $n_j$ nodes involves kernel computations over their node embeddings. Using a quadratic-time estimator, this takes $O(S(n_i^2 + n_j^2)d_{out})$.
- **KDE Score Calculation:** For a single graph, computing its density score requires comparing it to all $N$ reference graphs, involving $N$ MMD calculations and $N \times M$ evaluations. This totals $O(NSn^2d_{out} + NM)$ and since $M$ is the number of KDE bandwidths and $M \ll Sn^2d_{out}$, we get $O(NSn^2d_{out})$ for this KDE scoring process.

According to above analysis, in a training epoch with $N$ graphs, GNN embedding becomes $O(N \cdot L(md_{hid} + nd_{hid}^2))$, computing the full $N \times N$ MMD distance matrix requires then do KDE score calculation becomes $O(N^2Sn^2d_{out})$.

---

**Algorithm 3 Learnable Graph Kernel Density Estimation (LGKDE)**

---

**Require:**
1: Normal graph set $\mathcal{G} = \{G_1, \ldots, G_N\}$
2: GNN parameters $\boldsymbol{\theta}$ for encoder $\Phi_{\boldsymbol{\theta}}$, MMD kernel family $\mathcal{K}_{\text{emb}}$ with bandwidths $\Gamma_{\text{emb}}$
3: Multi-scale KDE bandwidth set $H_{KDE} = \{h_k\}_{k=1}^M$, learnable mixture weight parameters $\boldsymbol{\alpha}$
4: Perturbation parameters: $N_{\text{pert}}$, $r_{\text{swap}}$, energy thresholds $\tau_1, \tau_2$
5: Anomaly threshold parameter $\gamma_{TH}$ (percentile)
**Ensure:** Trained model parameters $\boldsymbol{\theta}, \boldsymbol{\alpha}$
6: **function** TRAIN($\mathcal{G}$)
7:     Initialize $\boldsymbol{\theta}$ and $\boldsymbol{\alpha}$ randomly
8:     **for** epoch = 1 to MaxEpoch **do**
9:         // Generate structure-aware perturbed samples
10:        Sample mini-batch $\mathcal{B} \subseteq \mathcal{G}$
11:        **for** each graph $G_i \in \mathcal{B}$ **do**
12:            **for** j = 1 to $N_{\text{pert}}$ **do**
13:                *flag* $\leftarrow$ randomly select from $\{0, 1\}$      $\triangleright$ 0: edge removal, 1: edge addition
14:                $\tilde{G}_i^{(j)} \leftarrow$ GENERATESAMPLE($G_i, r_{\text{swap}}, \tau_1, \tau_2, flag$)      $\triangleright$ Algorithm 1
15:            **end for**
16:        **end for**
17:        // Compute MMD distances between all graphs
18:        $\mathbf{D} \leftarrow$ COMPUTEMMDDISTANCES($\mathcal{B} \cup \{\tilde{G}_i^{(j)}\}, \mathcal{B}, \boldsymbol{\theta}, \mathcal{K}_{\text{emb}}$)      $\triangleright$ Algorithm 2
19:        // Compute multi-scale KDE for all graphs
20:        **for** each graph $G \in \mathcal{B} \cup \{\tilde{G}_i^{(j)}\}$ **do**
21:            **for** $k = 1$ to $M$ **do**
22:                $\phi_k(G) \leftarrow \frac{1}{|\mathcal{B}|} \sum_{G_m \in \mathcal{B}} K_{KDE}(d_{MMD}(G, G_m), h_k)$      $\triangleright$ k-th KDE component
23:            **end for**
24:            $\hat{f}(G) \leftarrow \sum_{k=1}^M \pi_k(\boldsymbol{\alpha})\phi_k(G)$, where $\pi_k(\boldsymbol{\alpha}) = \frac{\exp(\alpha_k)}{\sum_l \exp(\alpha_l)}$
25:        **end for**
26:        // Update parameters using contrastive objective
27:        Compute loss $\mathcal{L} \leftarrow -\sum_{i=1}^{|\mathcal{B}|} \sum_{j=1}^{N_{\text{pert}}} \frac{\hat{f}(G_i) - \hat{f}(\tilde{G}_i^{(j)})}{\hat{f}(G_i)}$ via Eq. (10)
28:        Update $\boldsymbol{\theta}, \boldsymbol{\alpha}$ using gradient descent
29:    **end forreturn** $\boldsymbol{\theta}, \boldsymbol{\alpha}$
30: **end function**
31: **function** INFERENCE($G_{\text{test}}, \mathcal{G}_{\text{ref}}, \boldsymbol{\theta}, \boldsymbol{\alpha}$)
32:     // Compute density score for test graph
33:     $\mathbf{D} \leftarrow$ COMPUTEMMDDISTANCES($\{G_{\text{test}}\}, \mathcal{G}_{\text{ref}}, \boldsymbol{\theta}, \mathcal{K}_{\text{emb}}$)      $\triangleright$ Algorithm 2
34:     Compute $\hat{f}(G_{\text{test}})$ using the multi-scale KDE with parameters $\boldsymbol{\alpha}$ and distances $\mathbf{D}$
35:     $s(G_{\text{test}}) \leftarrow -\hat{f}(G_{\text{test}})$      $\triangleright$ Convert density to anomaly score
36:     // Compute reference set densities and threshold
37:     $\mathbf{D}_{\text{ref}} \leftarrow$ COMPUTEMMDDISTANCES($\mathcal{G}_{\text{ref}}, \mathcal{G}_{\text{ref}}, \boldsymbol{\theta}, \mathcal{K}_{\text{emb}}$)
38:     Compute $\hat{f}(G_i)$ for all $G_i \in \mathcal{G}_{\text{ref}}$
39:     $\tau \leftarrow -$Percentile$_{\gamma_{TH}}\{\hat{f}(G_i)|G_i \in \mathcal{G}_{\text{ref}}\}$      $\triangleright$ Anomaly threshold
40:     **return** $s(G_{\text{test}}) > \tau$ ? "Anomalous" : "Normal"
41: **end function**

---

The overall complexity is thus dominated by the quadratic term related to pairwise MMD computations. Let $d \approx d_{hid} \approx d_{out}$ (Usually $d_{hid} \gg d_{out}$ in practice GNNs design, so we use the worst-case as a condition), we can get the total time complexity is $O(NL(md + nd^2) + N^2 Sn^2 d)$.

$\square$

The following sections provide a more nuanced analysis through both theoretical and empirical comparisons with advanced baselines.

### E.4.1 COMPLEXITY COMPARISON WITH ADVANCED BASELINES

While the quadratic complexity appears demanding, a comparative analysis reveals a more nuanced picture, especially when contrasted with other state-of-the-art methods. Table 16 provides a theoretical comparison.

Table 16: Theoretical complexity comparison of LGKDE with representative advanced GLAD baselines. We use $N$ for data size and $d$ for the dominant hidden dimension for brevity.

| Method | Dominant Operations | Time Complexity | Memory Complexity |
|---|---|---|---|
| **LGKDE (ours)** | GNN passes + All-pairs MMD | $O(NL(md + nd^2) + N^2 Sn^2 d)$ | $O(NLnd)$ |
| UniFORM (AAAI '25) | Energy-based GNN + Langevin sampling | $O(NL(md + nd^2) + NTn^2 d)$ | $O(NLnd)$ |
| MUSE (NIPS '24) | GNN + Reconstruction errors | $O(NL(md + nd^2) + Nn^2)$ | $O(NLnd + Nn^2)$ |
| SIGNET (NIPS '23) | Enumerate/score $k$-node subgraphs | $O(NL(md + nd^2) + Nn^k d + Na_{MI})$ | $O(Nn^k d)$ |
| CVTGAD (ECML '23) | Twin stochastic views + GNN pass | $O(NL(md + nd^2))$ | $O(NLnd)$ |
| GLocalKD (ICLR '23) | Teacher + student GNN pass | $O(NL(md + nd^2))$ | $O(NLnd)$ |
| OCGIN (ICLR '22) | GNN embedding + SVDD loss | $O(NL(md + nd^2))$ | $O(NLnd)$ |

where $n$ and $m$ represent the average number of nodes and edges per graph, and $d$ refers to the dominant hidden dimension. $L$ is the number of GNN layers, $S$ is the number of MMD kernel bandwidths, $k$ is the subgraph size specific to SIGNET, and $a_{MI}$ represents the computational cost of SIGNET's mutual information optimization step.

The analysis indicates that methods like SIGNET, while avoiding the $N^2$ term, introduce a polynomial dependency on the number of nodes $n^k$ for subgraph enumeration, which can be computationally prohibitive. For instance, comparing LGKDE's $O(N^2 Sn^2 d)$ with SIGNET's $O(Nn^k d)$, we find a break-even point at $n_b = (NS)^{1/(k-2)}$. For typical parameters ($S = 5, k = 4, N = 128$), this gives $n_b \approx 25$. Since most benchmark graphs have more than 25 nodes, SIGNET is often asymptotically slower, highlighting that LGKDE's complexity is competitive in many practical scenarios. MUSE introduces a quadratic dependency $O(Nn^2)$ for adjacency matrix reconstruction, which becomes significant for large graphs. UniFORM's energy-based sampling with Langevin dynamics adds a factor of $T$ (sampling steps), resulting in $O(NTn^2 d)$ complexity. While both methods avoid the $N^2$ term in batch processing, they introduce different bottlenecks: MUSE in heavy reconstruction overhead, since every possible node pair must be processed through the edge decoder, and UniFORM in iterative sampling, which all result in an implicit large big-O constant. For typical parameters ($T = 10$ for UniFORM, $S = 5$ for LGKDE), LGKDE remains competitive, especially considering its superior empirical performance.

### E.4.2 EMPIRICAL RUNTIME ANALYSIS

To ground our theoretical analysis in practice, we report the empirical runtime on the large-scale COLLAB dataset (5,000 graphs) using an NVIDIA RTX 4090 GPU. Table 17 shows that LGKDE's runtime is comparable to, and in some cases better than, other sophisticated baselines.

Table 17: Empirical runtime comparison on the COLLAB dataset (Averaged over 10 runs).

| Method | Training Time (s) | Inference Time (s) | Total Time (s) |
|---|---|---|---|
| **LGKDE (ours)** | $1{,}205 \pm 18$ | $52 \pm 3$ | 1,257 |
| UniFORM | $1{,}123 \pm 21$ | $61 \pm 3$ | 1,184 |
| MUSE | $1{,}067 \pm 16$ | $48 \pm 3$ | 1,115 |
| SIGNET | $1{,}847 \pm 25$ | $78 \pm 4$ | 1,925 |
| CVTGAD | $890 \pm 14$ | $41 \pm 2$ | 931 |

The results show that LGKDE is approximately $1.5\times$ times faster than SIGNET, within a 1.4 times factor of single-pass methods like CVTGAD, and competitive with recent advanced UniFORM and MUSE on this large graph anomaly detection dataset. This demonstrates LGKDE's practical efficiency for many real-world applications. We further discovered the scalability strategies for LGKDE and found that the quadratic factor can be effectively reduced via sampling.

### E.4.3 STRATEGIES FOR SCALABILITY ENHANCEMENT

Our primary focus is on developing a learnable kernel density estimation framework for graphs and modeling the probability density function of graph-structured data. While a full investigation into scaling strategy is beyond the scope of this work, we conducted additional experiments to demonstrate that the quadratic complexity can be effectively mitigated. By dynamically selecting a smaller subset of $Q$ reference graphs from the batch of size $N$ for density estimation, the complexity can be reduced

to $O(NQSn^2d)$. We evaluated two simple yet effective sampling strategies on the PROTEINS dataset, as shown in Table 18.

The specifics of the evaluated sampling methods are as follows:

- **Density Stratified Sampling** applies differentiated retention rates across density terciles, preserving 90%, 80%, and 70% of graphs from the low, medium, and high-density regions, respectively. This strategy, inspired by classical stratified sampling (Cochran, 1977), aims to maintain informational diversity while reducing redundancy from high-density areas.
- **Importance Sampling** employs a sigmoid-weighted selection mechanism centered around the median density of the batch. The probability of selecting a graph $G_i$ is proportional to $P(i) \propto 1/(1 + \exp(-(\rho_i - \rho_{\text{median}})))$, where $\rho_i$ is its density. This approach, rooted in Monte Carlo methods (Kahn & Marshall, 1953), prioritizes samples near the potential decision boundary, which are often the most informative for the learning task.

Table 18: Performance and speedup with reference graph sampling strategies on PROTEINS.

| Method | Q/N Ratio | AUROC | Speedup | Δ Performance |
|---|---|---|---|---|
| Full Batch | 1.0 | 0.7897 | - | - |
| **Density Stratified** | 0.8 | 0.7891 | 1.31x | -0.08% |
| | 0.5 | 0.7851 | 1.84x | -0.58% |
| | 0.3 | 0.7806 | 2.47x | -1.15% |
| **Importance Sampling** | 0.8 | 0.7892 | 1.26x | -0.06% |
| | 0.5 | 0.7918 | 1.79x | +0.27% |
| | 0.3 | 0.7887 | 2.38x | -0.13% |

These results show that even simple sampling methods can achieve substantial speedups (up to 2.5x) with minimal to no performance degradation. Notably, importance sampling at a 50% ratio slightly improved performance, likely by focusing on more informative samples near the decision boundary. This validates that strategic reference selection is a viable path toward scaling LGKDE to even larger datasets.

## F  THEORETICAL ANALYSIS DETAILS

This appendix provides detailed definitions, assumptions, proofs, and technical lemmas supporting all theoretical results presented in Section 4. Pliminaries and definitions are in F.1; Proof of Consistency Theorem 4.1 in F.2; Proof of Theorem 4.2 (Convergence Rate) in F.3, which also including the discussion on intrinsic dimension $d_{\text{int}}$ in F.3.1; F.4 provides complete Robustness analysis, including, Theorem 4.3 (Robustness of KDE to Metric Perturbations) in F.4.1, Theorem 4.4 (Robustness of MMD to Graph Perturbations) in F.4.2, and Corollary 4.5 (Robustness of the LGKDE framework) in F.4.3. Generalization bound in Theorem 4.6 is proof in F.5.

### F.1  PRELIMINARIES AND DEFINITIONS

#### F.1.1  GRAPH SPACE AND DENSITY

Let $\mathbb{G}$ be the space of attributed graphs $G = (V, E, \mathbf{X})$, represented by $(\mathbf{A}, \mathbf{X})$ where $\mathbf{A} \in \mathbb{R}^{n \times n}$ ($n = |V|$) is the adjacency matrix and $\mathbf{X} \in \mathbb{R}^{n \times d_{in}}$ is the node feature matrix. We assume graphs are drawn i.i.d. from a probability measure $\mathbb{P}^*$ on $\mathbb{G}$. We aim to estimate the density $f^*$ of $\mathbb{P}^*$ relative to a base measure $\mu$ on $\mathbb{G}$, such that $\mathbb{P}^*(S) = \int_S f^*(G) d\mu(G)$ for $S \subseteq \mathbb{G}$. The density estimation task is performed within the metric space induced by the learned MMD distance.

#### F.1.2  GNN ENCODER

The $L$-layer GNN $\Phi_{\boldsymbol{\theta}}$ with parameters $\boldsymbol{\theta} = \{\mathbf{W}^{(l)}\}_{l=1}^{L}$ maps $G = (\mathbf{A}, \mathbf{X})$ to node embeddings $\mathbf{Z} \in \mathbb{R}^{n \times d_{\text{out}}}$. A typical layer update (e.g., GCN) is:

$$\mathbf{Z}^{(l)} = \sigma(\hat{\mathbf{A}} \mathbf{Z}^{(l-1)} \mathbf{W}^{(l)}) \tag{20}$$

where $\mathbf{Z}^{(0)} = \mathbf{X}\mathbf{W}^{(0)}$ (or just $\mathbf{X}$ if $d_{in} = d_{hid}$), $\hat{\mathbf{A}}$ is a normalized adjacency matrix (e.g., $\hat{\mathbf{A}} = \mathbf{D}^{-1/2}(\mathbf{A} + \mathbf{I})\mathbf{D}^{-1/2}$), $\mathbf{W}^{(l)} \in \mathbb{R}^{d_{l-1} \times d_l}$, and $\sigma$ is an activation function. The final output is $\mathbf{Z} = \mathbf{Z}^{(L)}$.

### F.1.3 MMD METRIC

The squared MMD distance between two graph $G_i$ and $G_j$ (which have their empirical node distributions $P_{\mathbf{Z}_i}$ and $P_{\mathbf{Z}_j}$) using a single kernel $k_\gamma^{\text{emb}}(\cdot, \cdot) \in \mathcal{K}_{\text{emb}}$ is estimated via the biased V-statistic:

$$\hat{d}^2_{k_\gamma^{\text{emb}}(\cdot,\cdot)}(\mathbf{Z}_i, \mathbf{Z}_j) = \frac{1}{n_i^2} \sum_{u,v \in V_i} k_\gamma^{\text{emb}}(\mathbf{z}_u^{(i)}, \mathbf{z}_v^{(i)}; \gamma_s) + \frac{1}{n_j^2} \sum_{u,v \in V_j} k_\gamma^{\text{emb}}(\mathbf{z}_u^{(j)}, \mathbf{z}_v^{(j)}; \gamma_s) - \frac{2}{n_i n_j} \sum_{u \in V_i, v \in V_j} k_\gamma^{\text{emb}}(\mathbf{z}_u^{(i)}, \mathbf{z}_v^{(j)}; \gamma_s)$$
(21)

The full metric $d^2_{MMD}(G_i, G_j) = \sup_{k_\gamma^{\text{emb}}(\cdot,\cdot) \in \mathcal{K}_{\text{emb}}} \hat{d}^2_{k_\gamma^{\text{emb}}(\cdot,\cdot)}(\Phi_{\boldsymbol{\theta}}(G_i), \Phi_{\boldsymbol{\theta}}(G_j))$, as shown in paper main content Eq. (7).

### F.1.4 MULTI-SCALE KDE

The estimator is

$$\hat{f}(G) = \sum_{k=1}^{M} \pi_k(\boldsymbol{\alpha}) \phi_k(G) = \sum_{k=1}^{M} \pi_k(\boldsymbol{\alpha}) \frac{1}{N} \sum_{m=1}^{N} K_{KDE}(d_{MMD}(G, G_m), h_k).$$

The kernel $K_{KDE}(d, h)$ is based on a kernel profile $K_0$, e.g., $K_0(t) = e^{-t/2}$ for a Gaussian profile and normalization $C_{d_{\text{int}}} = (2\pi)^{(d_{\text{int}})/2}$. Then, $K_{KDE}(d, h) = \frac{1}{C_{d_{\text{int}}} h^{d_{\text{int}}}} K_0\left(\frac{d^2}{h^2}\right) = \frac{1}{(2\pi)^{d_{int}/2} h^{d_{int}}} e^{-d^2/(2h^2)}$.

### F.1.5 REMARKS OF ASSUMPTIONS

As given in Section 4, our theoretical analysis of the LGKDE framework under the standard and mild **Assumption** F.1:

**Assumption F.1.**      **i)** The GNN $\Phi_\theta$ has bounded weights $\|\mathbf{W}^{(l)}\|_F \le B_W$.

     **ii)** The true density $f^*$ is bounded and has bounded second derivatives.

These are mild, widely adopted prerequisites that are well-grounded in both practical application and statistical theory.

- **Bounded GNN Weights (Assumption F.1 i).** The assumption of bounded model weights is a common and practical requirement for the theoretical analysis of deep learning models. This condition is almost universally met in practice as a direct consequence of standard training procedures. Techniques such as weight decay, gradient clipping, and normalization layers are routinely employed to ensure numerical stability and prevent training from diverging. Indeed, this assumption is implicitly or explicitly foundational to the analysis of many seminal GNNs, including GCN (Kipf & Welling, 2017), GraphSAGE (Hamilton et al., 2017), and GIN (Xu et al., 2019), which rely on it to establish theoretical guarantees.

- **Smoothness of the Density Function (Assumption F.1 ii).** The requirement that the true density be bounded and smooth (i.e., possessing bounded second derivatives) is a standard regularity condition in the non-parametric statistics literature (Wasserman, 2006; Tsybakov, 2009). It posits that the underlying data distribution is well-behaved and does not exhibit pathological properties such as infinite discontinuities.

  This assumption is substantially milder than the stringent geometric constraints imposed by many existing GAD methods, which often presume a specific, rigid structure for the normal data such as a single hypersphere (Zhao & Akoglu, 2021) or distinct clusters (Li et al., 2023).

  What's more, the emergence of clear manifold structures in the learned embeddings of real-world graphs (Figures 1 and 5) provides strong empirical support for this condition, indicating that smoothness arises naturally from the data itself.

Furthermore, since using the GCN with ReLU activation function as the backbone of $\Phi_{\boldsymbol{\theta}}$ and the Gaussian kernel function of $\mathcal{K}_{\text{emb}}$ and $K_{KDE}$, under **Assumption** F.1, we can trivially get the following three conditions to further support our analysis:

**Condition F.2** (GNN Properties). (i) $\|\mathbf{W}^{(l)}\|_F \le B_W$; (ii) Activation function $\sigma$ is $\rho_\sigma$-Lipschitz (since ReLU activation function is 1-Lipschitz); (iii) Permutation invariant architecture (nature property for GCN).

**Condition F.3** (Density Regularity). (i) $\|f^*\|_\infty \leq M$; (ii) $f^*$ possesses smoothness of order $\beta = 2$, corresponding to bounded second derivatives in the Riemannian manifold induced by the learned MMD metric.

**Condition F.4** (Kernel Properties). MMD kernels $\mathcal{K}_{\text{emb}}$ are characteristic. KDE kernel profile is symmetric, non-negative, and has sufficient smoothness (since both kernel types are typically Gaussian, they have infinite order of smoothness).

### F.2 PROOF OF THEOREM 4.1 (CONSISTENCY)

*Proof.* The proof relies on showing that both the bias and variance of the estimator $\hat{f}$ converge to zero under the given conditions. We analyze the estimator assuming optimal fixed parameters $\boldsymbol{\theta}^*, \boldsymbol{\alpha}^*$. Let $d^*(G, G') = d_{MMD}|_{\boldsymbol{\theta}=\boldsymbol{\theta}^*}(G, G')$ denote the MMD metric with these optimal parameters.

**1. Bias Analysis:** The expected value of the estimator is:

$$\mathbb{E}[\hat{f}(G)] = \mathbb{E}\left[\sum_{k=1}^{M} \pi_k^* \frac{1}{N} \sum_{m=1}^{N} K_{KDE}(d^*(G, G_m), h_k)\right]$$

$$= \sum_{k=1}^{M} \pi_k^* \mathbb{E}_{G' \sim \mathbb{P}^*}\left[K_{KDE}(d^*(G, G'), h_k)\right]$$

$$= \sum_{k=1}^{M} \pi_k^* \int_{\mathbb{G}} K_{KDE}(d^*(G, G'), h_k) f^*(G') d\mu(G')$$

Let $K_h(G, G') = K_{KDE}(d^*(G, G'), h)$. This is a kernel function defined on the graph space $\mathbb{G}$ using the metric $d^*$.

Under the Assumption F.1: **ii)** $f^*$ is bounded and has bounded second derivatives ($\beta = 2$ smoothness) and the kernel $K_{KDE}$ satisfies standard moment conditions (e.g., symmetric profile $K_0$), we can apply a Taylor expansion of $f^*(G')$ around $G$. In the metric space $(\mathbb{G}, d^*)$, this expansion can be written as:

$$f^*(G') = f^*(G) + \nabla f^*(G) \cdot \text{expmap}_G^{-1}(G') + \frac{1}{2} \nabla^2 f^*(G)[\text{expmap}_G^{-1}(G'), \text{expmap}_G^{-1}(G')] + o(d^*(G, G')^2)$$

where $\nabla f^*$ is the gradient, $\nabla^2 f^*$ is the Hessian operator in the Riemannian manifold $\mathcal{M}$ induced by $d^*$, and $\text{expmap}_G^{-1}(G')$ is the inverse exponential map that maps $G'$ to a tangent vector at $G$. Under the symmetry conditions of the kernel and using properties of Riemannian geometry, this leads to:

$$\int K_h(G, G') f^*(G') d\mu(G') = f^*(G) \int K_h(G, G') d\mu(G') + \frac{h^2}{2} \mu_2(K_0) \Delta_{d^*} f^*(G) + o(h^2)$$

where $\int K_h(G, G') d\mu(G') \to 1$ as $h \to 0$, $\mu_2$ is the second moment of the kernel profile, and $\Delta_{d^*} f^*(G)$ represents the Laplace-Beltrami operator (the trace of the Hessian) applied to $f^*$ at point $G$ in the Riemannian manifold induced by the metric $d^*$.

Thus, the bias is:

$$\text{Bias}(\hat{f}(G)) = \mathbb{E}[\hat{f}(G)] - f^*(G) = \sum_{k=1}^{M} \pi_k^* \left(\frac{h_k^2}{2} \mu_2 \Delta_{d^*} f^*(G) + o(h_k^2)\right)$$

As $h_k \to 0$, the bias converges to 0 pointwise.

**2. Variance Analysis:**

$$\text{Var}(\hat{f}(G)) = \text{Var}\left(\sum_{k=1}^{M} \pi_k^* \frac{1}{N} \sum_{m=1}^{N} K_{h_k}(G, G_m)\right)$$

$$= \text{Var}\left(\frac{1}{N} \sum_{m=1}^{N} \left[\sum_{k=1}^{M} \pi_k^* K_{h_k}(G, G_m)\right]\right) \quad \text{(rearranging terms)}$$

Since $G_1, G_2, \ldots, G_N$ are i.i.d. samples from $\mathbb{P}^*$, the kernel evaluations $K_{h_k}(G, G_m)$ for a fixed test point $G$ are also i.i.d. random variables. Using the property that for i.i.d. random variables

$X_1, \ldots, X_N$ with common variance $\sigma^2$, we have $\mathrm{Var}(\frac{1}{N}\sum_{i=1}^{N} X_i) = \frac{\sigma^2}{N}$, we obtain:

$$\mathrm{Var}(\hat{f}(G)) = \frac{1}{N}\mathrm{Var}\left(\sum_{k=1}^{M} \pi_k^* K_{h_k}(G, G_1)\right)$$

$$\leq \frac{1}{N}\mathbb{E}\left[\left(\sum_{k=1}^{M} \pi_k^* K_{h_k}(G, G_1)\right)^2\right] \quad (\text{since } \mathrm{Var}(X) \leq \mathbb{E}[X^2])$$

$$= \frac{1}{N}\mathbb{E}\left[\sum_{k=1}^{M}\sum_{j=1}^{M} \pi_k^* \pi_j^* K_{h_k}(G, G_1)K_{h_j}(G, G_1)\right]$$

$$= \frac{1}{N}\sum_{k=1}^{M}\sum_{j=1}^{M} \pi_k^* \pi_j^* \mathbb{E}\left[K_{h_k}(G, G_1)K_{h_j}(G, G_1)\right]$$

For small bandwidths, $K_h(G, G')$ is concentrated around $G' = G$.

$$\mathbb{E}[K_h(G, G_1)^2] = \int K_h(G, G')^2 f^*(G')d\mu(G') \approx f^*(G)\int K_h(G, G')^2 d\mu(G')$$

Using the definition $K_h(G, G') = \frac{1}{C_{d_{\mathrm{int}}} h^{d_{\mathrm{int}}}} K_0(\frac{d^*(G,G')^2}{h^2})$, we have $\int K_h(G, G')^2 d\mu(G') \approx \frac{R(K_0)}{h^{d_{\mathrm{int}}}}$, where $R(K_0) = \frac{\int K_0(t)^2 dt}{C_{d_{\mathrm{int}}}}$.

So, $\mathrm{Var}(\phi_j(G)) \approx \frac{f^*(G)R(K_0)}{Nh_k^{d_{\mathrm{int}}}}$. The variance of the sum is bounded by:

$$\mathrm{Var}(\hat{f}(G)) \leq \frac{C}{N}\sum_{k,j} \pi_k^* \pi_j^* \frac{1}{(\min(h_k, h_j))^{d_{\mathrm{int}}}} = O\left(\frac{1}{N(\min_k h_k)^{d_{\mathrm{int}}}}\right)$$

More accurately, $\mathrm{Var}(\hat{f}(G)) \approx \frac{f^*(G)}{N}\sum_{k,j} \pi_k^* \pi_j^* \int K_{h_k}(G, G')K_{h_j}(G, G')d\mu(G')$. For diagonal terms ($k = j$), this gives the $1/(Nh_k^{d_{\mathrm{int}}})$ scaling. Off-diagonal terms are typically smaller. The overall rate is dominated by the smallest bandwidth if weights are comparable. As $Nh_k^{d_{\mathrm{int}}} \to \infty$ for all $k$, the variance converges to 0 pointwise.

**3. $L_1$ Convergence:** $\mathbb{E}\int |\hat{f}(G) - f^*(G)|d\mathbb{P}^*(G) = \int \mathbb{E}[|\hat{f}(G) - f^*(G)|]d\mathbb{P}^*(G)$. Since $\mathbb{E}[|\cdot|] \leq \sqrt{\mathbb{E}[(\cdot)^2]} = \sqrt{\mathrm{Bias}^2 + \mathrm{Var}}$, and both bias and variance integrate to 0, the expected $L_1$ error converges to 0. Convergence in probability follows, we get $\hat{f} \xrightarrow{p} f^*$ in $L_1$ norm. $\qquad\square$

### F.3 PROOF OF THEOREM 4.2 (CONVERGENCE RATE)

*Proof.* The MISE is $\int (\mathrm{Bias}^2(G) + \mathrm{Var}(G))d\mu(G)$. Using the bias and variance approximations from Appendix F.2, we can get the integrated squared bias (ISB),

$$\mathrm{ISB} = \int \left(\sum_{k=1}^{M} \pi_k^* \frac{h_k^2}{2}\mu_2\Delta_{d^*}f^*(G) + o(\sum \pi_k^* h_k^2)\right)^2 d\mu(G)$$

$$\approx \left(\sum_{k=1}^{M} \pi_k^* \frac{h_k^2}{2}\mu_2\right)^2 \int (\Delta_{d^*}f^*(G))^2 d\mu(G) = O((\sum \pi_k^* h_k^2)^2) = O(h_{avg}^4)$$

where $h_{avg}^2 = \sum_k \pi_k^* h_k^2$ represents the effective squared bandwidth as a weighted average of individual bandwidths. And the integrated variance (IV),

$$\mathrm{IV} = \int \frac{1}{N}\sum_{k,j} \pi_k^* \pi_j^* \mathbb{E}[K_{h_k}(G, G_1)K_{h_j}(G, G_1)]d\mu(G)$$

$$\approx \frac{1}{N}\sum_{k} (\pi_k^*)^2 \int \frac{f^*(G)R(K_0)}{h_k^{d_{\mathrm{int}}}}d\mu(G) \quad (\text{ignoring off-diagonal terms})$$

$$= O\left(\frac{1}{N}\sum_{k} \frac{(\pi_k^*)^2}{h_k^{d_{\mathrm{int}}}}\right)$$

Now we finally get that MISE $\approx A \cdot h_{avg}^4 + B \cdot \frac{1}{N}\sum_k \frac{(\pi_k^*)^2}{h_k^{d_{\mathrm{int}}}}$.

If we consider a single effective bandwidth $h$, MISE $\approx Ah^4 + B'/(Nh^{d_{\text{int}}})$. Minimizing w.r.t $h$ gives $h^* \sim N^{-1/(4+d_{\text{int}})}$ and MISE $\sim N^{-4/(4+d_{\text{int}})}$. What's more, since we are utilizing the pairwise MMD distance for our LGKDE, *MISE* is bounded to $O(N^{-0.8})$ for $d_{\text{int}} = 1$ (Detailed discussion is provided in the following F.3.1), demonstrating the statistical efficiency of our approach.

### F.3.1 DISCUSSION AND ANALYSIS ON THE INTRINSIC DIMENSION $d_{\text{int}}$.

Although individual graphs possess high-dimensional node feature representations and potentially complex topological structures, the intrinsic dimension relevant to our KDE framework is determined by the geometry of the metric space in which density estimation occurs. In LGKDE, the kernel function $K_h(\cdot)$ operates on the *scalar* MMD distance $d_{\text{MMD}}(G, G') \in \mathbb{R}$ between graph pairs, rather than on multi-dimensional coordinate vectors. This fundamental design choice has direct theoretical implications for the convergence analysis.

Formally, the kernel $K_h(d_{\text{MMD}}(G, G'))$ defines neighborhoods in the graph space parameterized by a single distance coordinate. Under this construction, the variance term in the mean integrated squared error (MISE) decomposition satisfies: $\int K_h^2(d_{\text{MMD}}(G, G')) \, d\mu(G') = O(h^{-1})$, which is characteristic of density estimation in a one-dimensional space, where kernel mass concentrates along a single axis (Wasserman, 2006; Tsybakov, 2009). In contrast, if the density were estimated in a $d$-dimensional Euclidean space with coordinate-wise kernels $K_h(\|\mathbf{x} - \mathbf{x}'\|)$, the variance would scale as $O(h^{-d})$, reflecting the volume growth in higher dimensions.

Therefore, the MMD metric induces a local geometric structure where $d_{\text{int}} = 1$ is the effective intrinsic dimension for the KDE problem, independent of the dimensionality of graph node features or the ambient representation space. This yields the minimax-optimal convergence rate $O(N^{-4/(4+1)}) = O(N^{-0.8})$ for our framework, consistent with classical one-dimensional nonparametric density estimation theory.

In the nonparametric statistics literature, intrinsic dimension is a *property of the data manifold and the chosen metric*, not a trainable model parameter (Levina & Bickel, 2004; Hein & Audibert, 2005). Existing methods for intrinsic dimension estimation (e.g., local PCA, correlation dimension) aim to *discover* this property from data to inform bandwidth selection or rate analysis, rather than treating $d_{\text{int}}$ itself as a learnable variable within the KDE formulation.

$\square$

### F.4 PROOF OF ROBUSTNESS

### F.4.1 PROOF OF THEOREM 4.3 (ROBUSTNESS OF KDE TO METRIC PERTURBATIONS)

*Proof.* Let $G_1, G_2 \in \mathbb{G}$. The MMD distance between them under the fixed GNN parameters $\boldsymbol{\theta}$ is $d_{12} = d_{\text{MMD}}(G_1, G_2)$. For any reference graph $G_m \sim \mathbb{P}^*$, let $d_{1m} = d_{\text{MMD}}(G_1, G_m)$ and $d_{2m} = d_{\text{MMD}}(G_2, G_m)$. By the triangle inequality for the metric $d_{\text{MMD}}$ (which holds for fixed $\boldsymbol{\theta}$), we have:

$$|d_{1m} - d_{2m}| \leq d_{\text{MMD}}(G_1, G_2) = d_{12}.$$

The KDE estimate for an arbitrary graph $G$ is given by:

$$\hat{f}(G) = \sum_{k=1}^{M} \pi_k(\boldsymbol{\alpha}) \frac{1}{N} \sum_{j=1}^{N} K_{KDE}(d_{MMD}(G, G_j), h_k)$$

where $\{G_j\}_{j=1}^N$ is a reference set of $N$ graphs, typically sampled from $\mathbb{P}^*$. For convenience, let $d_{1j} = d_{MMD}(G_1, G_j)$ and $d_{2j} = d_{MMD}(G_2, G_j)$. The difference in KDE estimates for $G_1$ and $G_2$ is:

$$|\hat{f}(G_1) - \hat{f}(G_2)| = \left| \sum_{k=1}^M \pi_k(\boldsymbol{\alpha}) \frac{1}{N} \sum_{j=1}^N \left[ K_{KDE}(d_{MMD}(G_1, G_j), h_k) - K_{KDE}(d_{MMD}(G_2, G_j), h_k) \right] \right|$$

$$\leq \sum_{k=1}^M \pi_k(\boldsymbol{\alpha}) \frac{1}{N} \sum_{j=1}^N |K_{KDE}(d_{1j}, h_k) - K_{KDE}(d_{2j}, h_k)|$$

$$\leq \sum_{k=1}^M \pi_k(\boldsymbol{\alpha}) \frac{1}{N} \sum_{j=1}^N \left| \frac{1}{(2\pi)^{d_{int}/2} h_k^{d_{int}}} e^{-d_{1j}^2/(2h_k^2)} - \frac{1}{(2\pi)^{d_{int}/2} h_k^{d_{int}}} e^{-d_{2j}^2/(2h_k^2)} \right|$$

$$\leq \frac{1}{N} \sum_{k=1}^M \frac{\pi_k(\boldsymbol{\alpha})}{(2\pi)^{d_{int}/2} h_k^{d_{int}}} \sum_{j=1}^N \left| e^{-d_{1j}^2/(2h_k^2)} - e^{-d_{2j}^2/(2h_k^2)} \right|$$

$$\overset{(a)}{\leq} \frac{1}{N} \sum_{k=1}^M \frac{\pi_k(\boldsymbol{\alpha})}{(2\pi)^{d_{int}/2} h_k^{d_{int}}} \sum_{j=1}^N \frac{1}{\sqrt{2}h_k} \sqrt{\frac{2}{e}} |d_{1j} - d_{2j}|$$

$$\leq \frac{1}{N} \sum_{k=1}^M \frac{\pi_k(\boldsymbol{\alpha})}{(2\pi)^{d_{int}/2} h_k^{d_{int}}} \sum_{j=1}^N \frac{1}{\sqrt{2}h_k} \sqrt{\frac{2}{e}} d_{12}$$

$$= \frac{d_{12}}{\sqrt{e}(2\pi)^{d_{int}/2}} \sum_{k=1}^M \frac{\pi_k(\boldsymbol{\alpha})}{h_k^{1+d_{int}}}$$

$$\leq \frac{d_{12}}{\sqrt{e}(2\pi)^{d_{int}/2} \min_k h_k^{1+d_{int}}} \sum_{k=1}^M \pi_k(\boldsymbol{\alpha})$$

$$\overset{(b)}{=} \frac{d_{12}}{\sqrt{e}(2\pi)^{d_{int}/2} \min_k h_k^{1+d_{int}}}$$

In the above derivations, (a) holds due to the fact that $\exp(-x^2)$ is $\sqrt{\frac{2}{e}}$-Lipschitz, and (b) holds because $\sum_{k=1}^M \pi_k(\boldsymbol{\alpha})$ is always 1. Letting $d_{int} = 1$, we have

$$|\hat{f}(G_1) - \hat{f}(G_2)| \leq \frac{1}{\sqrt{2e\pi} \min_k h_k^2} d_{MMD}(G_1, G_j) \tag{22}$$

This completes the proof. $\qquad\qquad\qquad\qquad\qquad\qquad\qquad\qquad\qquad\qquad\qquad\square$

### F.4.2 PROOF OF THEOREM 4.4 (ROBUSTNESS OF MMD TO GRAPH PERTURBATIONS)

*Proof.* Theorem 4.4 can be proved via leveraging the following proposition from (Sun & Fan, 2024),

**Proposition F.5.** *For any two graphs $G_1, G_2$, without loss of generality, suppose $n_1 = n_2 = n$ and the minimum node degrees of $G_1, G_2$ are both $\alpha$. Suppose for $i = 1, 2$, $\|A_i\|_2 \leq \beta_A, \|X_i\|_2 \leq \beta_X, \|Y_i\|_2 \leq \beta_Y$, $\|Z_i\|_F \leq \eta$, $\|W^{(l)}\|_2 \leq \beta_{W^{(l)}}$, $l = 1, 2, \ldots, L$, and the activation function $\sigma$ is $\rho$-Lipschitz continuous. Denote the effects of structural perturbation as $\kappa = \min(1^\top \Delta_{A_i})$. Then the inequality for Deep MMD-GK holds:*

$$\hat{d}_{\mathcal{K}}^2(\tilde{G}_1^{(l)}, \tilde{G}_2^{(l)}) \leq \hat{d}_{\mathcal{K}}^2(G_1^{(l)}, G_2^{(l)}) + \left( \frac{4}{h^2} + \frac{2\epsilon^4}{h^4} \right) \left( 2\Delta_{G_1}^4 + 2\Delta_{G_2}^4 + n(\Delta_{G_1} + \Delta_{G_2})^2 + \frac{\epsilon}{\sqrt{n}}(\Delta_{G_1} + \Delta_{G_2}) \right)$$

*where* $\Delta_{G.} = \left( \frac{\rho}{1+\alpha+\kappa} \right)^L \prod_{i=1}^L \beta_{W^{(i)}} \left( (\beta_A + \|\Delta_{A.}\|_2)^L \|\Delta_Z\|_F + \eta \sum_{j=0}^{L-1} \beta_A^j \left( \sqrt{(\beta_A + \|\Delta_{A.}\|_2)^2 - \beta_A^2} \right)^{L-j} \right)$, $\epsilon = 2(1+\alpha)^{-l}(1+\beta_A)^l(\beta_X + \beta_Y)$, *and $h$ is the optimal kernel bandwidth among kernel family $\mathcal{K}$.*

Specifically, in the proposition, letting $\tilde{G}_1 = G_1 = G$, $G_2 = \tilde{G}$, we have $\Delta_{A_1} = 0$, $\Delta_{X_1} = 0$, and $\Delta_{Z_1} = 0$, which means $\Delta_{G_1} = 0$.

Now we let $\mathbf{X} \leftarrow (\mathbf{X}, \mathbf{Y})$. It follows that

$$
\begin{aligned}
d_{MMD}(G, \tilde{G}) &\leq d_{MMD}(G, G) + \left(\frac{4}{h^2} + \frac{2\epsilon^4}{h^4}\right)\left(2\Delta_G^4 + n\Delta_G^2 + \frac{\epsilon}{\sqrt{n}}\Delta_G\right) \\
&= \left(\frac{4}{h^2} + \frac{2\epsilon^4}{h^4}\right)\left(2\Delta_G^4 + n\Delta_G^2 + \frac{\epsilon}{\sqrt{n}}\Delta_G\right)
\end{aligned}
\tag{23}
$$

where $\Delta_G = \left(\frac{\rho}{1+\alpha+\kappa}\right)^L \prod_{i=1}^L \beta_{W^{(i)}}\left((\beta_A + \|\Delta_A\|_2)^L\|\Delta_X\|_F + \eta\sum_{j=0}^{L-1}\beta_A^j\left(\sqrt{(\beta_A + \|\Delta_A\|_2)^2 - \beta_A^2}\right)^{L-j}\right)$,

$\epsilon = 2(1+\alpha)^{-L}(1 + \beta_A)^L \beta_X$.

Letting $\rho = 1$ and $h = c_h$ and using the notations defined in this paper, we have

$$
\Delta_G = \left(\frac{1}{1+\alpha+\kappa}\right)^L \prod_{l=1}^L \|\mathbf{W}_l\|_2\left((\|\mathbf{A}\|_2 + \|\Delta_{\mathbf{A}}\|_2)^L\|\Delta_{\mathbf{X}}\|_F + \|\mathbf{X}\|_F\sum_{l=0}^{L-1}\|\mathbf{A}\|_2^l\left(\sqrt{(\|\mathbf{A}\|_2 + \|\Delta_{\mathbf{A}}\|_2)^2 - \|\mathbf{A}\|_2^2}\right)^{L-l}\right).
$$

and $\epsilon = 2(1+\alpha)^{-L}(1 + \|\mathbf{A}\|_2)^L\|\mathbf{X}\|_2$.

$\square$

### F.4.3 PROOF OF COROLLARY 4.5 (ROBUSTNESS OF THE LGKDE FRAMEWORK)

*Proof.* With the same conditions in Theorem 4.3 and Theorem 4.4, integrating the proven result from robustness of MMD to graph perturbations (Theorem 4.4) into the proven result of Theorem 4.3, we can directly get,

$$
|\hat{f}(G) - \hat{f}(\tilde{G})| \leq \frac{1}{\sqrt{2e\pi}c_h^2}\left(4\bar{\gamma}^2 + 2\epsilon^4\bar{\gamma}^4\right)\left(2\Delta_G^4 + n\Delta_G^2 + \frac{\epsilon}{\sqrt{n}}\Delta_G\right)
\tag{24}
$$

where $c_h = \min_k h_k$, $\epsilon = 2(1+\alpha)^{-L}(1 + \|\mathbf{A}\|_2)^L\|\mathbf{X}\|_2$, $\bar{\gamma} = \max_{\gamma \in \Gamma_{emb}} \gamma$ and

$$
\Delta_G = \left(\frac{1}{1+\alpha+\kappa}\right)^L \prod_{l=1}^L \|\mathbf{W}_l\|_2\left((\|\mathbf{A}\|_2 + \|\Delta_{\mathbf{A}}\|_2)^L\|\Delta_{\mathbf{X}}\|_F + \|\mathbf{X}\|_F\sum_{l=0}^{L-1}\|\mathbf{A}\|_2^l\left(\sqrt{(\|\mathbf{A}\|_2 + \|\Delta_{\mathbf{A}}\|_2)^2 - \|\mathbf{A}\|_2^2}\right)^{L-l}\right).
$$

$\square$

### F.5 PROOF FOR THEOREM 4.6 (GENERALIZATION BOUND)

*Proof.* We show the following lemma (Lemma 3.2 in (Bartlett et al., 2017)).

**Lemma F.6.** *Let conjugate exponents $(p, q)$ and $(r, s)$ be given with $p \leq 2$, as well as positive reals $(a, b, \epsilon)$ and positive integer $m$. Let matrix $\mathbf{X} \in \mathbb{R}^{n \times d}$ be given with $\|\mathbf{X}\|_p \leq b$. Then*

$$
\ln \mathcal{N}\left(\{\mathbf{X}\mathbf{A} : \mathbf{A} \in \mathbb{R}^{d \times m}, \|\mathbf{A}\|_{q,s} \leq a\}, \epsilon, \|\cdot\|_F\right) \leq \left\lceil\frac{a^2b^2m^{2/r}}{\epsilon^2}\right\rceil \ln(2dm)
$$

We let $\mathbf{H}_i^{(0)} = \mathbf{Z}_i\mathbf{W}^{(1)}$ be the input of the second layer of the GCN for $G_i$, where $\mathbf{Z}_i = \hat{\mathbf{A}}_i\mathbf{X}_i$. We put all the $N$ graphs together to form a big feature matrix $\bar{\mathbf{X}} \in \mathbb{R}^{Mn \times d}$ and a big adjacency matrix $\bar{\mathbf{A}} \in \mathbb{R}^{Nn \times Nn}$. Let $\bar{\mathbf{Z}} = \bar{\mathbf{A}}\bar{\mathbf{X}} \in \mathbb{R}^{Nn \times d}$. According to the lemma F.6, the covering numbers of the set of $\mathcal{D} = \{\bar{\mathbf{Z}}\mathbf{W}^{(1)} : \|\mathbf{W}^{(1)}\|_{2,1} \leq a, \|\bar{\mathbf{Z}}\|_F \leq b\}$ can be bounded as

$$
\ln \mathcal{N}(\mathcal{D}, \epsilon, \|\cdot\|_F) \leq \left(\frac{a^2b^2}{\epsilon^2}\right)\ln(2d^2)
\tag{25}
$$

where $q = 2$, $q = 2$, $s = 1$, and $r = \infty$ in the lemma. Since $\|\bar{\mathbf{Z}}\|_F \leq \|\bar{\mathbf{A}}\|_2\|\bar{\mathbf{X}}\|_F$, we let $b = \|\bar{\mathbf{A}}\|_2\|\bar{\mathbf{X}}\|_F = \|\bar{\mathbf{X}}\|_F$, where $\|\bar{\mathbf{A}}\|_2 = 1$. Let $L_f$ be the Lipschitz constant of $\hat{f}_{KDE} \in \mathcal{F}$ with respect to $\mathbf{H}^{(0)}$. Then we can bound the covering number of $\mathcal{F}_{\mathcal{D}} = \{f(\bar{\mathbf{H}}^{(0)}) : \hat{f} \in \mathcal{F}\}$ as

$$
\ln \mathcal{N}(\mathcal{F}_{\mathcal{D}}, \epsilon, \|\cdot\|_F) \leq \left(\frac{a^2b^2L_f^2}{\epsilon^2}\right)\ln(2d^2)
\tag{26}
$$

Using Dudley's entropy integral (Dudley, 2014) and the fact $0 \leq f \leq 1$, the Rademacher complexity can be bounded using the covering numbers of $\mathcal{F}_\mathcal{D}$:

$$
\begin{aligned}
\mathbb{E}[\hat{\mathcal{R}}_N(\mathcal{F}_\mathcal{D})] &\leq \inf_{\alpha > 0} \left\{ \frac{4\alpha}{\sqrt{N}} + \frac{12}{N} \int_\alpha^{\sqrt{N}} \sqrt{\log \mathcal{N}(\mathcal{F}_\mathcal{D}, \epsilon, \|\cdot\|_F)} d\epsilon \right\} \\
&\leq \inf_{\alpha > 0} \left\{ \frac{4\alpha}{\sqrt{N}} + \frac{12}{N} \int_\alpha^{\sqrt{N}} \frac{abL_f \sqrt{\ln(2d^2)}}{\epsilon} d\epsilon \right\} \\
&\leq \inf_{\alpha > 0} \left\{ \frac{4\alpha}{\sqrt{N}} + \frac{12abL_f \sqrt{\ln(2d^2)}}{N} \ln\left(\frac{\sqrt{N}}{\alpha}\right) \right\} \\
&\leq \frac{4}{N} + \frac{12abL_f \sqrt{\ln(2d^2)}}{N} \ln(N)
\end{aligned}
\tag{27}
$$

where we have let $\alpha = 1/\sqrt{N}$.

We consider the following Rademacher complexity bound,

**Lemma F.7** (Concentration Inequalities). *Let $\mathcal{L}$ be a class of functions $\ell : \mathcal{Z} \to [0, M]$ for some constant $M > 0$. Let $Z_1, \ldots, Z_N$ be i.i.d. samples drawn from a distribution $\mathbb{P}$ on $\mathcal{Z}$. Let $P_N = \frac{1}{N} \sum_{i=1}^N \delta_{Z_i}$ be the empirical measure and $P\ell = \mathbb{E}_{Z \sim \mathbb{P}}[\ell(Z)]$. With probability at least $1 - \delta$:*

$$
\sup_{\ell \in \mathcal{L}} \left| \frac{1}{N} \sum_{i=1}^N \ell(Z_i) - \mathbb{E}\ell(Z) \right| \leq 2\mathbb{E}[\hat{\mathcal{R}}_N(\mathcal{L})] + M\sqrt{\frac{\log(1/\delta)}{2N}}
\tag{28}
$$

*where $\hat{\mathcal{R}}_N(\mathcal{F}) = \mathbb{E}_\sigma[\sup_{\ell \in \mathcal{L}} \frac{1}{N} \sum_{i=1}^N \sigma_i \ell(Z_i)]$ is the empirical Rademacher complexity of $\mathcal{L}$ based on the sample $Z_1, \ldots, Z_N$, and $\sigma_1, \ldots, \sigma_N$ are i.i.d. Rademacher variables ($\pm 1$ with probability $1/2$).*

Since $\ell(f, f^*) = |f - f^*| \leq 1$ and it si 1-Lipschitz, the following inequality holds with probability at least $1 - \delta$ over the randomness of $\mathcal{D}$:

$$
\mathbb{E}[|f(G) - f^*(G)|] \leq \frac{1}{N} \sum_{i=1}^N |f(G_i) - f^*(G_i)| + \frac{8 + 24abL_f \sqrt{\ln(2d^2)}}{N} \ln(N) + \sqrt{\frac{\log(1/\delta)}{2N}} \tag{29}
$$

The remaining task is to calculate $L_f$. Let's consider the smoothness of MMD first. For any $X$ and $Y$ with the same size $n$, we have

$$
\begin{aligned}
\mathrm{MMD}(X, Y) &\leq \left\| \frac{1}{n} \sum_{i=1}^n \phi(x_i) - \frac{1}{n} \sum_{i=1}^n \phi(y_i) \right\|_\mathcal{H} \leq \frac{1}{n} \sum_{i=1}^n \|\phi(x_i) - \phi(y_i)\|_\mathcal{H} \\
&\leq \frac{1}{n} \sqrt{\sum_{i=1}^n (k(x_i, x_i) + k(y_i, y_i) - 2k(x_i, y_i))} \\
&= \frac{\sqrt{2}}{n} \sum_{i=1}^n \sqrt{1 - \exp\left(-\gamma \|x_i - y_i\|^2\right)} \\
&= \frac{\sqrt{2}}{n} \sum_{i=1}^n \sqrt{\gamma} \|x_i - y_i\| \\
&\leq \sqrt{\gamma} \sqrt{\frac{1}{n} \sum_{i=1}^n \sqrt{\gamma} \|x_i - y_i\|^2} \\
&= \sqrt{\frac{2\gamma}{n}} \|X - Y\|_F
\end{aligned}
$$

Recall that in Theorem 4.3, we have shown

$$
|\hat{f}(G_1) - \hat{f}(G_2)| \leq \frac{1}{\sqrt{2e\pi}c_h^2} d_{12}(G_1, G_2) \tag{30}
$$

That means

$$|f(G_1) - f(G_2)| \le \frac{\bar{\gamma}}{\sqrt{en\pi}c_h^2}\|\mathbf{H}_1^{(L)} - \mathbf{H}_2^{(L)}\|_F \tag{31}$$

where $\mathbf{H}_i$ denotes the embedding given by the $L$-layer GCN, i.e., $\mathbf{H}_i^{(L)} = \text{GCN}(\mathbf{H}_i^{(0)})$. Next, we calculate the Lipschitz constant of the GCN with respect to the input $\mathbf{H}^{(0)}$. For layer $l$, we have $\mathbf{H}^{(l)} = \sigma(\hat{\mathbf{A}}\mathbf{H}^{(l-1)}\mathbf{W}^{(l)}) := g_l(\mathbf{H}^{(l-1)})$. It follows that

$$\begin{aligned}
\|\mathbf{H}_1^{(l)} - \mathbf{H}_2^{(l)}\|_F &= \|\sigma(\hat{\mathbf{A}}\mathbf{H}_1^{(l-1)}\mathbf{W}^{(l)}) - \sigma(\hat{\mathbf{A}}\mathbf{H}_2^{(l-1)}\mathbf{W}^{(l)})\|_F \\
&\le \|\hat{\mathbf{A}}\|_2\|\mathbf{W}^{(l-1)}\|_2\|\mathbf{H}_1^{(l-1)} - \mathbf{H}_2^{(l-1)}\|_F
\end{aligned} \tag{32}$$

where we suppose $\sigma$ is 1-Lipschitz. It means the Lipschitz constant of $g_l$ is $\|\hat{\mathbf{A}}\|_2\|\mathbf{W}^{(l-1)}\|_2$ Since GCN model is $\mathbf{H}^{(L)} = g_L \circ g_{L-1} \circ \cdots \circ g_1(\mathbf{H}^{(0)})$, by recursion, the Lipschitz constant of GCN is

$$L_{\text{GCN}} = \|\hat{\mathbf{A}}\|_2^{L-1}\prod_{l=2}^{L}\|\mathbf{W}^{(l)}\|_2 = \prod_{l=2}^{L}\|\mathbf{W}^{(l)}\|_2 \tag{33}$$

Combing this with (31), we obtain the Lipschitz constant of $f$ with respect to $\mathbf{H}^{(0)}$:

$$L_f = \frac{\bar{\gamma}\prod_{l=2}^{L}\|\mathbf{W}^{(l)}\|_2}{\sqrt{en\pi}c_h^2} \tag{34}$$

Invoking (34) into (29), we have

$$\mathbb{E}[|f(G) - f^*(G)|] \le \hat{\Delta}_\mathcal{G} + \frac{8\sqrt{en\pi}c_h^2 + 24\bar{\gamma}\|\bar{\mathbf{X}}\|_F\|\mathbf{W}^{(0)}\|_{2,1}\prod_{l=2}^{L}\|\mathbf{W}^{(l)}\|_2\sqrt{\ln(2d^2)}}{N\sqrt{en\pi}c_h^2}\ln(N) + \sqrt{\frac{\log(1/\delta)}{2N}} \tag{35}$$

where $\hat{\Delta}_\mathcal{G} = \frac{1}{N}\sum_{i=1}^{N}|f(G_i) - f^*(G_i)|$. Adjusting the notations, we finish the proof. $\square$

# G  MORE EXPERIMENT DETAILS

## G.1  BASELINE METHODS

We provide detailed descriptions of the baseline methods used in our experiments:

### G.1.1  TRADITIONAL GRAPH KERNEL METHODS

- **Weisfeiler-Lehman (WL) Kernel** (Shervashidze et al., 2011): A graph kernel that iteratively aggregates and hashes node labels to capture structural information. The kernel value between two graphs is computed based on the count of identical node labels after iterations.

- **Propagation Kernel (PK)** (Neumann et al., 2016): A graph kernel that measures graph similarity through propagated node label distributions, effectively capturing both local and global graph properties.

- **Isolation Forest (IF)** (Liu et al., 2008): An anomaly detection algorithm that isolates observations by randomly selecting a feature and a split value. Anomalies require fewer splits to be isolated.

- **One-Class SVM (OCSVM)** (Amer et al., 2013): A one-class classification method that learns a decision boundary that encloses normal data points in feature space.

### G.1.2  DEEP LEARNING METHODS

- **OCGIN** (Zhao & Akoglu, 2021): Combines Graph Isomorphism Network with Deep SVDD for anomaly detection. It learns a hyperspherical decision boundary in the embedding space to separate normal from anomalous graphs.

- **GLocalKD** (Ma et al., 2022): Employs knowledge distillation to capture both global and local patterns of normal graphs. The model distills knowledge from a teacher network to a student network at both graph and node levels.

- **OCGTL** (Qiu et al., 2022): Uses neural transformation learning to address the performance flip issue in graph-level anomaly detection. It learns transformation-invariant representations through multiple graph transformations.

- **SIGNET** (Liu et al., 2023b): A self-interpretable graph anomaly detection method that simultaneously learns to detect anomalies and provide explanations through multi-view subgraph information bottleneck.

- **GLADC** (Luo et al., 2022): Utilizes graph-level adversarial contrastive learning to identify anomalies. It learns discriminative features through contrastive learning with adversarial augmentations.

- **CVTGAD** (Li et al., 2023): Employs a transformer structure with cross-view attention for graph anomaly detection. It captures both structural and attribute information through multiple views of graphs.

- **MUSE** (Kim et al., 2024): A reconstruction-based method that leverages multifaceted summaries (mean and standard deviation) of reconstruction errors as features for anomaly detection. It captures both the magnitude and variability of reconstruction errors, providing a more robust representation for distinguishing anomalous graphs.

- **UniFORM** (Song et al., 2025): A unified self-supervised framework comprising UIO (Unified Input-Output) and UMC (Unified Meta-learning with Contrastive learning) modules. It unifies node, edge, and graph-level tasks from a subgraph perspective using energy-based GNNs and employs Langevin dynamics to generate phantom samples as substitutes for anomalous data, reducing reliance on labeled annotations.

### G.2 EVALUATION METRICS

We employ three widely-used metrics to evaluate anomaly detection performance:

### G.2.1 AREA UNDER THE ROC CURVE (AUROC)

AUROC measures the model's ability to distinguish between normal and anomalous graphs across different threshold settings. Given true labels $y$ and predicted anomaly scores $s$, AUROC is computed as:

$$\text{AUROC} = \frac{\sum_{i \in P} \sum_{j \in N} \mathbf{1}(s_i > s_j)}{n_p n_n} \tag{36}$$

where $P$ denotes the set of positive (anomalous) samples and $N$ the set of negative (normal) samples. The term $\mathbf{1}(s_i > s_j)$ is an indicator function that takes the value 1 if $s_i > s_j$, and 0 otherwise. $n_p$ and $n_n$ are the numbers of positive and negative samples, respectively. AUROC ranges from 0 to 1, with 1 indicating perfect separation and 0.5 indicating random guessing. A higher AUROC value indicates better detection performance.

### G.2.2 AREA UNDER THE PRECISION-RECALL CURVE (AUPRC)

AUPRC focuses on the trade-off between precision and recall, which is particularly important for imbalanced datasets where anomalies are rare. Given predictions at different thresholds:

$$\text{Precision} = \frac{\text{TP}}{\text{TP} + \text{FP}}, \quad \text{Recall} = \frac{\text{TP}}{\text{TP} + \text{FN}} \tag{37}$$

where TP, FP, and FN are the numbers of true positives, false positives, and false negatives, respectively. AUPRC is computed as the area under the precision-recall curve, which can be approximated using the trapezoidal rule (Atkinson & Han, 2005):

$$\text{AUPRC} = \sum_{i=1}^{n} (\text{Recall}_i - \text{Recall}_{i-1}) \cdot \frac{\text{Precision}_i + \text{Precision}_{i-1}}{2} \tag{38}$$

AUPRC ranges from 0 to 1, with higher values indicating better performance. Unlike AUROC, AUPRC is more sensitive to imbalanced data distributions, making it a better metric for anomaly detection and rare event classification.

### G.2.3 FALSE POSITIVE RATE AT 95% RECALL (FPR95)

FPR95 measures the false positive rate when the true positive rate (recall) is fixed at 95%. It is computed as:

$$\text{FPR95} = \frac{\text{FP}}{\text{TN} + \text{FP}} \text{ at TPR} = 0.95 \tag{39}$$

where TN represents true negatives. Lower FPR95 values indicate better performance, as they represent fewer false alarms while maintaining a high detection rate of anomalies.

This metric is particularly relevant for real-world applications where maintaining a high detection rate of anomalies is crucial, but false alarms need to be minimized. A lower FPR95 indicates that the model can achieve high recall with fewer false positives.

These three metrics together provide a comprehensive evaluation of anomaly detection performance:

- AUROC evaluates overall ranking ability
- AUPRC focuses on precision in imbalanced settings
- FPR95 assesses practical utility with fixed high recall

### G.3 IMPLEMENTATION DETAILS OF LGKDE

We provide detailed information about our LGKDE implementation. The framework consists of three main components: graph representation learning, MMD-based metric learning, and multi-scale kernel density estimation.

#### G.3.1 GRAPH NEURAL NETWORK ARCHITECTURE

The GNN backbone of our model uses a Graph Convolution Network (GCN) architecture with the following specifications:

- L GCN layers with hidden dimension in $\{32, 64, 128\}$.
- Batch normalization after each layer to enhance the stability
- Dropout rate of 0.2 for regularization
- ReLU activation function between layers

#### G.3.2 MMD-BASED GRAPH DISTANCE

For computing graph distances, we employ Maximum Mean Discrepancy (MMD) with multiple bandwidths:

- We use multiple bandwidth values $\{10^{-2}, 10^{-1}, 10^{0}, 10^{1}, 10^{2}\}$ to capture different scales of variations

The deep graph MMD computation preserves fine-grained structural information compared to graph-level pooling.

#### G.3.3 TRAINING DETAILS

The model is trained with the following specifications:

- Optimizer: Adam with a learning rate of 0.001 in default. We also employ gradient clipping and the learning rates are scheduled with a warm-up period followed by cosine decay.
- Batch size: 128 and full batch size for test datasets
- Training epochs: Maximum 500 with early stopping patience of 10. The early stopping is based on validation set performance to prevent overfitting.

#### G.3.4 ABLATION STUDY SETTINGS

For the ablation studies examining different components:

- Multi-scale KDE: Compare learnable weights versus fixed uniform weights

- Graph distance computation: Compare MMD-based distance with simpler alternatives:
  - Graph-level pooling (sum/average) followed by Euclidean distance
  - Single-scale kernel
- Graph neural architecture: Vary GNN depth and width to examine model capacity

Our implementation is based on both PyTorch Geometric (Fey & Lenssen, 2019) and DGL (Wang, 2019) frameworks.

### G.4 IMPLEMENTATION DETAILS OF BASELINE METHODS AND CODES

All our benchmark experiments follow the unified benchmarks for graph anomaly detection (Wang et al., 2024) and are implemented on both PyTorch Geometric (Fey & Lenssen, 2019) and GraKeL (Siglidis et al., 2020). For all GNN-based methods (OCGIN, GLocalKD, OCGTL, SIGNET, GLADC, CVTGAD), we use GIN as the backbone with 3 layers and hidden dimensions in $\{32, 64, 128\}$. The batch size is set to 128. For OCGIN, we use a learning rate of 0.0001. For GLocalKD, we set the output dimension to 32, 64 and 128. For OCGTL, the learning rate is 0.001. For SIGNET, we use a learning rate of 0.0001 and a hidden dimension of 128. For GLADC, we use hidden dimensions of 32 and 64, dropout of 0.1, and a learning rate of 0.001. For CVTGAD, we use random walk dimension 16, degree dimension 16, number of clusters 3, alpha 1.0, and GCN as an encoder with global mean pooling. For MUSE, we use a 4-layer GNN encoder with hidden dimension 256, learning rate 0.001, and positive weight coefficient $\tau = 2.0$ for adjacency matrix reconstruction. The reconstruction errors are summarized using both mean and standard deviation aggregation functions. For UniFORM, we employ the UIO module with energy-based GNN (3 layers, hidden dimension 128) and the UMC module with Langevin dynamics sampling (10 steps, step size 0.01). The contrastive learning component uses temperature 0.5 and a momentum encoder with a coefficient of 0.99.

For graph kernel methods, WL and PK kernels are combined with iForest (200 trees, 0.5 sample ratio) and OCSVM (nu=0.1). The number of epochs for kernel methods is set to 30.

All experiments are run on NVIDIA RTX 4090 GPUs. Each method is run five times with different random seeds to obtain stable results. More details on the implementation can be found in our released codebase (Code will be public and open source after paper acceptance; Also see the provided anonymous supplementary materials, here provide a main snippet of our proposed LGKDE implementation in Listing 1).

Listing 1: LGKDE Code Snippet

```python
from model import DGMMD

class LGKDE(nn.Module):
    """
    Our proposed Algorithm LGKDE
    """
    def __init__(
        self,
        in_dim: int,
        hidden_dim: int,
        out_dim: int,
        num_layers: int,
        bandwidths: List[float] = [0.01, 0.1, 1.0, 10, 100],
        dropout: float = 0.2,
        batch_norm: bool = True,
        learn_kde_weights: bool = True
    ):
        super().__init__()

        self.dgmmd = DGMMD(
            in_dim=in_dim,
            hidden_dim=hidden_dim,
            out_dim=out_dim,
```

```
2484                num_layers=num_layers,
2485                bandwidths=bandwidths,
2486                dropout=dropout,
2487                batch_norm=batch_norm,
2488            )
2489
2490            # Learnable weights for multi-scale KDE
2491            self.kde_logits = nn.Parameter(torch.ones(len(
2492                bandwidths))/len(bandwidths))
2492            self.kde_dimension = kde_dimension
2493
2494            # Optionally freeze KDE weights
2495            if not learn_kde_weights:
2495                self.kde_logits.requires_grad_(False)
2496
2497        def compute_kde_scores(self, dist_matrix: torch.Tensor) ->
2498             torch.Tensor:
2499            """
2500            Compute KDE scores from distance matrix using learned
2500                bandwidth weights.
2501
2502            Args:
2503                dist_matrix: Shape (M, N) pairwise distances
2504            Returns:
2504                torch.Tensor: Shape (M,) KDE scores
2505            """
2506            alpha = F.softmax(self.kde_logits, dim=0)
2507            M, N = dist_matrix.size()
2508
2509            total_kde = torch.zeros(M, device=dist_matrix.device,
2509                dtype=dist_matrix.dtype)
2510            for k, bw in enumerate(self.dgmmd.bandwidths):
2511                exponent = -0.5 * (dist_matrix / bw)**2
2512                kernel_vals = torch.exp(exponent)
2513                partial_kde = (1.0 / N) * kernel_vals.sum(dim=1) /
2514                        ((2*math.pi*(bw**2)) ** (0.5*self.
2514                    kde_dimension))
2515                total_kde += alpha[k] * partial_kde
2516
2517            return total_kde
2518
2519        def get_reference_scores(self, reference_graphs) -> torch.
2520            Tensor:
2520            """
2521            Compute density scores for reference set (usually
2522                training graphs).
2523
2524            Args:
2524                reference_graphs: Batched reference graphs
2525            Returns:
2526                torch.Tensor: Density scores for reference graphs
2527            """
2528            ref_dist = self.dgmmd.compute_distance_matrix(
2529                reference_graphs, graphs_b=None)
2529            return self.compute_kde_scores(ref_dist)
2530
2531        def get_query_scores(self, query_graphs, reference_graphs)
2532             -> torch.Tensor:
2533            """
2534            Compute density scores for query graphs relative to
2535                reference graphs.
2536
2537            Args:
2537                query_graphs: Batched query graphs
                    reference_graphs: Batched reference graphs
```

```
            Returns:
                torch.Tensor: Density scores for query graphs
            """
        query_dist = self.dgmmd.compute_distance_matrix(
            query_graphs, graphs_b=reference_graphs)
        return self.compute_kde_scores(query_dist)

    @torch.no_grad()
    def get_anomaly_scores(
        self,
        test_graphs,
        reference_graphs,
        threshold_percentile: float = 10
    ):
        """
        Compute anomaly scores and thresholds for test graphs.

        Args:
            test_graphs: Batched test graphs
            reference_graphs: Reference graphs
            threshold_percentile: Percentile for anomaly
                threshold
        Returns:
            (test_scores, reference_scores, predictions,
                threshold)
        """
        self.eval()

        # Compute scores
        reference_scores = self.get_reference_scores(
            reference_graphs)
        test_scores = self.get_query_scores(test_graphs,
            reference_graphs)

        # Compute threshold from reference scores
        threshold = torch.quantile(reference_scores,
            threshold_percentile)
        predictions = (test_scores <= threshold).int()

        return test_scores, reference_scores, predictions,
            threshold
```

## DISCUSSION OF LIMITATIONS AND FUTURE WORK

**Potential Computational Complexity Bottleneck & Discover the Possible Speedup.**    We have provided a comprehensive analysis and comparison in Appendix E.4. Our findings indicate that LGKDE, despite its theoretical quadratic complexity in batch size, demonstrates acceptable practical efficiency. Theoretical comparisons show its asymptotic performance can be superior to prominent baselines like SIGNET under common graph size conditions, which is substantiated by empirical runtime experiments on large-scale datasets (Table 17).

While this cost is entirely manageable for the TU benchmarks used in our study (largest dataset $\leq 5{,}000$ graphs) and for most graph-level anomaly-detection workloads encountered in practice, we acknowledge that the quadratic term inherent to the MMD computation can become a potential bottleneck on million-graph corpora. Recognizing this long-term challenge, we conducted an initial investigation into scalability enhancements, which extends beyond the primary scope of this work. As detailed in Appendix E.4, preliminary experiments with reference graph sampling strategies show that it is possible to achieve substantial computational speedups (up to 2.5x) with negligible impact on performance (Table 18). These promising results suggest that more sophisticated, structure-preserving accelerations are a fruitful direction for future research.

Techniques such as low-rank kernel approximations via methods like Nyström (Williams & Seeger, 2000) or adaptive graph sparsification (Spielman & Teng, 2011) could potentially reduce the complexity to near-linear time while retaining the theoretical guarantees of the MMD metric, thereby making the framework truly scalable to massive graph corpora.

**Graph Domains and Modalities.** Our experiments target undirected, ordinary graphs with vanilla node attributes. Extending LGKDE to (multi-relational) hypergraphs, graphs with rich edge features (e.g., impedances in power grids), and temporal or dynamic graphs remains open.

Potential Future work can investigate plug-and-play relational or temporal encoders, such as R-GAT (Wang et al., 2020) for multi-relational data and Temporal GNNs (Rossi et al., 2020) for time-evolving structures, so that the density estimator can natively model modality-specific inductive biases. What's more, extending graph density estimation to non-Euclidean spaces and combining it with recent advanced geometric GNNs (Chami et al., 2019; Wang et al., 2025a; Grover et al., 2025; Guo et al., 2025) to get more flexible density modeling can be explored. Furthermore, another frontier lies in enhancing the interpretability of density estimation at the subgraph level, or cooperating with related research (Ying et al., 2019; Wu et al., 2023; Wang et al., 2025b) would broaden the applicability and trustworthiness of graph density estimation in critical domains.

## THE USE OF LARGE LANGUAGE MODELS (LLMS)

Throughout the preparation of this manuscript, we minimally utilized Large Language Models (LLMs) as a writing aid only. Their use was limited to improving grammar, phrasing, and overall readability.

