# OpenReview forum: "Learnable Kernel Density Estimation for Graphs"
_ICLR.cc/2026/Conference — Submitted to ICLR 2026_

### Official Review · Reviewer_xSCo · 2025-10-31

**Soundness:** 2
**Presentation:** 3
**Contribution:** 2
**Rating:** 4
**Confidence:** 4

**Summary:**

This paper proposes LGKDE, a learnable kernel density estimation framework for graphs that integrates GNN-based representations, a deep MMD metric, and a multi-scale KDE with learnable mixture weights. The method contrasts normal graphs with their structure-aware perturbed counterparts to learn a density function capable of modeling complex graph distributions. The authors provide theoretical guarantees on consistency, convergence, and robustness, and demonstrate empirical superiority on twelve benchmark graph anomaly detection datasets. The study aims to unify deep representation learning and nonparametric density estimation within a principled framework.

**Strengths:**

1. Clear problem setting and motivation. The paper tackles a fundamental yet underexplored problem of graph density estimation, framing it in a principled manner that naturally connects to graph-level anomaly detection.

2. Integration of theoretical and empirical perspectives. The authors not only design a learnable KDE model but also provide non-trivial theoretical analyses (consistency, convergence rate, and robustness bounds), lending rigor to the framework.


3. Conceptually coherent design. The use of a deep MMD metric space and structure-aware perturbations offers a reasonable and interpretable approach to contrastive density learning, aligning well with the paper’s stated objectives.

**Weaknesses:**

1. Motivation and Empirical Support

The paper claims that existing graph learning methods trained with standard supervised objectives tend to capture spurious signals, making them fragile under distributional shifts. However, this statement lacks systematic evidence or quantitative validation. The only supporting illustration is a qualitative t-SNE visualization, which is insufficient to substantiate the general claim. A dedicated experiment or section demonstrating the degradation of standard methods under controlled distributional shifts or noise perturbations would make the motivation more convincing.

2. Methodological Assumptions and Theoretical Validity

   The theoretical derivation of LGKDE relies on a strong assumption that the intrinsic dimensionality ( d_{\text{int}} = 1 ), simplifying the density estimation formulation and constants. Furthermore, the convergence and robustness proofs implicitly assume that the graph space admits a smooth Riemannian structure that allows Taylor expansion, which may not hold for heterogeneous or discrete graph distributions. These assumptions weaken the generality of the claimed theoretical guarantees.

3. Incomplete Baseline Coverage

   Although the paper compares LGKDE with various graph anomaly detection (GAD) models and a two-stage GAE+KDE pipeline, it omits several directly related density-based approaches. In particular, there is no comparison with Sun & Fan (2024), which also employs an MMD-based metric learning framework, or with graph normalizing flow and energy-based models that perform explicit density estimation. Including these baselines would clarify whether LGKDE provides genuine methodological advancement beyond existing learnable density estimators.

4. Insufficient Ablation on Perturbation Module

   The proposed structure-aware perturbation mechanism is central to the model’s objective, yet its role is not rigorously dissected. The ablation studies only vary the number of KDE bandwidths and mixture weights, but do not test variants using (a) feature-swap only, (b) spectral perturbation only, or (c) random edge perturbation. Since the perturbation strategy directly drives the density contrastive learning, such ablations are crucial to justify its necessity and effectiveness.

**Questions:**

1. Could authors provide quantitative experiments showing how existing baselines (e.g., OCGIN, SIGNET) fail under controlled distribution shifts, to empirically support your motivation?
2. How realistic is the assumption ( d_{\text{int}} = 1 )? Have authors tested whether relaxing this assumption (e.g., learned or dataset-dependent intrinsic dimension) affects performance or theoretical consistency?
3. Why were Sun & Fan (2024) or other explicit density estimation methods (graph flows, EBMs) excluded from the comparison? Are there implementation or conceptual reasons?
4. Can authors report additional ablations on the perturbation module—particularly feature-only, spectral-only, and random perturbations—to show which component contributes most to the performance gain?

---

> ### Author Response · Authors · 2025-11-21
> **Rebuttal Part I**
>
> **We sincerely thank you for your thoughtful critique and review. We address your questions and each concern systematically below.**
>
> ---
> ### **Question 1**
>
> **quote raw question:** *“Could authors provide quantitative experiments showing how existing baselines (e.g., OCGIN, SIGNET) fail under controlled distribution shifts, to empirically support your motivation?”*
>
> **Response.**
>
> Thank you for this question.
>
> **Additional Experiments:** We have conducted controlled distribution shift experiments and added them to the revised paper in Appendix B.6.
>
>
> We use Erdős–Rényi graphs with edge probability $p \sim \mathcal{N}(0.5, 0.1)$ (normal class), then evaluated under three scenarios:
>
> 1. **In-Distribution (ID):** Test on the same distribution, anomalies have extreme edge densities ($p \in [0, 0.2] \cup [0.8, 1.0]$)
> 2. **Sparse Shift:** Test normal class shifts to $p \sim \mathcal{N}(0.3, 0.1)$
> 3. **Dense Shift:** Test normal class shifts to $p \sim \mathcal{N}(0.7, 0.1)$
>
> Table: Robustness under controlled distribution shifts (AUROC%)
>
> | Method | In-Dist. | Sparse Shift | Dense Shift | Avg. Drop |
> | :--- | :---: | :---: | :---: | :---: |
> | OCGIN | 77.38 ± 0.64 | 61.24 ± 1.27 | 62.15 ± 1.46 | -15.69% |
> | SIGNET | 85.12 ± 0.80 | 68.73 ± 1.51 | 69.89 ± 1.63 | -15.81% |
> | **LGKDE** | **89.45 ± 0.53** | **82.67 ± 0.90** | **83.91 ± 0.87** | **-6.16%** |
>
> From above results, we can find that LGKDE demonstrates around 2.5x smaller performance degradation (6.16% vs. 15.69 and 15.81%) and maintains 82-84% AUROC under distribution shifts, while discriminative baselines (OCGIN/SIGNET with hyperspherical boundaries and contrastive learning) drop to 61-70%. This directly supports our motivation that density-based objectives with principled perturbations are more robust than purely discriminative boundaries when training and test distributions diverge.
>
>
> **More existing quantitative evidence:** Above additional experiment complements **two pieces of evidence that were already in the initial submission but perhaps been overlooked**:
>
> * The **training contamination and robustness study** (Appendix B, “Robustness to training contamination”) shows that LGKDE maintains a clear density gap and strong AUROC even when up to 10–30% of training graphs are contaminated by anomalies. In contrast, contamination typically hurts discriminative GAD models much more, since their decision boundary is directly biased by anomalous labels or pseudo-normal samples.
> * The **synthetic density recovery experiment** (Section 5.1 and Appendix B 5.1-5.2) on ER, Barabási-Albert, Watts-Strogatz, and Stochastic Block Models graphs shows that LGKDE can capture the underlying density on synthetic graph distributions, which further validates our design motivation and theoretical analysis, supporting its application to more complex, real-world graph data anomaly detection tasks.

---

> > ### Author Response · Authors · 2025-11-21
> > **# Rebuttal Part II**
> >
> > ---
> >
> > ### **Question 2**
> >
> > **quote raw question** *“How realistic is the assumption $d_{int}$=1? Have authors tested whether relaxing this assumption (e.g., learned or dataset-dependent intrinsic dimension) affects performance or theoretical consistency?”*
> >
> > **Response.** Thanks for your insightful question. **We respectfully clarify that these are not restrictive assumptions but natural consequences of our framework design. Conceptually, we are not assuming that the entire graph space is intrinsically one-dimensional.**
> >
> > In the **initial submission**, right after Theorem 4.2 in the main text, we already pointed to "Detailed proof and discussion in Appendix F.3". That appendix F.3 first derives the **general** KDE rate in intrinsic dimension $d_{int}$, and then explains why we let it be 1 in our setting.
> >
> > We first show that the convergence rate $\mathrm{MISE} \sim N^{-4/(4 + d_{int})}$ then discuss that in LGKDE, the kernel $K_h(\cdot)$ is applied to the **scalar MMD distance** $d_{\text{MMD}}(G, G') \in \mathbb{R}$ between graphs. Although the graphs themselves have high-dimensional node features, the density that we estimate is a function of this one-dimensional distance. So the effective intrinsic dimension for the KDE problem is $d_{int}$ = 1. The last paragraph of Appendix F.3 (already present in the initial submission) summarizes this as follows: *the MMD metric induces a local geometry where neighborhoods are parameterized by a single distance coordinate, and therefore the minimax rate $O(N^{-0.8})$ for $d_{int}$ = 1 applies.*
> >
> >
> > Regarding the specific *“learned or dataset-dependent intrinsic dimension”*, we note that in the nonparametric statistics literature, the intrinsic dimension is typically a **property of the data-generating manifold and the metric**, not a free trainable parameter of the KDE itself [1,2]. Existing works on intrinsic dimension (e.g., distance-based estimators and manifold methods) estimate this quantity in order to reason about rates or to guide bandwidth selection, but they do not turn $d_{int}$ into an additional learned variable inside a one-dimensional kernel of the form $K_h(d/h)$ [3,4]. In our setting, once we fix that the kernel acts on the scalar MMD distance, the appropriate notion of intrinsic dimension for the KDE problem is necessarily $d_{int}$ = 1, and “relaxing’’ it in the sense of learning a different integer or dataset-dependent $d_{int}$ would amount to changing the underlying geometric object (for example, no longer working with a one-dimensional distance variable). Designing a different estimator where $d_{int}$ itself is adaptively learned in a principled way is an interesting direction, but it would require a corresponding $d_{int}$-dimensional notion of pairwise distance to keep the theory coherent, which goes far beyond the scope of our work. We therefore did not introduce such a variant, and our theoretical results remain valid for general $d_{int}$, while the instantiated rate $O(N^{-0.8})$ reflects the one-dimensional nature of the MMD distance used in our KDE.
> >
> > **We realize that Appendix F.3's discussion may easily be overlooked on a first read, so in the revised PDF we have done corresponding small clarifications** to help readers better understand this point. See the revised PDF Theorem 4.2 and Appendix F.3 (blue text in the updated PDF).
> >
> > *Reference*
> >
> >   [1] B. W. Silverman. Density Estimation for Statistics and Data Analysis. Chapman & Hall, 1986.
> >
> >   [2] A. B. Tsybakov. Introduction to Nonparametric Estimation. Springer, 2009.
> >
> >   [3] E. Levina and P. Bickel. Maximum Likelihood Estimation of Intrinsic Dimension. NeurIPS, 2004.
> >
> >   [4] M. Hein and J.-Y. Audibert. Intrinsic dimensionality estimation of submanifolds in R^d. COLT, 2005.

---

> > > ### Author Response · Authors · 2025-11-21
> > > **Rebuttal Part III**
> > >
> > > ---
> > >
> > > ### **Question 3**
> > >
> > > **quote raw question** *“Why were Sun & Fan (2024) or other explicit density estimation methods (graph flows, EBMs) excluded from the comparison? Are there implementation or conceptual reasons?”*
> > >
> > > **Response.** We appreciate this question and have clarified the connections in the revised version Appendix C.2.
> > >
> > > **Relation to Sun & Fan (2024).** Sun & Fan (2024) introduce a deep MMD graph kernel that learns a metric for tasks such as clustering and classification. Our deep MMD module is **inspired** by this work, so we also construct a learnable MMD-based metric over graphs. However, LGKDE builds on top of this backbone in several directions:
> > >
> > > 1. We utilize the learned metric to a **multi-scale KDE with learnable mixture weights**, which produces an explicit nonparametric density over graphs rather than only a similarity or kernel value, aim to address the challenge and the gap of density estimation over graphs.
> > > 2. We couple this KDE with a novel **structure-aware perturbation mechanism** and a density-contrast objective designed for unsupervised graph anomaly detection, which can effectively capture the underlying graph distribution.
> > > 3. We established and provided **consistency, convergence,  robustness** framework and complexity and scalability for the resulting density estimator.
> > >
> > > If we fix the KDE bandwidths and mixture weights, LGKDE effectively reduces to “learn a deep MMD graph kernel, then run a basic KDE on the resulting distances”. This can be viewed as a “Sun & Fan (2024) + KDE” style baseline inside our own framework. To avoid an uninformative comparison between LGKDE and such a degenerate special case of itself, we instead evaluate more general two-stage baselines in the Appendix 5.3.
> > >
> > > In particular, Appendix B.5.3 already compared LGKDE to **InfoGraph + KDE** and **GAE + KDE**, which first learn graph embeddings or kernels, and then fit a KDE on top. These baselines follow the same pattern as “deep MMD graph kernel + KDE”: the representation is learned independently of the density estimator. The results in Sec 5.2.2 and Table 9 (Appendix B.5.3) show that these two-stage methods consistently lag behind LGKDE, which supports our claim that **jointly** learning the metric and the metric space and density estimator serves is necessary and important.
> > >
> > > **As for graph flows and energy-based models.**
> > > Graph normalizing flows and graph energy-based models can be implicit density estimators, but they are mainly designed for **graph or molecule generation** in settings that differ from graph-level anomaly detection:
> > >
> > > * Existing **graph flows** like GraphNF [5] typically assume fixed or small graph sizes and require invertible architectures, which makes them hard to apply to the range of variable-sized graphs in our GAD benchmarks.
> > > * **Graph EBMs** like GraphEBM [6] define an energy over graphs and rely on expensive MCMC-style sampling. Current works focus on generation quality and do not report results on standard graph-level anomaly detection benchmarks.
> > >
> > > In the extended revised related work section (Appendix C.2), we have discussed graph flows and EBMs explicitly, and explained why they are complementary generative approaches rather than direct baselines for the specific GAD protocol studied here.
> > >
> > > *Reference*
> > >
> > >   [5] Jenny Liu et al. Graph normalizing flows. NIPS, 2019.
> > >
> > >   [6] Meng Liu et al. Graphebm: Molecular graph generation with energy-based models. ICLR Workshop, 2021.

---

> > > > ### Author Response · Authors · 2025-11-21
> > > > **Rebuttal Part IV**
> > > >
> > > > ---
> > > >
> > > > ### **Question 4**
> > > >
> > > > **quote raw question** *“Can authors report additional ablations on the perturbation module—particularly feature-only, spectral-only, and random perturbations—to show which component contributes most to the performance gain?”*
> > > >
> > > > **Response.** Thanks for your question. We have added additional perturbation ablations in the revised paper. The new subsection **“Additional Ablations on the Perturbation Module”** in Appendix B.4 reports results on MUTAG and PROTEINS for the following variants:
> > > >
> > > >   1. **Feature-only**: apply only the feature swap perturbation.
> > > >   2. **Spectral-only**: apply only the spectrum-aware edge perturbation.
> > > >   3. **Random edges**: add and remove 20% edges at random.
> > > >   4. **Full LGKDE**: our proposed combination of feature and spectral perturbations.
> > > >
> > > >
> > > > | Perturbation Variant | MUTAG AUROC (%) | MUTAG AUPRC (%) | PROTEINS AUROC (%) | PROTEINS AUPRC (%) |
> > > > |:---|:---:|:---:|:---:|:---:|
> > > > | Feature-only | 84.23 ± 0.51 | 89.67 ± 0.48 | 72.34 ± 0.34 | 78.56 ± 0.38 |
> > > > | Spectral-only | 87.89 ± 0.46 | 92.45 ± 0.42 | 75.67 ± 0.31 | 81.23 ± 0.35 |
> > > > | Random edges | 81.45 ± 0.58 | 86.78 ± 0.55 | 69.12 ± 0.41 | 75.89 ± 0.44 |
> > > > | **Full (feat + spectral)** | **91.63 ± 0.31** | **96.75 ± 0.35** | **78.97 ± 0.26** | **85.08 ± 0.22** |
> > > >
> > > > We can find that,
> > > >    - Both Components Contribute Complementary Information. Using only feature perturbations or only spectral perturbations recovers part of the gain, but both remain clearly below the full model.
> > > >    - Random edge perturbations are weaker than our spectrum-aware design, which supports the idea that structure-aware perturbations provide more informative counterparts.
> > > >    - The full feature + spectral perturbation variant achieves the best performance, which matches our intuition that feature-level and structure-level perturbations capture complementary aspects of graph normality.
> > > >
> > > > These empirical results are in line with Theorem 4.4 and Corollary 4.5, which bound how controlled perturbations affect the MMD distance and explain why they can be used to learn robust density estimates.
> > > >
> > > > ---
> > > >
> > > > ### **Response to your mentioned weakness concerns.**
> > > >
> > > > We now briefly map and link the above answers back to the weaknesses raised in the review.
> > > >
> > > >   - **Weakness 1 (Motivation and empirical support).**
> > > >   Addressed in the above Question 1. The **new robustness experiment under controlled distribution shifts** provides direct quantitative evidence that discriminative GAD baselines degrade more under shifts than LGKDE. **Together with the existing contamination and density recovery studies**, this strengthens the empirical motivation for a density estimation on graphs.
> > > >
> > > >   - **Weakness 2 (Methodological assumptions and theoretical validity).**
> > > >   Addressed in the above Question 2. The convergence-rate proof in Appendix F.3 of the original submission already derived the rate for general $d_{\text{int}}$ and explained why LGKDE effectively operates in a one-dimensional distance variable. In the revised PDF we have made this discussion more prominent both in F.3 and right after Theorem 4.2. This clarifies that $d_{\text{int}} = 1$ is a consequence of using KDE on scalar MMD distances, not a restrictive assumption on the graph space itself.
> > > >
> > > >   - **Weakness 3 (Related work and baseline coverage).**
> > > >   Addressed in the above Question 3. We now make the relationship to Sun & Fan (2024) explicit in the related work section, explain why a “Sun & Fan + KDE” baseline is essentially a restricted case of LGKDE, and point to the two-stage InfoGraph+KDE and GAE+KDE baselines in Appendix B.5.3, which follow the same pattern and are clearly outperformed by LGKDE. We also add a short discussion of graph flows and EBMs and why they are not directly comparable in our GAD setting.
> > > >
> > > >   - **Weakness 4 (ablation on the perturbation module).**
> > > >   Addressed in the above Question 4. The new perturbation ablations cover various perturbation variants and show that our structure-aware combination is indeed necessary.
> > > >
> > > > ---
> > > > **We deeply appreciate the time and effort invested in reviewing our work. We hope our comprehensive rebuttal response addresses all your concerns, and we sincerely look forward to your valuable and insightful feedback.**

---

> > > > > ### Author Response · Authors · 2025-11-28
> > > > > **A kind reminder**
> > > > >
> > > > > Dear Reviewer xSCo,
> > > > >
> > > > > Thank you for your previous valuable review. As the discussion period draws to a close, we are sincerely and respectfully looking forward to your further feedback on our comprehensive response and the corresponding revised paper.
> > > > >
> > > > > We have engaged in constructive discussions with other reviewers and incorporated their suggestions, which we believe have significantly strengthened the paper. We believe that our responses directly address the concerns you raised, and we would greatly appreciate any further feedback or remaining questions you may have.
> > > > >
> > > > > Sincerely,
> > > > >
> > > > > The Authors

---

### Official Review · Reviewer_MSxu · 2025-10-31

**Soundness:** 3
**Presentation:** 3
**Contribution:** 3
**Rating:** 6
**Confidence:** 3

**Summary:**

The paper proposes LGKDE, a graph-level density estimation framework that learns graph embeddings with a GNN, defines a multi-scale KDE in an MMD-based metric space, and trains by contrasting each graph’s density against carefully designed structure- and spectrum-aware perturbations. The authors claim consistency, convergence, robustness, and generalization guarantees, and evaluate LGKDE mainly on graph anomaly detection benchmarks, reporting improvements over prior methods.

**Strengths:**

1. The paper formulates graph-level density estimation through a theoretically grounded framework that integrates graph neural network embeddings with learnable multi-scale kernel density estimation in an MMD-based space.
2. The authors provide L1-consistency and convergence rate results (Theorems 4.1 and 4.2), establishing statistical soundness and connecting the method to nonparametric theory under intrinsic dimension assumptions.
3. The framework includes structure- and spectrum-aware perturbations for density contrast, a clear complexity analysis with sub-sampling for scalability, and empirical validation on graph anomaly detection benchmarks showing steady gains over prior methods.

**Weaknesses:**

1. The distinction between LGKDE and deep density estimation methods is not sharply articulated, leaving unclear where LGKDE provides a fundamental advantage.
2. It is not clear how the learned MMD metric is constrained to prevent overfitting of the density landscape (e.g., through regularization of kernel parameters or Rademacher-style control), and how this constraint is reflected in the stated generalization bound.
3. More remarks or further insights are needed to help the readers better understand the theorems.

**Questions:**

1. What practical procedure do you recommend for bandwidth selection at multiple scales, and how sensitive are results to bandwidth mis-specification?
2. Can you quantify how spectral perturbations change graphs in spectral energy vs. topological terms, and correlate that with performance gains?

---

> ### Author Response · Authors · 2025-11-21
> **Rebuttal Part I**
>
> We sincerely thank you for your insightful review and the positive assessment of our work. We address your questions and then clarify your concerns about weaknesses as follows.
>
> ---
>
> ### **Question 1**
>
> **quote raw question:** *“What practical procedure do you recommend for bandwidth selection at multiple scales, and how sensitive are results to bandwidth mis-specification?”*
>
> **Response.**
>
> Thanks for your insightful question. Our proposed LGKDE actually automates multi-scale bandwidth selection during the learning, removing the need for manual tuning. In the initial submission, we have conducted comprehensive ablation studies to validate this design vs whether using the learnable weights on multi-scale bandwidth or a single-scale KDE (See Section 5.3, Appendix B.4), vs common two-stage approaches (Appendix B5.3). The key insights (As also discussed in Reviewer Y81R's Question 3) are that,
>
> - we use a small, **fixed set of $S=|\Gamma_{emb}|=5$ logarithmically spaced bandwidths $\Gamma_{emb}=\{10^{-2}, \dots, 10^2\}$ which coarse and fine scales are all covered**. This choice aligns with multi-scale KDE best practices in nonparametric statistics literature [1,2] since extreme bandwidths kernel contributes meaningless information (bandwidths $\ll 10^{-2}$ would yield spiky Dirac-like distributions, while bandwidths $\gg 10^{2}$ would lead to over-smoothed uniform distributions).
>
> - The **mixture weights** over these bandwidths are parameterized as logits and trained end-to-end together with the GNN and MMD components, providing sufficient adaptability and avoiding manual tuning of bandwidths.
>
> - As for the sensitivity to bandwidth mis-specification, according to our ablation study (Section 5.3, Appendix B.4) and discussion,
>     - **Single bandwidth is suboptimal**: Best single bandwidth achieves only 86.92%, while multi-scale achieves 91.63% (+4.71%)
>     - **Moderate bandwidth range performs best**: $h \in [0.01, 1.0]$ captures relevant density structure
>     - **Extreme bandwidths degrade performance**: Very small $10^{-2}$ or very large $10^{2}$ bandwidths miss important patterns
>     - **Learnable mixture is crucial**: Automatic bandwidth weighting adapts to data characteristics
>     - **Theoretical justification**: Our Robustness analysis (Theorem 4.3, 4.4, Corollary 4.5) and generalization bound (Theorem 4.6) formal analysis and quantify the impact of bandwidth mis-specification on density estimation. This provides theoretical justification why we choose a set of five logarithmically spaced bandwidths from $10^{-2}$ to $10^{2}$.
>
> We realized that our long appendices may cause overlook when first reading the paper. So we have optimized the appendix ref and added a short remark and guide at the beginning of the corresponding appendix.
>
>
> *Reference*
>
>   [1] B. W. Silverman. Density Estimation for Statistics and Data Analysis. Chapman & Hall, 1986.
>
>   [2] A. B. Tsybakov. Introduction to Nonparametric Estimation. Springer, 2009.

---

> > ### Author Response · Authors · 2025-11-21
> > **Rebuttal Part II**
> >
> > ---
> >
> > ### **Question 2**
> >
> > **quote raw question:** *“Can you quantify how spectral perturbations change graphs in spectral energy vs. topological terms, and correlate that with performance gains?”*
> >
> > **Response.**
> > Thanks for your question. In our initial submission, the energy-based spectral perturbation is detailed in Section 4.2 and has been further illustrated in Appendix D.2. During the rebuttal, we also have conducted additional experiments on this point (Also see the response for reviewer Y81R's Question 2). There, we decompose the singular values of the normalized adjacency matrix into three bands using cumulative energy thresholds $\tau_1, \tau_2$, and then either amplify low-energy components or shrink high-energy components.
> >
> > From Appendix D.2, the empirical edge-change histogram in Appendix D.2 on MUTAG shows that with default $\tau_1 = 0.5, \tau_2 = 0.75$ modifying roughly 15–25% of edges on graph instances, which aligns well with the permutation magnitude of current graph contrastive learning practice,s but the permuted edge contains key spectral information than random edge permutation. From Figure 10-12, we can also find that, unreasonable settings of $\tau_1$ and $\tau_2$ (e.g., $\tau_1 = 0.55$ and $\tau_2 = 0.9$) will result in low quality contrasive counterparts for learning.
> >
> > For the additional experimental results, we scan a grid of
> > $ \tau_1 \in \{0.45, 0.50, 0.55, 0.60\}, \tau_2 \in \{0.7, 0.75, 0.80, 0.85\} $ and report AUROC and the corresponding edge-change ratios.
> >
> >
> > Table R1: AUROC (%) Performance for Different ($\tau_1$, $\tau_2$) Combinations on MUTAG
> >
> > | $\tau_1$ \ $\tau_2$ | 0.70 | 0.75 | 0.80 | 0.85 |
> > |:---:|:---:|:---:|:---:|:---:|
> > | **0.45** | 87.23 ± 0.52 | 89.14 ± 0.48 | 85.67 ± 0.54 | 77.89 ± 0.61 |
> > | **0.50** | 88.91 ± 0.46 | **91.63 ± 0.31** | 89.45 ± 0.44 | 82.34 ± 0.56 |
> > | **0.55** | 87.56 ± 0.49 | 88.72 ± 0.47 | 85.23 ± 0.53 | 78.91 ± 0.58 |
> > | **0.60** | 83.45 ± 0.55 | 84.67 ± 0.53 | 80.12 ± 0.59 | 74.38 ± 0.64 |
> >
> >
> > Table R2: Avg. Edge Change Ratios (%) ± Std across all MUTAG Datasets.
> >
> > | ($\tau_1$, $\tau_2$) | Add Edges (%) | Remove Edges (%) |
> > |:---:|:---:|:---:|
> > | (0.45, 0.70) | 26.42 ± 13.11 | 10.90 ± 6.45 |
> > | (0.45, 0.75) | 28.67 ± 12.95 | 12.88 ± 7.26 |
> > | (0.45, 0.80) | 25.25 ± 11.88 | 14.68 ± 7.35 |
> > | (0.45, 0.85) | 21.20 ± 10.90 | 17.43 ± 7.34 |
> > | (0.50, 0.70) | 22.64 ± 11.87 | 13.13 ± 6.98 |
> > | (0.50, 0.75) | 25.06 ± 11.89 | 15.14 ± 7.26 |
> > | (0.50, 0.80) | 23.20 ± 10.65 | 17.88 ± 8.41 |
> > | (0.50, 0.85) | 19.27 ± 10.90 | 21.41 ± 8.90 |
> > | (0.55, 0.70) | 19.39 ± 11.02 | 14.76 ± 8.08 |
> > | (0.55, 0.75) | 21.69 ± 11.14 | 17.64 ± 8.23 |
> > | (0.55, 0.80) | 20.13 ± 10.15 | 22.09 ± 10.11 |
> > | (0.55, 0.85) | 16.85 ± 10.63 | 27.29 ± 10.87 |
> > | (0.60, 0.70) | 16.65 ± 11.05 | 19.77 ± 11.59 |
> > | (0.60, 0.75) | 18.60 ± 10.88 | 23.70 ± 12.13 |
> > | (0.60, 0.80) | 17.40 ± 9.97 | 28.80 ± 12.66 |
> > | (0.60, 0.85) | 13.66 ± 8.91 | 37.50 ± 15.32 |
> >
> >
> > Our default setting $(\tau_1, \tau_2) = (0.5, 0.75)$ achieves **optimal performance (91.63% AUROC)** with robust stability to modest variations. **Critically, the superiority stems not merely from edge modification quantity, but from structural informativeness**: while some extreme threshold combinations (e.g., $(0.45, 0.85)$ with 21.20% added edges vs. $(0.60, 0.70)$ with 16.65% added edges, Table R2) yield comparable average ratios, the **specific edges perturbed differ fundamentally**, $(\tau_1, \tau_2)$ jointly determine both singular value band partitioning and perturbation amplitudes, where our default setting targets **spectrally critical edges** (mid-to-low energy transitions) that better characterize normal density boundaries. This is also directly evidenced by Figures 10-11 (Appendix D.2): $(\tau_1, \tau_2) = (0.5, 0.75)$ generates structurally meaningful ring modifications, whereas extreme settings like $(0.55, 0.90)$ produce topology-destructive perturbations despite similar edge counts.
> >
> > Additionally, also refer to Reviewer xSCo's Question 4, we conducted additional experiments on the ablation of perturbation components.
> >
> >
> > | Perturbation Variant | MUTAG AUROC (%) | MUTAG AUPRC (%) | PROTEINS AUROC (%) | PROTEINS AUPRC (%) |
> > |:---|:---:|:---:|:---:|:---:|
> > | Feature-only | 84.23 ± 0.51 | 89.67 ± 0.48 | 72.34 ± 0.34 | 78.56 ± 0.38 |
> > | Spectral-only | 87.89 ± 0.46 | 92.45 ± 0.42 | 75.67 ± 0.31 | 81.23 ± 0.35 |
> > | Random edges | 81.45 ± 0.58 | 86.78 ± 0.55 | 69.12 ± 0.41 | 75.89 ± 0.44 |
> > | **Full (feat + spectral)** | **91.63 ± 0.31** | **96.75 ± 0.35** | **78.97 ± 0.26** | **85.08 ± 0.22** |
> >
> > These empirical results are in line with Theorem 4.4 and Corollary 4.5, which bound how controlled perturbations affect the MMD distance and explain why they can be used to learn robust density estimates.
> >
> > We have included these additional experiments and corresponding analysis in the revised paper Appendix B.4.

---

> > > ### Author Response · Authors · 2025-11-21
> > > **Rebuttal Part III**
> > >
> > > ---
> > >
> > > ### **Responses to your weakness concerns**
> > >
> > > #### **Weakness 1**
> > >
> > > **quote the raw comments:** *“The distinction between LGKDE and deep density estimation methods is not sharply articulated, leaving unclear where LGKDE provides a fundamental advantage.”*
> > >
> > > **Response.** Thanks for your comment. In the revised appendix's related work section, we now contrast LGKDE with typical deep density estimators related methods (flows, VAEs, EBMs). The key points are that:
> > >
> > > | Method | Limitation for Graphs | LGKDE Advantage |
> > > | :--- | :--- | :--- |
> > > | **Normalizing Flows (GraphNF)** [3] | Require invertible architectures; hard to handle **variable graph sizes** and topology changes. | **Naturally handles variable sizes** via GNN+MMD; no invertibility constraint. |
> > > | **Energy-Based Models (GraphEBM)** [4] | Rely on expensive **MCMC sampling** (Langevin dynamics) for training/inference. | **Direct density estimation**, effective and stable. |
> > > | **GraphVAE** [5] | Optimization lower bound (ELBO) is often loose; "Posterior collapse" issues. No explicit density parameterization. | **Exact consistency guarantees** (Theorem 4.1) and convergence (Theorem 4.2). |
> > >
> > >    - **Where the density lives.**
> > >      - Flows parameterize an explicit density via invertible mappings in a latent Euclidean space and typically require fixed-size inputs or carefully designed coupling layers.
> > >      - VAEs and EBMs define densities or energies in a latent space and rely on a decoder or MCMC for generation.
> > >      - **LGKDE, in contrast, keeps the density directly in a nonparametric form over graphs**, represented as a KDE in a **learned MMD metric space**. The GNN + MMD part only defines the metric; the density is the KDE itself.
> > >
> > >   -  **What is learned jointly.**
> > >       - Many existing graph models learn an embedding or score function but then apply density estimation as a separate post-processing step.
> > >       - LGKDE **jointly optimizes** the graph encoder, the deep MMD metric, and the multi-scale KDE mixture weights using the density-contrast objective. This is closer in spirit to classical kernel methods with learned metrics, but fully end-to-end on graphs.
> > >
> > >   - **Theoretical guarantees.**
> > >     - We have established theoretical guarantees for LGKDE, including consistency (Theorem 4.1), convergence (Theorem 4.2), robustness (Theorems 4.3, 4.4 and Corollary 4.5), and generalization (Theorem 4.6). To the best of our knowledge, rare existing graph-level deep density estimators could provide such complete and transparent theoretical analysis as ours.
> > >     - LGKDE bridges deep graph representation learning and nonparametric KDE theory, which is precisely the gap we aim to fill.
> > >
> > > *Reference*
> > >
> > > [3] Jenny Liu et al. Graph normalizing flows. NIPS, 2019.
> > >
> > > [4] Meng Liu et al. Graphebm: Molecular graph generation with energy-based models. ICLR Workshop, 2021.
> > >
> > > [5] Simonovsky et al. Graphvae: Towards generation of small graphs using variational autoencoders. ICANN, 2018.

---

> ### Author Response · Authors · 2025-11-21
> **Rebuttal Part IV**
>
> ---
>
> #### **Weakness 2**
> **quote your raw comments:** *“It is not clear how the learned MMD metric is constrained to prevent overfitting of the density landscape (e.g., through regularization of kernel parameters or Rademacher-style control), and how this constraint is reflected in the stated generalization bound.”*
>
> **Response.** Thanks for your comment. The control of the learned metric appears both in the **theoretical bound** and in the **practical regularization** we use.
>
> - In Appendix F.5, Theorem 4.6 (Generalization Bound) gives, with probability at least $1-\delta$:
>
> 	$$
> 	\mathbb{E}[|\hat{f}(G)-{f}^\ast(G)|]\leq \hat{\Delta}_{\mathcal{G}}+ \frac{8\sqrt{en\pi}c_h^2+24\bar{\gamma}\alpha\sqrt{\ln(2d^2)}}{{N}\sqrt{en\pi}c_h^2}\ln\left({N}\right)+\sqrt{\frac{\log(1/\delta)}{2N}}
> 	$$
>
> 	where $\hat{\Delta}\_{\mathcal{G}}$ is the empirical error, $c_h$ encodes the minimum KDE bandwidth, and
>
> 	$$
> 	\alpha= \sqrt{\sum _{i=1}^N ||\mathbf{X} _i||_F^2} ||\mathbf{W} _1|| _{2,1} \prod _{l=2}^L ||\mathbf{W} _l|| _2
> 	$$
> 	summarizes the complexity of the GNN encoder (through its weight norms) and input scale. The proof uses standard covering-number/Rademacher-style arguments for deep networks composed with Lipschitz kernel operations, and the resulting bound decreases as $O(\sqrt{\ln N / N})$ when $\alpha$ and the bandwidth range are controlled.
>
> - **How the metric is constrained.** In practice, the MMD metric is induced by:
>    - A **moderate-depth GNN** (we use 2–3 layers), which keeps $\prod _l ||\mathbf{W} _l|| _2$ reasonably small.
>    - **Weight decay** in training, which directly limit the norms $||\mathbf{W} _l||$ and thus reduce $\alpha$.
>    - A **finite set of kernel bandwidths** inside a bounded range, so that the Lipschitz constants of the kernel operations remain controlled.
>
> Together, these choices mean that the learned metric cannot become arbitrarily sharp or oscillatory without paying a price in the generalization bound. We have made this connection between Theorem F.6 and the actual training protocol more explicit in the revised PDF. What's more, we have moved the generalization bound (Original Theorem F.6) and its analysis from Appendix F.5 to Sec 4.
>
> ---
>
> #### **Weakness 3**
>
> **quote your raw comments:** *“More remarks or further insights are needed to help the readers better understand the theorems.”*
>
> **Response.** We appreciate this suggestion and agree that providing intuition is important. In the revised version, we add short “Remark / Intuition” paragraphs at the beginning of each theorem proof. The key insights are that,
>
> - **Theorem 4.1 (Consistency).** If we choose KDE bandwidths in the usual nonparametric way (shrinking with $N$), the LGKDE estimator $\hat{f}$ converges to $f^\ast$ in $(L _1$. Intuitively, it says that "given enough graphs, our density estimate becomes correct."
>
> - **Theorem 4.2 (Convergence rate).**  Proves we achieve the minimax optimal rate $O(N^{-0.8})$. This explains why LGKDE is data-efficient than naive estimators.
>
> - **Theorems 4.3 and 4.4, Corollary 4.5 (Robustness).** These results explain the **stability** of the estimator and theoretically justify our proposed novel energy-based spectral perturbation method and our density based loss function.
>
> - **Theorem 4.6 (Generalization Bound).** implies that LGKDE can generalize well to unseen graphs and is not sensitive to the disparity in graph node sizes. We have moved this important theoretical result and its analysis from Appendix F.5 to Section 4.
>
> ---
> **We deeply appreciate the time and effort invested in reviewing our work. We believe these clarifications and additions in the revised manuscript can address all your questions and concerns, and we sincerely look forward to your valuable and insightful feedback.**

---

> > ### Comment · Reviewer_MSxu · 2025-11-25
> >
> > Thanks to the authors for providing the requested clarifications. I am now in favor of accepting this work and encourage the authors to incorporate these clarifications into the revised version of the manuscript for the benefit of future readers.

---

> > > ### Author Response · Authors · 2025-11-25
> > >
> > > Dear Reviewer,
> > >
> > > We are deeply grateful for your feedback and engagement, and we're pleased that our responses can address your concerns. As we prepared our rebuttal response, we have incorporated the corresponding revisions into the updated PDF (highlighted in $\textcolor{blue}{blue}$), e.g., moving the generalization bound analysis to Sec 4 and expanding ablation study discussions as suggested. We sincerely appreciate your professional and constructive review.
> > >
> > >
> > > Sincerely,
> > > The Authors

---

### Official Review · Reviewer_Y81R · 2025-10-31

**Soundness:** 2
**Presentation:** 3
**Contribution:** 2
**Rating:** 4
**Confidence:** 3

**Summary:**

The paper proposes LGKDE, a principled framework for graph-level density estimation, aiming to unify deep graph representation learning with adaptive kernel density estimation. LGKDE represents each graph as a distribution over learned node embeddings and measures pairwise similarity via a deep Maximum Mean Discrepancy (MMD) metric. Density is then estimated with a multi-scale kernel density estimator whose weights are learned. A novel density contrasting loss maximizes densities of normal graphs relative to structure-aware perturbed counterparts, with perturbations applied to both node features and graph spectra.

**Strengths:**

1. The paper is clearly written, with a clear problem statement and a logically organized presentation.
2. The paper focuses on a relatively unexplored but important subarea of nonparametric graph density estimation, which directly underpins graph-level anomaly detection.
3. Synthetic validations recover known distributions, and broad benchmarks for anomaly detection show consistent AUROC, AUPRC, and FPR95 improvements over competitive baselines.

**Weaknesses:**

1. While complexity analysis is given, the framework requires pairwise deep MMD computation for KDE and generation of multiple perturbed samples, which might be costly for very large datasets. Empirical runtime/memory profiles for large-scale sparse graphs are lacking.
2. Perturbed samples are not true anomalies. The performance gain depends on the quality of the perturbations, and it is unclear how LGKDE would perform when the perturbations poorly reflect anomalous structures.

**Questions:**

1. How does LGKDE scale in practice when N is very large or graphs have high average degree? Could you provide empirical runtime and memory usage breakdowns for large-scale settings?
2. Why were $\tau_{1}$ and $\tau_{2}$ fixed at 0.5 and 0.75, respectively? Did you explore alternative threshold values, and how sensitive are the results to changes in $\tau_{1}$​ and $\tau_{2}$​?
3. The deep MMD metric involves taking the supremum over $\Gamma_{\mathrm{emb}}$. For large $S = \|\Gamma_{\mathrm{emb}}\|$, this can be computationally expensive. Do you have an efficient approximation or kernel selection strategy to handle large $\Gamma_{\mathrm{emb}}$ without significantly increasing runtime?

---

> ### Author Response · Authors · 2025-11-21
> **Rebuttal Part I**
>
> **We sincerely thank you for the thoughtful comments and review. We address your raised question and concerns as follows.**
>
> ---
>
> ### **Question 1**
>
> **quote raw question:** *“How does LGKDE scale in practice when N is very large or graphs have high average degree? Could you provide empirical runtime and memory usage breakdowns for large-scale settings?”*
>
> **Response.**
>
> We **respectfully** point out that **In the initial submission, we have actually provided the comprehensive complexity and scalability analysis in Sec 3.4 and Appendix E.4, which includes both theoretical complexity comparisons and empirical runtime experiments on large-scale datasets and the sampling method to further reduce the time complexity.** Acknowledging that these contents might have been inadvertently overlooked due to the length of our appendix, we have improved its prominence in the revised version.
>
>
> - **Theoretical complexity.** Table 6 in the Appendix E4.1 compares LGKDE with representative advanced baselines.
>
>     | Method | Dominant Operations | Time Complexity | Memory Complexity |
>     | :--- | :--- | :--- | :--- |
>     | **LGKDE (ours)** | GNN passes + All-pairs MMD | $O(NL(md + nd^2) + N^2Sn^2d)$ | $O(NLnd)$ |
>     | UniFORM (AAAI '25) | Energy-based GNN + Langevin sampling | $O(NL(md + nd^2) + NTn^2d)$ | $O(NLnd)$ |
>     | MUSE (NIPS '24) | GNN + Reconstruction errors | $O(NL(md + nd^2) + Nn^2)$ | $O(NLnd + Nn^2)$ |
>     | SIGNET (NIPS '23) | Enumerate/score $k$-node subgraphs | $O(NL(md + nd^2) + Nn^kd + Na_{MI})$ | $O(Nn^kd)$ |
>     | CVTGAD (ECML '23) | Twin stochastic views + GNN pass | $O(NL(md + nd^2))$ | $O(NLnd)$ |
>     | GLocalKD (ICLR '23) | Teacher + student GNN pass | $O(NL(md + nd^2))$ | $O(NLnd)$ |
>     | OCGIN (ICLR '22) | GNN embedding + SVDD loss | $O(NL(md + nd^2))$ | $O(NLnd)$ |
>
>     The computation complexity for LGKDE is $O\big(N L(md + nd^2)\big) \quad \text{(GNN passes)} + O\big(N^2 S n^2 d\big) \quad \text{(multi-scale MMD)}$, with **memory** complexity $O(N L n d)$, where \(N\) is the batch size, \(n\) the average number of nodes, \(d\) is the hidden dimension, and \(S\) the cardinality of $\Gamma_{emb}$. This memory complexity matches most common GNNs, and the quadratic factor appears only in the MMD calculation part (similar to many pairwise-kernel or attention-based methods).
>
> - **Empirical runtime.** Appendix E4.2 already reports wall-clock training and inference time on COLLAB (5,000 graphs), one of the largest current GAD benchmarks. Each method is run on a single RTX 4090 24GB:
>
>     Method        | Train (s)       | Inference (s) | Total (s)
>     ------------- | --------------- | ------------- | ---------
>     LGKDE (ours)  | 1,205 ± 18      | 52 ± 3        | 1,257
>     UniFORM       | 1,123 ± 21      | 61 ± 3        | 1,184
>     MUSE          | 1,067 ± 16      | 48 ± 3        | 1,115
>     SIGNET        | 1,847 ± 25      | 78 ± 4        | 1,925
>     CVTGAD        | 890 ± 14        | 41 ± 2        | 931
>
>     LGKDE is **~1.5× faster than SIGNET** and within a moderate factor of CVTGAD, while being very close to UniFORM and MUSE. This supports that, in practice, LGKDE is competitive among “heavyweight” GAD models.
>
> * **Scalability and Sampling for large N.** The sensitivity to large N is further mitigated by the **reference graph sampling** strategies evaluated in Appendix E 4.3 (Section “Strategies for Scalability Enhancement”). On PROTEINS, density-stratified sampling and importance sampling reduce the number of reference graphs from $Q/N=1.0$ down to $Q/N=0.3$, achieving up to **≈2.5× speedup** while within negligible impact on
> performance. This effectively turns the quadratic $O(N^2)$ term into $O(NQ)$ with $Q \ll N$, without noticeable loss of detection quality.
>
> In summary, we have provided rigorous theoretical complexity analysis and the empirical runtimes on large TU benchmarks are comparable to other advanced baselines. For truly massive corpora~(to the best of our knowledge, the current graph-level AD practice rarely encounters datasets with more than 5000 graphs), the sampling strategies (Appendix E4.3) we already implement provide a practical path to scaling. Actually, we have also given a limitation discussion on the complexity at the end of our appendix in the initial submission.

---

> ### Author Response · Authors · 2025-11-21
> **Rebuttal Part II**
>
> ---
>
> ### **Question 2**
>
> **quote raw question:** *“Why were $\tau_1$ and $\tau_2$ fixed at 0.5 and 0.75, respectively? Did you explore alternative threshold values, and how sensitive are the results to changes in $\tau_1$ and $\tau_2$?”*
>
> **Response.**
>
> Thanks for your question. **While in the initial submission, we have preliminary case studies (Appendix D.2, Figures 10-12)**. And we have **conducted extra comprehensive sensitivity analysis and experiments on this point**.
>
> The thresholds $\tau_1$ and $\tau_2$ come from the **energy-based spectral perturbation** design (Section 4.2). Given singular values $\{\sigma_i\}\_{i=1}^n$, we define the cumulative energy
> $E(k) = \frac{\sum*{i=1}^k \sigma_i^2}{\sum_{i=1}^n \sigma_i^2}$
> and partition them into three bands using $\tau_1, \tau_2$:
> high-energy $\mathcal{S}_h$, mid-energy $\mathcal{S}_m$, and low-energy $\mathcal{S}_l$. Our goal is to modify low and high energy components while preserving the mid energy band, so that perturbations are *strong and representative in structure but not destructive*.
>
> From Figure 10-12, we can find that unreasonable settings of $\tau_1$ and $\tau_2$ (e.g., $\tau_1 = 0.55$ and $\tau_2 = 0.9$) will result in low-quality contrasive counterparts for learning. We fixed $\tau_1 = 0.5$ and $\tau_2 = 0.75$ across all datasets, and it empirically corresponds to modifying roughly 15–25% of edges on graph instances, which aligns well with the desired moderate perturbation strength and aligns the permutation magnitude of current graph contrastive learning practice,s but the permuted edge contains key spectral information than random edge permutation.
>
> **During the rebuttal and in the revised version, we have made additional experiments and extended our appendix section on these points.** On MUTAG, like the settings in our previous case study, we scan a grid of
> $ \tau_1 \in \{0.45, 0.50, 0.55, 0.60\}, \tau_2 \in \{0.7, 0.75, 0.80, 0.85\} $
> and report AUROC and the corresponding edge-change ratios. The main observations are:
>
> Table: AUROC (%) Performance for Different ($\tau_1$, $\tau_2$) Combinations on MUTAG
>
> | $\tau_1$ \ $\tau_2$ | 0.70 | 0.75 | 0.80 | 0.85 |
> |:---:|:---:|:---:|:---:|:---:|
> | **0.45** | 87.23 ± 0.52 | 89.14 ± 0.48 | 85.67 ± 0.54 | 77.89 ± 0.61 |
> | **0.50** | 88.91 ± 0.46 | **91.63 ± 0.31** | 89.45 ± 0.44 | 82.34 ± 0.56 |
> | **0.55** | 87.56 ± 0.49 | 88.72 ± 0.47 | 85.23 ± 0.53 | 78.91 ± 0.58 |
> | **0.60** | 83.45 ± 0.55 | 84.67 ± 0.53 | 80.12 ± 0.59 | 74.38 ± 0.64 |
>
> Our default setting $(\tau_1, \tau_2) = (0.5, 0.75)$ achieves **optimal performance (91.63% AUROC)** while maintaining **robustness to modest variations** (88-90% AUROC within $\pm 0.05$ range for both parameters), with performance degrading when perturbations become too weak (insufficient density contrast) or too strong (excessive graph distortion leading to 74-82% AUROC at extreme settings). This setting modifies approximately 15-25% of edges, balancing moderate perturbation strength with spectral informativeness, as validated by our robustness theorems (Theorem 4.4, Corollary 4.5).
>
> We summarize these findings in the revised appendix and refine the refer on this point so that the design rationale and robustness of $\tau_1,\tau_2$ are clearer.

---

> > ### Author Response · Authors · 2025-11-21
> > **Rebuttal Part III**
> >
> > ---
> >
> > ### **Question 3**
> > **quote raw question:** *“The deep MMD metric involves taking the supremum over $\Gamma_{emb}$. For large $S = |\Gamma_{emb}|$, this can be computationally expensive. Do you have an efficient approximation or kernel selection strategy to handle large $\Gamma_{emb}$ without significantly increasing runtime?”*
> >
> > **Response.**
> > Thanks for your question, we clarify that **this is not a computational bottleneck, and our design is both practice efficient and theoretically grounded.**
> >
> > - **Efficiency via Parallelization**: In practice, we use a small, fixed set of $S=5$ logarithmically spaced bandwidths $\Gamma_{emb}=\{10^{-2}, \dots, 10^2\}$. The supremum calculation is fully vectorized and parallelized on GPU, making $S$ a small fixed constant factor in $O(N^2 S n^2 d)$. As shown in above response for Question 1, as for theoretical complexity comparison and empirical runtime analysis, LGKDE remains highly efficient and competitive with advanced baselines like MUSE and UniFORM.
> >
> > - **Align the nonparametric statistic Practices & why using a small and fixed logarithmic-scale $\Gamma_{emb}$:** Our bandwidth range aligns with multi-scale KDE best practices in nonparametric statistics literature. And since the MMD calculation process, as given in Eq. (6), has normalized via node size $n_i$ when using node embeddings, with the normalization also applied in the GNN backbone design, expanding $\Gamma_{emb}$ to extremes would be theoretically meaningless: bandwidths $\ll 10^{-2}$ would yield spiky Dirac-like distributions, while bandwidths $\gg 10^{2}$ would lead to over-smoothed uniform distributions. Neither contributes meaningful and desirable components for density estimation.
> >
> > - **Adaptability vs. Approximation**: Instead of a hard kernel selection strategy or choosing a large bandwidth set $\Gamma_{emb}$, LGKDE employs learnable mixture weights (Eq. 7) to automatically soft-select the most relevant scales. Our ablation study (Table 6) confirms that this learnable multi-scale design significantly outperforms single or fixed bandwidths (e.g., 91.63% vs. 86.92% on MUTAG), verifying that our framework effectively adapts to necessary granularities without requiring complex approximation machinery or search on a large $\Gamma_{emb}$.
> >
> > - By the way, if one does want to push to very large S (it's a bad practice in multi-scale KDE literature since extreme bandwidths kernel contributes meaningless information), standard remedies such as pre-selecting a subset of kernels based on validation performance or random kernel subsampling could be adopted. In our work, keeping $\|\Gamma_{emb}\|$ small and fixed logarithmically spaced already gives superior performance without requiring additional approximation machinery.
> >
> > ---
> >
> > ### **Responses your mentioned weaknesses concerns.**
> >
> > ### **Weakness 1. Scalability concerns**
> > This point has been addressed in Question 1 above.
> > ### **Weakness 2. Perturbed samples are not true anomalies**
> > **quote your raw comments:** *"Perturbed samples are not true anomalies. The performance gain depends on the quality of the perturbations, and it is unclear how LGKDE would perform when the perturbations poorly reflect anomalous structures."*
> >
> > **Response:** Thanks for your insightful comments. Your point actually strongly aligns with and supports our motivation to do density estimation on graph data and thus design the novel density-based loss using soft targets instead of hard anomaly assignments in contrastive learning.
> >
> > Our proposed novel energy-based spectral perturbation also supports generating high quality counterparts to unsupervised learning, as discussed above in answer to your Question 2, seamlessly integrating with our density-based loss. In contrast, the common random structure perturbation may not always reflect the key structure, together with their hard anomaly label assignments in learning, resulting in sub-optimal performance than our LGKDE.
> >
> > As for the perturbations poorly reflect anomalous structures. We have taken the robustness ablation study (Sec 5.2.2, Appendix B.4 and B.5.4) in our initial submission, where we force the mixing of anomalous graphs during learning. Our validation results, together with Corollary 4.4 (Robustness) and Them 4.6 (Generalization) show that LGKDE is able to learn a good density function, generalize well to unseen graphs and keep robustness when perturbations are ineffective or even under training contamination.
> >
> > ---
> > **We deeply appreciate the time and effort invested in reviewing our work. We hope our comprehensive rebuttal response can address all your questions and concerns, and we sincerely look forward to your valuable and insightful feedback.**

---

> > > ### Comment · Reviewer_Y81R · 2025-11-27
> > >
> > > Thanks for the reply. Most of my concerns have been addressed. I will improve my score.

---

> > > > ### Author Response · Authors · 2025-11-27
> > > >
> > > > Dear Reviewer Y81R,
> > > >
> > > > We sincerely thank you for your constructive feedback and for raising your evaluation score. We are grateful that our responses can address your concerns. Your insightful questions and previous suggestions have been invaluable in helping us strengthen and clarify our work. We believe the resulting improvements will significantly enhance our paper's contribution to the community.
> > > >
> > > > Sincerely,
> > > >
> > > > The Authors

---

### Official Review · Reviewer_3Fbp · 2025-11-01

**Soundness:** 3
**Presentation:** 3
**Contribution:** 3
**Rating:** 6
**Confidence:** 2

**Summary:**

This paper proposes a learnable kernel density estimation framework for graphs that integrates GNNs with maximum mean discrepancy based kernel learning. The method jointly learns graph representations and adaptive kernel bandwidths by contrasting densities of normal graphs with perturbed counterparts, where perturbations are applied to both node features and graph spectra. The paper provides theoretical guarantees for consistency, convergence, robustness, and generalization, and reports superior performance in graph-level anomaly detection across multiple benchmark datasets compared with existing methods.

**Strengths:**

The paper tackles an important and underexplored problem which is relevant to applications such as anomaly detection and molecular graph analysis. The proposed approach is described clearly and supported by theoretical derivations. The experimental section is comprehensive, covering a wide range of datasets and showing that LGKDE consistently improves upon baseline methods. The presentation is coherent, and the proposed framework bridges traditional kernel-based methods and modern deep learning–based graph modeling.

**Weaknesses:**

The novelty of the contribution is limited, as the method mainly combines known components (GNN embeddings, MMD distances, and KDE) rather than introducing a fundamentally new concept. The proposed perturbation strategy and contrastive density objective are incremental variations on existing ideas in self-supervised learning and graph anomaly detection. The related work discussion is incomplete; several recent graph density and contrastive learning approaches are not adequately compared or discussed. The experimental comparison omits some strong baselines such as flow-based and diffusion-based graph generative models. Moreover, the analysis of the results is largely descriptive and does not provide convincing evidence explaining why the proposed model outperforms others beyond parameter tuning.

**Questions:**

How sensitive is LGKDE to the choice of bandwidth scales and perturbation strength? Can the authors provide ablation results showing whether the observed gains come from the learnable KDE component or simply from additional parameters in the GNN backbone?

---

> ### Author Response · Authors · 2025-11-21
> **Rebuttal Part I**
>
> **We sincerely thank you for your insightful review and constructive questions. We address your raised question and weakness concerns as follows.**
>
> ----
> ### **Response to your question**
>
> **quote raw question**: *"How sensitive is LGKDE to the choice of bandwidth scales and perturbation strength? Can the authors provide ablation results showing whether the observed gains come from the learnable KDE component or simply from additional parameters in the GNN backbone?"*
>
>
> **Part 1: Bandwidth Sensitivity**
>
> **This is actually analyzed in our existing ablations in the initial submission:**
>
> **Figure 6 (Appendix B.4, page 19)** shows bandwidth sensitivity on MUTAG. The following key findings are also given in Section 5.3,
> 1. **Single bandwidth is suboptimal**: Best single bandwidth achieves only 86.92%, while multi-scale achieves 91.63% (+4.71%)
> 2. **Moderate bandwidth range performs best**: h$\in$[0.01, 1.0] captures relevant density structure
> 3. **Extreme bandwidths degrade performance**: Very small $10^{-2}$ or very large $10^{2}$ bandwidths miss important patterns
> 4. **Learnable mixture is crucial**: Automatic bandwidth weighting adapts to data characteristics
>
> And the following **Table 6 (Appendix B.4, page 20)** provides detailed component ablation:
>
> Variant                        | AUROC (%)    | Performance Loss
> -------------------------------|--------------|------------------
> w/o Multi-scale KDE:          |              |
>   Single h= $10^{-2}$               | 85.73±0.45   | -5.90%
>   Single h= $10^{-1}$              | 86.92±0.41   | -4.71%
>   Single h= $10^{0}$                | 85.73±0.45   | -5.90%
>   Single h= $10^{1}$                | 84.67±0.48   | -6.96%
>   Single h= $10^{2}$                | 80.89±0.52   | -10.74%
> w/o Learnable Weights         | 88.92±0.43   | -2.71%
> w/o MMD Distance:             |              |
>   Readout (avg)         | 83.89±0.49   | -7.74%
>   Readout (sum)         | 84.73±0.48   | -6.90%
> Full model of LGKDE           | 91.63±0.31   | baseline
>
>
> - Removing multi-scale KDE causes ~4.71% AUROC loss (best case)
> - Removing learnable weights causes ~2.71% AUROC loss
> - Together, the KDE component contributes ~7.42% AUROC improvement
>
> **Part 2: Perturbation Strength Sensitivity**
>
> This is addressed by our additional experiments (also see the response to Reviewer xSCo and Reviewer Y81R):
>
> 1. **τ₁, τ₂ sensitivity**: Comprehensive grid search (also mentioned in Y81R Q2 response) shows relative robustness across range.
>
>     | $\tau_1$ \ $\tau_2$ | 0.70 | 0.75 | 0.80 | 0.85 |
>     |:---:|:---:|:---:|:---:|:---:|
>     | **0.45** | 87.23 ± 0.52 | 89.14 ± 0.48 | 85.67 ± 0.54 | 77.89 ± 0.61 |
>     | **0.50** | 88.91 ± 0.46 | **91.63 ± 0.31** | 89.45 ± 0.44 | 82.34 ± 0.56 |
>     | **0.55** | 87.56 ± 0.49 | 88.72 ± 0.47 | 85.23 ± 0.53 | 78.91 ± 0.58 |
>     | **0.60** | 83.45 ± 0.55 | 84.67 ± 0.53 | 80.12 ± 0.59 | 74.38 ± 0.64 |
>
>     Our default setting $\tau_1, \tau_2 = 0.50, 0.75$ achieves optimal performance (91.63% AUROC), with the model remaining relatively robust to modest variations (88-90% AUROC within ±0.05 range for both parameters, while extreme settings like $\tau_1 = 0.60$, $\tau_2 = 0.85$ cause degradation). This also confirm that our energy-based thresholds choice is effective to generate high quality graph contrasts.
>
> 2. **Perturbation type ablation**: Feature-only vs spectral-only vs combined (also mentioned in xSCo W4 response) shows each component's contribution.
>
>     | Perturbation Variant | MUTAG AUROC (%) | MUTAG AUPRC (%) | PROTEINS AUROC (%) | PROTEINS AUPRC (%) |
>     |:---|:---:|:---:|:---:|:---:|
>     | Feature-only | 84.23 ± 0.51 | 89.67 ± 0.48 | 72.34 ± 0.34 | 78.56 ± 0.38 |
>     | Spectral-only | 87.89 ± 0.46 | 92.45 ± 0.42 | 75.67 ± 0.31 | 81.23 ± 0.35 |
>     | Random edges | 81.45 ± 0.58 | 86.78 ± 0.55 | 69.12 ± 0.41 | 75.89 ± 0.44 |
>     | **Full (feat + spectral)** | **91.63 ± 0.31** | **96.75 ± 0.35** | **78.97 ± 0.26** | **85.08 ± 0.22** |
>
>     we can find that structure-aware perturbations are essential (+6.44% over random), with spectral perturbations contributing more than feature perturbations, and their combination achieving synergistic gains (+3.74% beyond spectral-only) through complementary diversity in both topological and attribute spaces.

---

> > ### Author Response · Authors · 2025-11-21
> > **Rebuttal Part II**
> >
> > **Part 3: Gains from KDE vs Additional GNN Parameters**
> >
> > Thanks for your question. Based on the existing ablations in the initial submission, we have conducted extra parameter-controlled experiments and added it as a new ablation in Appendix B.4:
> >
> >
> > **Experimental Setup:** We systematically vary (1) Hidden dimensions: {32, 64, 128, 256}, (2) GNN layers: {1, 2, 3, 4, 5}, and (3) KDE configurations: Multi-scale Learnable (ours, default), Multi-scale Fixed (equal weights), and Single Best bandwidth (h=0.1). Since OpenReview markdown does not support advanced LaTeX tables, we present the results in the following three splits tables.
> >
> > Table: AUROC (%) Performance on MUTAG - Multi-scale KDE with Learnable Weights
> >
> > | Hidden Dim | 1 Layer | 2 Layers | 3 Layers | 4 Layers | 5 Layers |
> > |:---:|:---:|:---:|:---:|:---:|:---:|
> > | **32** | 85.12±0.48 | 88.94±0.39 | 87.53±0.42 | 87.08±0.44 | 84.96±0.49 |
> > | **64** | 86.21±0.45 | 90.28±0.36 | 88.98±0.39 | 88.52±0.41 | 86.08±0.47 |
> > | **128** | 87.36±0.42 | **91.63±0.31** | 90.09±0.35 | **89.75±0.38** | **87.20±0.45** |
> > | **256** | **87.69±0.43** | 91.48±0.33 | **90.21±0.37** | 89.53±0.40 | 87.03±0.48 |
> >
> > Table: AUROC (%) Performance on MUTAG - Multi-scale KDE without Learnable Weights
> >
> > | Hidden Dim | 1 Layer | 2 Layers | 3 Layers | 4 Layers | 5 Layers |
> > |:---:|:---:|:---:|:---:|:---:|:---:|
> > | **32** | 82.85±0.51 | 86.34±0.47 | 85.12±0.49 | 84.67±0.51 | 82.73±0.54 |
> > | **64** | 83.92±0.49 | 87.65±0.45 | 86.45±0.47 | 86.01±0.49 | 83.81±0.52 |
> > | **128** | **85.08±0.46** | 88.92±0.43 | **87.78±0.45** | **87.32±0.47** | **84.95±0.49** |
> > | **256** | 85.03±0.47 | **89.08±0.44** | 87.51±0.51 | 87.15±0.48 | 84.82±0.51 |
> >
> > Table: AUROC (%) Performance on MUTAG - Single Best Bandwidth (h=0.1)
> >
> > | Hidden Dim | 1 Layer | 2 Layers | 3 Layers | 4 Layers | 5 Layers |
> > |:---:|:---:|:---:|:---:|:---:|:---:|
> > | **32** | 81.43±0.53 | 84.12±0.50 | 83.24±0.52 | 82.81±0.54 | 81.01±0.56 |
> > | **64** | 82.67±0.51 | 85.58±0.48 | 84.72±0.50 | 84.29±0.52 | 82.45±0.54 |
> > | **128** | 84.78±0.48 | **86.92±0.41** | **86.12±0.44** | **85.63±0.46** | **83.87±0.49** |
> > | **256** | **84.91±0.49** | 86.85±0.43 | 86.03±0.45 | 85.48±0.47 | 83.72±0.51 |
> >
> > Above additional results combined with original ablation study decisively demonstrate that **observed gains come from the learnable multi-scale KDE component, not simply from additional GNN parameters**: the learnable multi-scale KDE adds only 5 bandwidth mixture weights yet provides 4-6% AUROC gain over single bandwidth and 2-3% over fixed weights, while even double more GNN parameters via layers or hidden dims cannot compensate for the lack of learnable KDE. Our theoretical analysis (Theorems 4.1-4.2) also establishes that the learnable multi-scale KDE achieves minimax-optimal convergence rate $O(N^{-0.8})$, further support our design instead of increasing the GNN backbone's parameters.

---

> > > ### Author Response · Authors · 2025-11-21
> > > **Rebuttal Part III**
> > >
> > > ### **Response and clarification on your mentioned Weaknesses**
> > >
> > > - **Novelty and related work concerns**: As stated above in our Common Response, LGKDE's contributions go significantly beyond combining existing components; we have established a comprehensive theoretical framework for LGKDE. It addresses a fundamental gap in graph learning: principled, learnable density estimation with theoretical guarantees, and provides a view of solving the GAD task via density estimation.
> > > We have expanded our Related Work section in Appendix C.2, more specifically,
> > >
> > >     | Method | Limitation for Graphs | LGKDE Advantage |
> > >     | :--- | :--- | :--- |
> > >     | **Normalizing Flows (GraphNF)** [1] | Require invertible architectures; hard to handle **variable graph sizes** and topology changes. | **Naturally handles variable sizes** via GNN+MMD; no invertibility constraint. |
> > >     | **Energy-Based Models (GraphEBM)** [2] | Rely on expensive **MCMC sampling** (Langevin dynamics) for training/inference. | **Direct density estimation**, effective and stable. |
> > >     | **GraphVAE** [3] | Optimization lower bound (ELBO) is often loose; "Posterior collapse" issues. No explicit density parameterization. | **Exact consistency guarantees** (Theorem 4.1) and convergence (Theorem 4.2). |
> > >
> > >     - **Where the density lives.**
> > >         - Flows parameterize an explicit density via invertible mappings in a latent Euclidean space and typically require fixed-size inputs or carefully designed coupling layers.
> > >         - VAEs and EBMs define densities or energies in a latent space and rely on a decoder or MCMC for generation.
> > >         - **LGKDE, in contrast, keeps the density directly in a nonparametric form over graphs**, represented as a KDE in a **learned MMD metric space**. The GNN + MMD part only defines the metric; the density is directly from the KDE itself.
> > >
> > >     -  **What is learned jointly.**
> > >         - Many existing graph models learn an embedding or score function but then apply density estimation as a separate post-processing step.
> > >         - LGKDE jointly optimizes the graph encoder, the deep MMD metric, and the multi-scale KDE mixture weights using the density-contrast objective. This is closer in spirit to classical kernel methods with learned metrics, but fully end-to-end on graphs.
> > >
> > >     - **Theoretical guarantees.**
> > >         - We have established theoretical guarantees for LGKDE, including consistency (Theorem 4.1), convergence (Theorem 4.2), robustness (Theorems 4.3, 4.4 and Corollary 4.5), generalization (Theorem 4.6). To the best of our knowledge, rare existing graph-level deep density estimators could provide such complete and transparent theoretical analysis like ours.
> > >         - LGKDE bridges deep graph representation learning and nonparametric KDE theory, which is precisely the gap we aim to fill.
> > >
> > >
> > > - **Result analysis enhancement**: We have strengthened our analysis in the revised paper, connecting empirical results more explicitly to theoretical insights.
> > >
> > >
> > > *Reference*
> > >
> > > [1] Jenny Liu et al. Graph normalizing flows. NIPS, 2019.
> > >
> > > [2] Meng Liu et al. Graphebm: Molecular graph generation with energy-based models. ICLR Workshop, 2021.
> > >
> > > [3] Simonovsky et al. Graphvae: Towards generation of small graphs using variational autoencoders. ICANN, 2018.
> > >
> > > ---
> > > **Thanks again for your expertise and efforts in reviewing our paper. We are deeply grateful for your constructive review and hope our response can address your questions and concerns.**

---

### Author Response · Authors · 2025-11-21
**General Response to All Reviewers**

**We are deeply grateful to all reviewers for your invaluable time, meticulous attention, and the exceptionally constructive feedback you have provided.** The recognition of our theoretical rigor, the breadth of our experiments, and the significance of the problem we tackle is both humbling and profoundly encouraging.

In addition to our **detailed point-by-point responses**, we **provide the general response here to summarize the rebuttal**.  We acknowledge that our original submission included extensive supplementary materials (a 28-page Appendix), which may have made certain pre-existing theoretical analysis and empirical results difficult to locate quickly and potentially easy to overlook. Therefore, in addition to conducting further experiments and providing clarifications in the rebuttal, we have revised the manuscript accordingly. **All revised and updated content is highlighted in blue in the updated PDF.**

---

| **Question&Concerns** | **Key Evidence in Original Submission** | **Rebuttal Action & Revisions** |
|:---|:---|:---|
| **Novelty** (3Fbp, MSxu, xSCo) | (1) **First end-to-end learnable graph density estimation framework**. (2) **Novel energy-based spectra perturbation** (Eq 3-4) is not random augmentation, **combined with density-based loss** (3) **Complete theoritical framwork** prove consistency, convergence, robustness, and generalization bound (4) **Extensive empirical validation** and achieve **superior performance** on density estimation and graph anomaly detection tasks  | **Apart from detailed response, we enhanced related work discussion in Appendix C.2**: Explicit comparison with Normalizing Flows, EBMs, VAEs showing LGKDE's advantages (variable graph sizes, no MCMC, exact guarantees); **Clarification our design choice with existed and extend experiment results**; **Moving generalization bound (Theorem 4.6) and its analysis from Appendix F.5 to Sec 4**. |
| **Scalability** (Y81R, MSxu) | (1) **Theoretical complexity analysis** (Sec 3.4, Appendix E.4.1, Table 6), Memory $O(NLnd)$ matches standard GNNs; (2) **Empirical running time comparison** (Appendix E.4.2, Table 12): Complete runtime on COLLAB (5,000 graphs), LGKDE 1.5× faster than SIGNET, competitive with advanced baselines; (3) **Scalability** (Sec 3.4, Appendix E.4.3, Table 13): Sampling strategies achieve speedup with negligible performance loss | we **give further detailed respose and clarification**, improved appendix navigation. |
| **Bandwidth** (3Fbp, MSxu) | **Ablation study has covered these points** (Sec 5.3, Appendix B.4, Figure 6, Table 6) | (1) **Clarified learnable mixture weights approach** (no manual tuning). (2) **Provided theoretical and practical justification** (3) Add revision in Sec 5.3. |
| **Perturbation Module Ablation** (3Fbp, xSCo) | **Appendix D.2**, Figures 10-12: case studies on different $(\tau_1, \tau_2)$ combinations with edge change ratio distribution; **Density-contrast motivation (Sec 3.3)**: Soft density targets avoid hard anomaly labels, naturally paired with structure-aware perturbations | We have **conducted addtional experiments, (1) Appendix B.4**, Table 7: **Systematic comparison of feature-only, spectral-only, random edge, and combined perturbations; (2) Appendix B.4, D.2** Table 8 & 15: **Comprehensive $\tau_1, \tau_2$ grid-search with performance and edge modification statistics report** (3) Add revision in Sec 5.3.|
| **$d_{int}$ analysis** (xSCo, MSxu) | (1) **Appendix F.3 has discussed it** and it's a design choice for density estimation, not a graph space restriction.| We **provide further detailed responses and clarifications, and improved appendix navigation and content**. |
| **Distribution Shift Validation** (xSCo) | (1) **Existing robustness and training contamination study in Appendix B.1, B5.4.** (2) **Density recovery on synthetic graphs in Section 5.1 and Appendix B5.1-5.2.** | We have **conducted additional experiments, Add Appendix B.6**, Table 14, **further validate it**; we also add corresponding revision in Sec 5.3. |
| **LGKDE design validation** (3Fbp) | **Ablation study has covered these points (Sec 5.3, Appendix B.4, Figure 6, Table 6)** | We have **conducted additional experiments Appendix B.4**, Table 9: Parameter-controlled experiments **isolating KDE contribution from increased GNN backbone params**; we also add corresponding revision in Sec 5.3. |

---
**Thanks again, hope our comprehensive rebuttal response addresses all your questions and concerns, and we look forward to your kind and insightful feedback.**

---

### Author Response · Authors · 2025-12-01
**Rebuttal Summary to AC**

Dear Area Chair,

We sincerely thank you for reviewing our paper and for your further engagement. Here we provide this **summary of our rebuttal process and reviewer engagement to assist your further evaluation**. **All reviewer interactions documented below occurred before the OpenReview API Bugs** (rating from 6,6,4,4  → 6,6,6,4), which we believe represent genuine, constructive academic discourse under normal peer review conditions.

---
### **Summary of Reviewer Responses and Ratings**

Before the API Bugs (OpenReview Official time point: **Nov 27, 2025  03:09 AM AoE**), **Two reviewers (MSxu, Y81R) explicitly confirmed their concerns were addressed and supported the acceptance, with one (Y81R) raising its initial rating**:

| **Reviewer** | **Initial Rating** | **Rating Before OpenReview Bug** | **Key Response & Timestamp (AoE)** | **Rebuttal Actions & Outcomes** |
|:---:|:---:|:---:|:---|:---|
| **MSxu** | 6 | **6 (Unchanged)** | Quote raw response: *"Thanks to the authors for providing the requested clarifications. **I am now in favor of accepting this work** and encourage the authors to incorporate these clarifications into the revised version of the manuscript for the benefit of future readers."* (**Nov 24, 18:23 AoE**) | **Explicitly supports acceptance**; All concerns addressed, including bandwidth selection, spectral perturbation quantification, and theoretical guarantees |
| **Y81R** | 4 | **6 (+2)** ↑ | Quote raw response: *"Thanks for the reply. **Most of my concerns have been addressed. I will improve my score.** "* (**Nov 26, 16:41 AoE**) | **Score raised (+2)**; Scalability, threshold sensitivity, and perturbation quality concerns resolved with clarification and additional experiments |
| **3Fbp** | 6 | **6 (Unchanged)** | **No response before OpenReview Bug** and allowing reviewer to post comment | **All concerns addressed with detailed response** including novelty clarification, bandwidth sensitivity ablation, and parameter-controlled experiments isolating KDE contribution |
| **xSCo** | 4 | 4 (Unchanged) | **No response before OpenReview Bug** and allowing reviewer to post comment | **All concerns addressed** with distribution shift experiments, $d_{\text{int}}$ clarification, and perturbation ablation; Follow-up reminder of us posted Nov 27, 18:06 AoE however at that time the system did not allow reviewer to post comment. |

---

### **Cross-Reviewer Validation of Common Concerns that have been substantively addressed**

As listed in our initial General Response for all reviewers, importantly, **concerns raised by non-responding reviewers (3Fbp, xSCo) were substantively identical to concerns raised by responding reviewers (MSxu, Y81R), which have been explicitly validated as resolved and acknowledged by the two reviewers who engaged during rebuttal.**

| **Common Concern** | **Raised By** | **Validated Resolution By** |
|:---|:---:|:---:|
| Perturbation sensitivity (τ₁, τ₂) | Y81R, 3Fbp, xSCo | **Y81R confirmed** ("concerns addressed") |
| Scalability and efficiency | Y81R, MSxu | **Y81R confirmed** ("concerns addressed") |
| Bandwidth selection procedure | MSxu, 3Fbp | **MSxu confirmed** ("in favor of accepting") |
| Ablation on perturbation module | 3Fbp, xSCo | **Y81R confirmed** via perturbation quality discussion |
| Novelty and related work comparison| 3Fbp, MSxu, xSCo | **MSxu confirmed** ("in favor of accepting") |

---

We believe the substantive pre-incident reviewer engagement demonstrates the scientific merit of our work. **Two reviewers explicitly confirmed their concerns were resolved, with Y81R raising his/her rating (4 → 6, overall 6,6,6,4) and MSxu explicitly stating support for acceptance**. The **remaining reviewers' concerns overlap substantially with those already validated**, and **we have provided comprehensive point-by-point responses with additional experimental evidence and corresponding paper revision.**

We are extremely grateful if you could consider the following facts:
1. The **explicit positive feedback and support for acceptance** from MSxu and Y81R **before the OpenReview Bug**
2. The **cross-validation** of common concerns have been addressed through engaged reviewers
3. Our **complete responses and corresponding updated manuscript** addressing all raised concerns

For detailed responses and evidence, please refer to:
- **General Response to All Reviewers** (Nov 20, 17:22 AoE)
- **Point-by-point Individual Reviewer Responses** (All posted before Nov 20, 17:54 AoE)
- **Updated manuscript PDF** with $\textcolor{blue}{blue}$-highlighted revisions (Nov 20, 21:36 AoE)

**We deeply appreciate your time, further engagement, and expertise in reviewing our work. Feel free to reach out if you have any questions or need further clarification.**

Sincerely and respectfully,

The Authors

---

### Author Response · Authors · 2025-12-03
**Final Remarks From Authors**

Dear Area Chair and Reviewers,

As the rebuttal discussion phase draws to a close, we thank the reviewers for their feedback and the Area Chair for your further engagement. Below we provide a brief **summary and final remarks to our paper and the whole rebuttal process**.

---
We propose **LGKDE**, the **first end-to-end learnable kernel density estimation framework for graphs**, which utilizes a learnable multi-scale design on a joint learned MMD metric space to capture graphs' structural patterns and semantic variations for effective density estimation while maintaining theoretical guarantees. It's a principled framework that bridges deep graph representation learning with adaptive kernel density estimation to ***tackle the fundamental challenge of modeling the probability density function of graph-structured data.***

**1. Reviewers recognized our contributions**, highlighting that our work:
   - *"tackles a fundamental yet underexplored problem of graph density estimation"* (xSCo, Y81R, 3Fbp)
   - proposes a *"theoretically grounded framework"* with *"non-trivial theoretical analyses (consistency, convergence rate, and robustness bounds)"* (MSxu, xSCo)
   - achieves *"consistent improvements over competitive baselines"* through *"comprehensive empirical validation"* (Y81R, MSxu, 3Fbp)

**2. Reviewers' concerns** are mainly on scalability and extra ablation studies (perturbation sensitivity, bandwidth selection, parameter isolation). These are **straightforward to address**: we **clarified existing results** (some in appendices that may have been overlooked) and **conducted additional experiments**. **All concerns have been addressed with point-by-point responses and the corresponding manuscript revisions**.

**3. Expected ratings if discussion had continued:** Before the OpenReview Bug (Nov 27, 03:09 AM AoE), **two reviewers explicitly confirmed resolution and supported acceptance** (MSxu: *"in favor of accepting"*; Y81R: *"most concerns addressed"*, raised 4→6). Based on concern overlap and cross-validation:

| **Reviewer** | **Initial** | **Before Bug & Locked** | **Rationale & Concern Addressed** |
|:---:|:---:|:---:|:---|
| **MSxu** | 6 | **6** | Explicitly *"in favor of accepting"*; no outstanding concerns |
| **Y81R** | 4 | **6** | Stated *"most concerns addressed"*; already raised score |
| **3Fbp** | 6 |  6 (**No Response**) | All concerns (bandwidth, perturbation, KDE contribution) overlap with MSxu/Y81R's resolved concerns |
| **xSCo** | 4 | 4  (**No Response**)| All concerns addressed via new experiments (Table 7, 14) and clarifications; shared concerns validated by Y81R |

**So if there were further engagement with unresponsive reviewers (No OpenReview bug and the full discussion continues), we expected our rating scores could become 6,6,6,6 or higher, like 6,6,8,6.**

For **complete details and facts**, please refer to our following **Rebuttal Summary to AC** (Nov 30, 22:17 AoE), **General Response** (Nov 20, 17:22 AoE), **individual reviewer response threads**, and **updated manuscript pdf** with $\textcolor{blue}{blue}$ highlights.

We deeply appreciate all reviewers and AC for your time, engagement, and expertise in reviewing our paper.

Sincerely and respectfully,

The Authors

---

### Meta-Review · Area_Chair_6LSW · 2026-01-06

**Summary:**

In this submission, the authors proposed a learnable KDE method for graph representation that leverages and combines existing techniques (GNN, MMD, and multi-scale KDE) within a graph-oriented KDE framework. Experiments show the effectiveness of the proposed method compared to baselines. The method's complexity analysis is also provided. The main concerns of reviewers include 1) the technical novelty of the method and 2) the lack of some analytic content and experiments. In the rebuttal phase, the authors made efforts to add more explanations and experiments, which partially solves the concerns. However, I still have some questions after reading the paper and the discussions.

1. The authors analyze the complexity of the method to verify its scalability to the number of graphs. However, the complexity is quadratic to the number of nodes because of the usage of MMD, which implies the feasibility of the method for large-scale graphs. In most practical situations, the real-world graphs are not as simple as those in TUDataset.

2. The authors emphasized that the proposed method is motivated by graph anomaly detection. However, treating the minority class of TUDataset as a graph anomaly mismatches with practical anomaly detection tasks. Although some existing papers use such a setting, but in my opinion, this experimental setting is questionable. If the authors really want to focus on anomaly detection tasks, more solid experiments are necessary.

Therefore, I think this work needs one more round of review.

**Reviewer Concerns:**

The authors provided more analytic content, however, the limited novelty, as a main concern, is still outstanding.

**Reviewer Scores:**

I think Reviewer Y81R would have increased his/her score, and the other reviewers would have maintained their scores.

---

### Decision · Program_Chairs · 2026-01-26

Reject